## Replication

behaviour/cognition

bonobos, chimpanzees, cooperation, co-feeding, social tolerance, replication

**Author for correspondence:**
Suska Nolte
e-mail: suskanolte@gmx.de

# Does tolerance allow bonobos to outperform chimpanzees on a cooperative task? A conceptual replication of Hare *et al.*, 2007

Suska Nolte[1,2], Elisabeth H. M. Sterck[1,3] and Edwin J. C. van Leeuwen[1,2]

[1]Animal Behaviour and Cognition, Department of Biology, University Utrecht, Utrecht, The Netherlands
[2]Department of Comparative Cultural Psychology, Max Planck Institute for Evolutionary Anthropology, Leipzig, Germany
[3]Animal Science Department, Biomedical Primate Research Centre, Rijswijk, The Netherlands

SN, 0000-0002-2614-792X; EJCvL, 0000-0002-7729-2182

Across various taxa, social tolerance is thought to facilitate cooperation, and many species are treated as having species-specific patterns of social tolerance. Yet studies that assess wild and captive bonobos and chimpanzees result in contrasting findings. By replicating a cornerstone experimental study on tolerance and cooperation in bonobos and chimpanzees (Hare *et al.* 2007 *Cur. Biol.* **17**, 619–623 (doi:10.1016/j.cub.2007.02.040)), we aim to further our understanding of current discrepant findings. We tested bonobos and chimpanzees housed at the same facility in a co-feeding and cooperation task. Food was placed on dishes located on both ends or in the middle of a platform. In the co-feeding task, the tray was simply made available to the ape duos, whereas in the cooperation task the apes had to simultaneously pull at both ends of a rope attached to the platform to retrieve the food. By contrast to the published findings, bonobos and chimpanzees co-fed to a similar degree, indicating a similar level of tolerance. However, bonobos cooperated more than chimpanzees when the food was monopolizable, which replicates the original study. Our findings call into question the interpretation that at the species level bonobos cooperate to a higher degree because they are inherently more tolerant.

# 1. Introduction

Many theories that assess selective pressures and proximate mechanisms underlying human cooperative evolution are built around the idea that bonobos (*Pan paniscus*) and chimpanzees (*Pan troglodytes*) differ substantially in terms of their tolerance levels and cooperative abilities (e.g. [1–3]). Research with wild and captive *Pan* populations, however, results in mixed and sometimes contradicting findings with respect to which species behaves more tolerantly and cooperatively. Before justifying and describing our replication study of a cornerstone comparison of social tolerance and cooperation in bonobos and chimpanzees [4], we provide a brief sketch of the current data from wild and captive settings.

Wild populations of chimpanzees exhibit collective behaviours such as boundary patrols, territorial defence, group hunting and subsequent meat sharing (e.g. [5–7]). Both species form coalitions and support their partners during fights, which usually occurs among males in chimpanzees and among females in bonobos (e.g. [6,8–10]). Even though wild bonobos are rarely seen to perform actions such as boundary patrols and collective territorial defence [11–13], there is a growing body of evidence that they are capable of group hunting, albeit do so much more infrequently than chimpanzees [14–16]. Similar to chimpanzees, bonobos share meat and fruits (e.g. [8,12,15–20]). In stark contrast to rather aggressive and lethal intergroup encounters in chimpanzees, bonobo females [11,21] and males [22] actively maintain tolerant intergroup encounters and even share food with members of neighbouring communities [14,23]. Thus, in the wild, chimpanzees and bonobos both engage in collective behaviours, however, they differ in that chimpanzees exhibit more varied behaviours in aggressive and competitive contexts, while bonobos even share food with individuals outside their own group, engage in less collaborative hunting, and do not engage in lethal territorial defence.

In captivity, chimpanzees' and bonobos' tolerance levels and cooperative abilities have been scrutinized under various circumstances (reviewed by e.g. [24,25]). Based on such research we know that chimpanzees can instrumentally help humans and conspecifics [26–29], provide benefits to a conspecific [30], and understand the role of their partner during cooperation [31]. Captive bonobos have been observed to instrumentally help conspecifics [32], share food with other group members [33], and, corroborating research from the wild, also help obtain and share food with outgroup members [34,35]. Given the stark methodological differences across these studies, clear species differences cannot be derived based on single-species studies alone. Studies that directly compare the behaviour of the two species are scarce. Four studies found that bonobos were more cooperative and prosocial than chimpanzees, i.e. they co-fed on and cooperatively obtained food or transferred tools and tokens more often than chimpanzees ([4,36–38], with the latter publication showing differences in co-feeding but not sharing). Three other studies found that both species did not act prosocially, i.e. transferred tools or gave access to food [39–41]. By contrast, four additional studies found that chimpanzees acted more tolerantly and prosocially than bonobos, i.e. they distributed monopolizable food more tolerantly [42–44] and more proactively than bonobos [43], and tolerantly co-fed with more group members than bonobos [44–46]. Taken together, the picture emerges that, in captive settings, bonobos may be more adept or inclined to cooperate and help than chimpanzees, while chimpanzees may be more inclined to tolerantly or proactively distribute monopolizable resources (though see [4]; and co-feeding scores in [38]). Importantly, however, all except one study [44] found chimpanzees to be more prosocial or tolerant than bonobos employed group instead of dyadic tests. Thus, it is unclear to what extent group factors influence such differences.

To understand the origin of discrepancies in results, one tool is replicating previous studies and assessing whether outcomes are reproducible [47,48]. We currently do not know whether some of the discrepancies concerning *Pan* tolerance and cooperation might be due to noise, the study context, or intraspecific variation. In case the discrepant results of studies described above can be reproduced, we need to understand whether bonobos and chimpanzees indeed show different, and even contradicting, levels of tolerance and cooperation depending on the context, and what explanation underlies such a result. Moreover, replication experiments can be used to understand whether a species shows indications of cultural intergroup variation, i.e. 'behavioural variation across groups owing to social learning within groups', meaning that groups of the same species might differ markedly in terms of their behaviour such as social tolerance [49].

Here, we conceptually replicated a seminal study by Hare and colleagues ([4], *Cur. Biol.* 17, 619–623. doi:10.1016/j.cub.2007.02.040), who found that bonobos co-fed more often than chimpanzees on both monopolizable and sharable food resources and additionally outperformed chimpanzees on a cooperative task when the food was monopolizable. Based on these outcomes the researchers argued that bonobos' heightened tolerance levels enable them to cooperate even under potentially conflict-

inducing conditions. The researchers employed a co-feeding and a loose-string task [50] that we closely replicated. As in the original study, each pair was first presented with the co-feeding task and subsequently with the cooperation task. In both experiments, we placed food on a wooden 2.7 m long platform that was situated outside of the apes' cages and out of reach of the apes. Food was placed either at both ends of the platform or in the middle. In the co-feeding task, two experimenters pushed the platform toward the mesh once the two apes were located at equidistant locations right and left of the platform. The task therefore only assessed whether the dyad co-fed, meaning both could take at least one piece of food. In the cooperation task, a loose rope was threaded through metal loops and 20 cm of both ends were placed into the apes' cage at a distance that no single ape could pull the rope alone. Only by pulling at both ends of the rope could the platform be moved and pulled toward the mesh in order to reach the food. Hence, apes had to cooperate to get any food in this task. We were not allowed to replicate the amounts of food used in the original study owing to the current management restrictions on the apes' food intake. To replicate the original study by Hare *et al.* [4] as closely as possible and enable meaningful comparisons across conditions, we used the smallest amount of food used in the original study [4] and applied it to all of our test conditions. We recorded aggressive, playful and socio-sexual contact during both experiments. In order to rule out effects of housing conditions, we tested chimpanzees and bonobos that were housed at the same facility.

# 2. Methods

## 2.1. Subjects

We studied one group of 12 bonobos (4 adult females, 3 adult males, 5 individuals younger than 7 years) and one group of 19 chimpanzees (11 adult females, 4 adult males, 4 individuals younger than 7 years; see electronic supplementary material, table S1 for details). Both groups were housed at the Wolfgang Koehler Primate Research Center in Zoo Leipzig, Germany, and have extensive experience with various cognitive and behavioural tests. Water was provided ad libitum during the tests and at no point was any ape food deprived.

The number of individuals that participated in the co-feeding experiment was 9 bonobos (4 females, 5 males, $M_{age} = 18.9$) and 16 chimpanzees (10 females, 6 males, $M_{age} = 25.9$). We tested 11 unique dyads of bonobos; five adult female, four adult male, and two mixed dyads composed of females with their juvenile sons. Given that none of the bonobos in the group are maternally related, all dyads except the two mother-juvenile dyads were non-kin. One additional bonobo adult female dyad could not be tested because of immediate fighting. Due to current animal handling regulations for the bonobos, no adult mixed pairs could be tested. We tested 89 unique dyads of chimpanzees; 41 adult female, six adult male and 42 mixed dyads out of which two dyads were composed of females with their juvenile sons. The adult dyads encompassed five female–offspring dyads (3 male, 2 female offspring) and two maternal sibling dyads (1 male–male, 1 female–female). We needed to stop testing two pairs, one adult female and one adult mixed dyad, due to an unacceptable level of aggression or fear by one partner. Another four adult female chimpanzee dyads could not be tested because of immediate fighting or one partner being afraid of the other. The number of individuals that participated in the cooperation experiment was a subset of 7 bonobos (4 females, 3 males, $M_{age} = 22.5$) and 11 chimpanzees (7 females, 4 males, $M_{age} = 30.4$). We tested 9 unique dyads of adult bonobos and 53 unique dyads of adult chimpanzees. It was not possible to test the mother–juvenile offspring dyads for both species. Thus, none of the bonobos were related and we could only test four of the chimpanzee female–adult offspring dyads (2 male, 2 female offspring) and the two maternal sibling dyads (1 male–male, 1 female–female).

## 2.2. Apparatus

We used a wooden platform (17 cm × 270 cm) on top of which we installed three wooden dishes (17 cm × 27 cm each) located in the middle and at either end of the platform (figure 1). We fixated two metal loops on either side of the platform through which we could thread a rope that would span the entire length of the platform and reach into the apes' room on both outer sides of the platform. We used a slightly shorter platform than was used in the original study given that the local rooms did not allow for a length of 3.4 m. The distance between the food dishes was therefore slightly narrower as in the original study (i.e. 2.2 m instead of 2.7 m apart, table 1), but the two rope ends could still not be reached simultaneously by one individual alone.

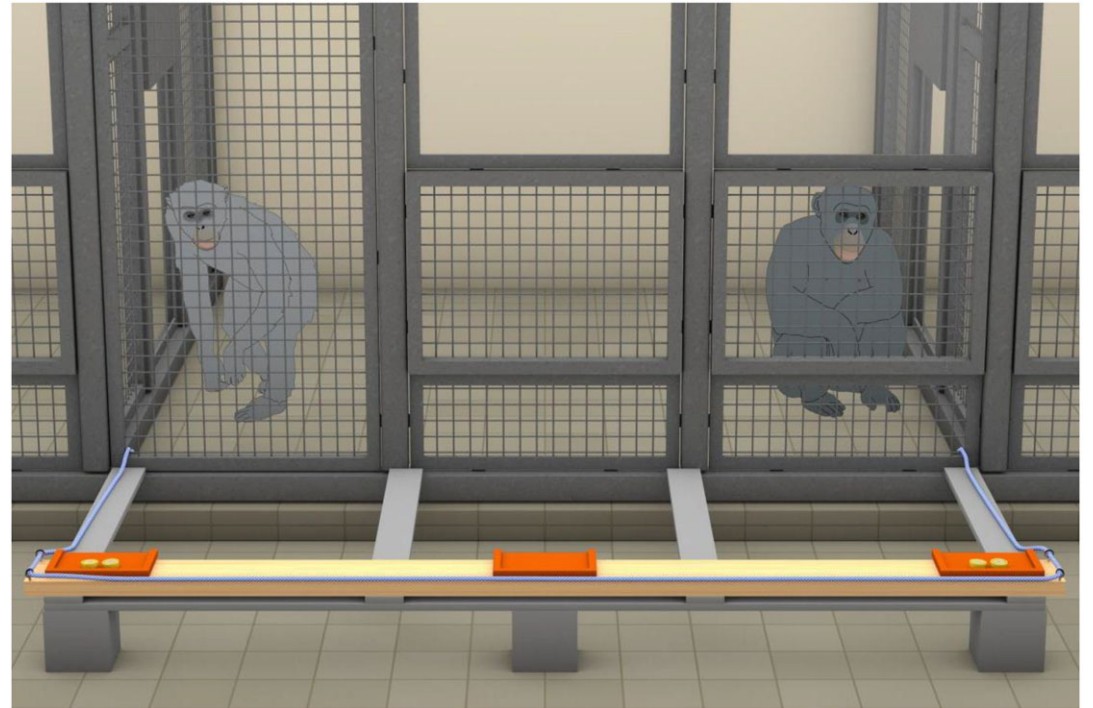

**Figure 1.** In the cooperation task, the two apes had to pull at both ends of the rope to pull in the platform. In the co-feeding task, no rope was used and, instead, two experimenters pushed the platform toward the mesh. The depiction of the cage mesh is representing the actual scale, the width of the platform was 2.7 m and its initial distance to the mesh was 1 m.

## 2.3. Design

We conducted two out of three experiments of the original study ([4], further called 'Hare2007'): The co-feeding test ('Experiment 1') and the second cooperation test ('Experiment 3'). In 'Experiment 2', the original authors conducted the cooperation test with shareable food distribution only and replicated both the methods and results in 'Experiment 3' while additionally incorporating a condition with monopolizable food distribution. We, therefore, did not conduct their first cooperation test ('Experiment 2') but replicated 'Experiment 3' that encompassed both conditions. Hare2007 used a within-subject design for comparing the co-feeding ('Experiment 1') and first cooperation test ('Experiment 2'), but a between-subject design for the second cooperation test ('Experiment 3'). We replicated the within-subject design, which enabled us to also compare co-feeding and cooperation with monopolizable food distribution.

Both experiments took place in a dyadic setting. Each dyad started with the co-feeding test before continuing with the cooperation test. After finishing both tasks with one partner, individuals were recombined with a new partner to test as many unique dyads as possible per group. To increase effective sample size, we tested apes in multiple dyads, while Hare2007 only used the same partners in two unique dyads. Importantly, all naïve apes experienced the tasks contingencies with another naïve partner first, before being paired with other individuals. This was done to ensure that neither of the partners would have a disadvantage because one already had more immediate experience with the set-up.

We tested the apes in their inside sleeping cages. In both experiments, we placed the platform outside of the apes' cages with a distance of 1 m to the cage. Depending on the condition, food was either placed at both ends of the platform or in the middle. In the co-feeding task (figure 1), two experimenters pushed the platform toward the mesh once the two apes were located at equidistant locations right and left of the platform. The food could be immediately reached by the apes once the experimenters pushed the platform to the mesh. In the cooperation task (figure 1), a loose rope was threaded through the metal loops and 20 cm of both ends were placed into the apes' cage at a distance of 2.7 m. In case the apes only pulled at one side of the rope, it slid through the metal loops and the platform remained at its initial location. Only by pulling at both ends of the rope simultaneously could the platform be moved and pulled toward the mesh. Hence, in this task, the apes had to cooperate in order to get any food. The mesh at the current facility was narrower than that in Hare2007, wherefore the apes could not stick their entire arm through the mesh to grab the food but retrieve it with their fingers (table 1). Different to Hare2007, bonobos did not play with the rope, except for one session during which one

**Table 1.** Methodological differences between the original and the current study.

| | Hare *et al.* [4] | current study |
|---|---|---|
| **cooperation platforms** | tray length is 3.4 m wide (feeding dishes 2.7 m apart).[a] | tray length is 2.7 m wide (feeding dishes 2.2 m apart) – subjects required to co-feed and cooperate in closer proximity. |
| **food amount** | *co-feeding task* tested with large amounts of food (0.5 kg of banana each trial). | only tested with small amounts of food (four 1.5 cm pieces of banana each trial, replicating what was originally used in the clumped condition) in *both experiments*. For additional information see electronic supplementary material, table S2. |
| | *cooperation task* tested with large amounts of food in the dispersed condition and small amounts of food in the clumped condition (0.5 kg of banana each trial versus four 1.5 cm pieces). | |
| **familiarization rope & individual training** | *familiarization rope*: bonobos received time to play with rope to reduce play behaviour afterwards | *familiarization rope*: none administered, no play behaviour |
| | *training*: one session of six trials | *training*: up to five sessions with multiple trials |
| | *criteria to pass into test*: none, all tested | *criteria to pass into test*: six successful consecutive trials in one session |
| **cooperation 'warm up'** | 'three-trials-success' criteria of successful cooperation with the given partner in condition *dispersed divisible* (up to 13 trials per dyad) provided to bonobos at both facilities and chimpanzees at WKPRC before administering cooperation experiment (Exp. 3) that included condition *dispersed divisible* and *clumped*. | no dyadic 'warm-up' to either species, only solitary pre-test trials for rope-pulling understanding provided to both species (see above) |
| **accessibility of food** | width of mesh allowed apes to reach for food by sticking their arm through the mesh.[a] | width of mesh was narrower: apes could reach for food by sticking their fingers though the mesh. |

[a]No published information provided for WKPRC apes.

female took the rope three times to play with it after her partner refused to cooperate. She gave back the rope twice after one and once after 3 min, and we could continue testing the remaining trials. Testing was therefore not hindered by play behaviour involving the rope.

All distinctions in methodology between the current and original study are outlined in table 1, based on which the current study represents a conceptual rather than faithful replication of Hare2007. Furthermore, we provide details on the population and sampling differences between the two studies that can aid our understanding of the generalizability of species differences (table 2).

## 2.4. Co-feeding experiment

### 2.4.1. Test procedure

The co-feeding task was used to assess tolerance levels between different partners in a dyadic setting when food is immediately available and can be monopolized by either one of the partners. Each dyad participated in six trials that we conducted on the same test day (session). Before the start of each trial, we placed and baited the platform at a distance of 1 m, to ensure that the apes could not grab the food immediately. The two experimenters then called the two apes to equidistant locations right

**Table 2.** Population and sampling differences between the original and the current study.

| | Hare et al. [4] | current study |
|---|---|---|
| **chimpanzee subjects** | subspecies *P. schweinfurthii* at Ngamba and *P. verus* at WKPRC. | subspecies *P. verus* and four females are *verus-schweinfurthii* hybrids. |
| **sample procedure** | Exp. 1 & 2:<br>*Lola ya Bonobo*: bonobos selected from several populations of 12–20 individuals.<br>*WKPRC*: bonobos lived together in social group.<br>*Ngamba Island*: Chimpanzees selected from one population of 39 individuals.<br>Exp. 3:<br>most tolerant chimpanzee dyads selected of previous population and additionally three tolerant dyads at WKPRC selected. Bonobo pairs of previous population were selected to match the age and sex of the most tolerant chimpanzee pairs. | *WKPRC*: both species sampled at same facility. Bonobos were selected from population of 12 individuals, and chimpanzees from population of 19 individuals. In both experiments, as many dyad combinations were sampled as was allowed by the current handling regime. Data collection was immediately stopped if either partner showed signs of fear toward being released into the same room with the other (see §2.1.). |
| **experimental history** | bonobos and chimpanzees experimentally naïve in Exp. 1 (co-feeding task) and Exp. 2 (cooperation task, condition dispersed divisible). | both species have extensive general experience in participating in tests, with experience across various cooperation tasks, and are trained to be separated from group. |
| **environment** | *sanctuaries*: 5–40 ha of primary tropical forest.<br>*WKPRC*: bonobos have access to indoor and outdoor enclosure (2600 m²). No information provided on chimpanzees (however see 'Current study'). | both species live in indoor enclosure (430 m² chimpanzees, 256 m² bonobos) and have access to outdoor enclosure (4000 m² chimpanzees, 2300 m² bonobos) during warm months. |
| **mean age and range** | *Exp. 1 & 2*:<br>bonobo: 9.6 years (range: 5–22);<br>chimpanzee: 11.7 years (range: 4–21)<br>*Exp. 3*:<br>bonobo: 7.9 years (range: 5–20)<br>chimpanzee: 7.5 years (range: 5–12) | *co-feeding experiment*:<br>bonobo: 18.9 years (range 6.2–36.5)<br>chimpanzee: 25.9 years (range 3.2–43.5)<br>*cooperation experiment* (sub-sample based on passing solitary training, only administered to adult apes):<br>bonobo: 22.5 years (range 13.6–36.5)<br>chimpanzee: 30.4 years (range 15.1–43.5)<br>age is incorporated in the statistical models. |
| **kinship** | selection of only non-kin dyads. | some dyads related; incorporated in the statistical models. |
| **test location** | 15 m² testing room and open space in sanctuaries, 9 m² testing room at WKPRC. | 7 m² testing room for bonobos with available adjacent rooms of 6.3 m² to left and of 11.3 m² to right. 13.32 m² testing room for chimps with available adjacent rooms of 6.8 m² to left and right. |
| **dyad composition** | in Exp. 3 (cooperation task) same partners used in two unique dyads. | to increase effective sample size, we tested apes in multiple dyads. |

(a) dispersed divisible

(b) clumped divisible

(c) clumped

2.7 m

**Figure 2.** Conditions used in the two experiments. Following Hare and colleagues [4], all three conditions were included in the co-feeding experiment, while only (a) and (b) were included in the cooperation experiment. Circles symbolize banana pieces of size 1.5 cm in (a) and (b) and 3 cm in (c).

and left of the platform in order to give both the same chance at moving to and taking the food. The platform was pushed to the mesh within reach of the apes only when both apes were in front of the two experimenters. A trial lasted 1 min or until the food was consumed. After each trial, the experimenters pulled back the platform, re-baited it and called the apes to their starting positions.

In the original study, two trials of three different conditions were administered in a counterbalanced order across dyads. We adapted the amount of food used in Hare2007 (table 1 and electronic supplementary material, table S2, Co-feeding), as the feeding regulations of the zoo did not allow us to conduct six trials in which an ape would have consumed a minimum of 1.5 kg and a maximum of 3 kg of fruits across trials. Instead, we followed what was originally done for condition 'clumped-divisible' of the cooperation task (see electronic supplementary material, table S2, Cooperation), and for the remaining conditions applied the same logic of dividing the food as in Hare2007 (e.g. food that was clumped in the middle was divided in half and placed at the outer sides). We therefore used the following conditions (figure 2a–c) with a counterbalanced order across dyads:

1. Dispersed-divisible: both dishes were placed on the platform with a distance of 2.7 m between them, two 1.5 cm thick pieces of banana were placed on either dish.
2. Clumped-divisible: a single dish was placed in the centre of the platform and four 1.5 cm thick pieces of banana were placed on the dish.
3. Clumped: a single dish was placed in the centre of the platform and two 3 cm thick pieces of banana were placed on the dish.

### 2.4.2. Coding

All sessions were video recorded and we coded the number of banana pieces each ape obtained during a given trial. We also noted down whether playful, socio-sexual, or aggressive interactions occurred before or during the trial (see electronic supplementary material, table S4 for ethogram). To obtain interrater reliability, a research assistant coded 20% of the data per species and was blind to the procedure and hypotheses. We selected the sessions pseudo-randomly in order for rare behaviours, such as aggression, to be included all together or for a minimum of 10 randomly selected sessions. We used Cohen's $\kappa$ to evaluate whether the two raters agreed on whether on a given trial the dyads co-fed ($K = 0.95$), played ($K = 0.70$) or engaged in aggressive ($K = 0.77$) or socio-sexual contact ($K = 0.87$).

## 2.5. Cooperation experiment

### 2.5.1. Training

In Hare2007, one session with six trials was administered and the ape transferred to the test regardless of whether she showed that she understood the task. We decided to check formally if an ape understood the mechanism of the apparatus before administering the cooperation test, and only tested those individuals

that knew or learned how to pull in the platform. Each ape was trained individually and, in order for a single ape to manipulate both ends simultaneously, we used a longer rope than during testing. At the start of each trial, we placed two small pieces of banana on the dish at either end of the platform that was located at a distance of 1 m to the mesh and then placed both ends of the rope into the ape's room. We administered a maximum of five training sessions that each lasted for a maximum of 15 min. Once an individual successfully pulled in the platform over six consecutive trials within the same session (i.e. the same amount of trials as used during the test) and during which the rope ends were spaced by a distance of at least 1 m, she transferred to the test (see electronic supplementary material, table S3 for number sessions needed per ape). All seven bonobos and 11 of 14 chimpanzees passed this training (see electronic supplementary material, table S3).

### 2.5.2. Test procedure

We tested only those dyads of which both partners were knowledgeable in how to pull in the platform and that already completed the co-feeding task. In the cooperation task, we threaded the loose rope through the metal loops attached to the platform and placed the platform at a distance of 1 m. Two experimenters stood at either side of the platform and, after baiting it, simultaneously called the apes' names. Once both apes positioned themselves in front of the respective experimenter or after 20 s in case only one or neither ape approached, both experimenters simultaneously placed their side of the rope into the cage at a distance of 2.7 m. A trial ended once the apes successfully pulled in the platform, until they pulled the rope out of either loop, or after 1 min was over.

As in Hare2007, we used two different conditions but counterbalanced the order across dyads. We could retain the amount of food that was originally used in the condition 'clumped-divisible' (see electronic supplementary material, table S2, Cooperation). However, we again needed to adapt the amount of food used in the 'dispersed-divisible' condition for the reasons stated in §4.1. This resulted in the following conditions (figure 2a,b) with a counterbalanced order across dyads:

1. Dispersed-divisible: both dishes were placed on the platform with a distance of 2.7 m between them, two 1.5 cm thick pieces of banana were placed on either dish.
2. Clumped-divisible: a single dish was placed in the centre of the platform and four 1.5 cm thick pieces of banana were placed on the dish.

Six trials were administered per condition and each condition was conducted only on one day, resulting in two sessions per dyad. Therefore, per partner, each ape participated in a total of 12 trials over the course of 2 days.

### 2.5.3. Coding

All sessions were video recorded and in addition to the behaviours we coded during the co-feeding task (see §4.3.), we recorded whether the two apes cooperated during a given trial by either simultaneously pulling at both ends of the rope, or by holding steady one end of the rope while pulling the other end. As explained in §4.3., we obtained interrater reliability for each variable and also evaluated whether the two raters agreed on whether the dyads successfully cooperated ($K = 0.93$) on a given trial.

## 2.6. Analyses

To allow comparison of our outcomes with those of Hare2007, we carried out three different sets of analyses: first, we replicated the original statistical analyses as conducted by Hare2007 with our newly obtained data; second, we ran improved statistical analyses with Generalized Linear Mixed Models (GLMM; [51]) on our newly obtained data; and finally, we ran GLMMs on the original data by Hare2007 (as published in their electronic supplementary material). Whenever we included interactions between variables, we also included all lower-level effects (i.e. two-way interactions and their respective main effects) in the model.

### 2.6.1. Step 1: original statistical analyses on new co-feeding and cooperation data

We first performed the same statistical analyses that were done by Hare2007 on our newly acquired data. Therefore, we used one-sided Welch independent $t$-tests with the null-hypothesis set to what was found

by Hare2007 or hypothesized in case of non-significant original results. We also ran two-sided Welch $t$-tests for analysing whether there is a species difference in the age of the tested apes.

### 2.6.2. Step 2a: new statistical analyses on new co-feeding data

To understand what factors influenced co-feeding in our data, we fitted a GLMM (see electronic supplementary material, 1.5. for full details) with a binomial error structure and logit link function [52]. The number of observations was 587 of 100 dyads (11 bonobo and 89 chimpanzee dyads). Our dependent variable was whether the two partners of a dyad co-fed during a given trial (Yes, No), hence, a 'No' was scored whenever one partner retrieved zero pieces of banana. As explanatory predictors, we included i) the two-way interaction between species (bonobo, chimpanzee) and condition (Dispersed-divisible, Clumped-divisible, Clumped), ii) the two-way interaction between condition (Dispersed-divisible, Clumped-divisible, Clumped) and whether affiliative behaviours such as play or socio-sexual contact occurred during or before each trial (Yes, No), and iii) the three-way interaction between species (Bonobo, Chimpanzee) and the age of both partners (in years, z-transformed).

We included several additional factors to control for their influence on the tendency to co-feed. Thus, we included the maternal relatedness (Yes, No), the sex combination within a dyad (M-M, M-F, F-F), the trial number per condition (1–2, z-transformed), and the overall trial number across conditions (1–6, z-transformed). Finally, to keep type 1 error rates at the nominal level of 5%, we also included the random intercepts [53,54] or dyad identity, partner 1, and partner 2. The random slopes components [53,54] within dyad identity were condition and the trial number across conditions. The random slopes components within partner 1 and partner 2 were condition, the trial number across conditions, sex combination, the age of the other partner, and the interaction of condition and the age of the other partner. All continuous variables were z-transformed and all categorical variables were centred to their mean before including them as random slope components.

Given that the full-null comparison was significant, we removed non-significant interactions from the model to test the two-way interactions or main effects of the predictors described above. All else remained the same as described above.

### 2.6.3. Step 2b: new statistical analyses on new cooperation data

To understand what factors influenced whether the partners of the two species that we tested cooperated with one another, we fitted a second GLMM (see electronic supplementary material, 1.5. for full details) with a binomial error structure and logit link function [52]. The number of observations was 737 of 62 dyads (9 bonobo and 53 chimpanzee dyads). Our dependent variable was whether the two partners of a dyad cooperated during a given trial (Yes, No), hence, a 'No' was scored whenever the platform was not pulled within reach. As explanatory predictors, we included i) the three-way interaction between species (bonobo, chimpanzee), condition (Dispersed-divisible, Clumped-divisible), and whether affiliative behaviours such as play or socio-sexual contact occurred during or before each trial (Yes, No), and ii) the three-way interaction between species (bonobo, chimpanzee), condition (Dispersed-divisible, Clumped-divisible) and the number of trials this dyad co-fed in the previous experiment (z-transformed). We could not estimate the potentially hindering effect of aggressive interactions on cooperation [55] given that we only observed ten trials (out of 629 total trials) of aggressive interactions across eight chimpanzee dyads and zero cases across all bonobo dyads, meaning results would not be robust due to a lack of sufficient data points.

As above, we included several additional factors to control for their influence on the tendency to cooperate. We included the interaction between the age of partner 1 and partner 2 (both in years, z-transformed), the sex combination within a dyad (M-M, M-F, F-F), the trial number per condition (1–6, z-transformed), and the order in which the two conditions were tested (1–2, z-transformed). We did not include maternal relationship because none of the bonobo and only six of the chimpanzee dyads that we tested in the cooperation experiment were maternally related and a model that included the factor resulted in substantially inflated standard errors. Finally, to keep type 1 error rates at the nominal level of 5%, we also included the random intercepts [53,54] for dyad identity, partner 1, and partner 2. The random slopes components [53,54] within dyad identity were condition, the trial number and the order of conditions. The random slopes components within partner 1 and partner 2 were condition, trial number, order of conditions, sex combination, affiliative behaviours, number of trials co-fed, the age of the other partner, and the interaction of condition and the number of trials co-fed. All continuous

variables were z-transformed and all categorical variables were centred to their mean before including them as random slope components.

Given that the full-null comparison was significant, we removed non-significant interactions from the model to test the two-way interactions or main effects of the predictors described above. All else remained the same as described above.

### 2.6.4. Step 3: new statistical analyses on original co-feeding and cooperation data

Finally, we extracted the original data published in the electronic supplementary material of Hare2007 and ran GLMMs (see electronic supplementary material, 1.5. for full details) on the co-feeding and the two sets of cooperation data.

To analyse what factors influenced the number of trials each dyad did versus did not co-feed in experiment 1, we derived the dyad name, species (bonobo, chimpanzee), condition (Dispersed-divisible, Clumped-divisible, Clumped), sex combination within a dyad (M-M, M-F, F-F) and age of both partners (in years) from table 3 (electronic supplementary material, of Hare2007). Furthermore, we derived the number of trials during which the partners had socio-sexual contact or played from table 1 (electronic supplementary material, of Hare2007). By summing the number of trials during which the partners had socio-sexual contact or played, we created a single factor called affiliative behaviours. We then ran a GLMM with a binomial error structure and logit link function [52]. The number of observations was 78 of 26 dyads (10 bonobo and 16 chimpanzee dyads). Mirroring the GLMM used to analyse our data as much as possible, we included i) the two-way interaction between species (bonobo, chimpanzee) and condition (Dispersed-divisible, Clumped-divisible, Clumped), and ii) the three-way interaction between species (bonobo, chimpanzee) and the age of both partners (in years, z-transformed). Further, we included the number of trials during which affiliative behaviours occurred (z-transformed). We included the sex combination within a dyad as an additional factor to control for its influence on the tendency to co-feed, and the random intercept [53,54] for dyad identity without random slopes components [53,54]. Given that the full-null comparison was significant, we removed non-significant interactions from the model to test the two-way interactions or main effects of the predictors described above. All else remained the same as described above.

Further, we investigated what factors influenced the total number of trials that each dyad cooperated in 'Experiment 2' when the food was dispersed divisible. The same dyads were tested as in the co-feeding experiment, however, the condition 'clumped divisible' was not included by Hare2007, which is the rationale behind not replicating this but only 'Experiment 3' that does include both conditions. We derived all necessary information from table 1 (electronic supplementary material, of Hare2007) and ran a GLMM with a binomial error structure and logit link function [52]. The number of observations was 24 of 24 dyads (8 bonobo and 16 chimpanzee dyads). As an explanatory predictor, we included the two-way interaction between species (bonobo, chimpanzee) and the number of trials during which the dyad co-fed in 'Experiment 1' (z-transformed). Further, we included the two-way interaction between the age of both partners (in years, z-transformed), and the sex combination within a dyad (M-M, M-F, F-F) as additional factors to control for their influence on the tendency to cooperate. Finally, we included the random intercept [53,54] for dyad identity without random slopes components [53,54]. The full-null comparison was marginally significant ($\chi^2 = 6.9$, d.f. = 3, $N = 24$, $p = 0.075$). Keeping this in mind, we removed the non-significant interaction from the model to test the main effects of the predictors described above. All else remained the same as described above.

Finally, we investigated what factors influenced the number of trials dyads cooperated in 'Experiment 3' where the authors tested both the dispersed divisible and clumped divisible condition. Hare2007 tested new dyads in this experiment compared to those tested in 'Experiments 1 and 2'. We derived all necessary information from table 4 (electronic supplementary material, of Hare2007) and ran a GLMM with a binomial error structure and logit link function [52]. The number of observations was 24 dyads (12 bonobo and 12 chimpanzee dyads). As the explanatory predictor, we included the two-way interaction between species (bonobo, chimpanzee) and condition (Dispersed-divisible, Clumped-divisible). As above, we included several additional factors to control for their influence on the tendency to cooperate. Thus, we included the interaction between the age of partner 1 and partner 2 (both in years, z-transformed), the sex combination within a dyad (M-M, M-F, F-F), and the order in which the two conditions were tested (1–2, z-transformed). Further, we included the random intercepts [53,54] for group identity, dyad identity, partner 1 and partner 2, and the random slopes components of the age of partner 1 and partner 2 and the order tested within group identity [53,54].

# 3. Results

We present the results of original statistical analyses as conducted by Hare2007 with our newly obtained data (*Step 1*), the results of new statistical analyses (GLMMs) on our newly obtained co-feeding and cooperation data (*Step 2*), and finally the results of new statistical analyses (GLMMs) on the original data by Hare2007 (*Step 3*). All statistical comparisons can be found in tables 3 and 4, for the co-feeding and cooperation data, respectively.

## 3.1. Step 1: original statistical analyses on new co-feeding and cooperation data

We replicated all *t*-tests that were published in the original paper on the newly obtained co-feeding (see electronic supplementary material, table S5 for comparison with original data) and cooperation data (see electronic supplementary material, table S6 for comparison with original data).

### 3.1.1. Co-feeding tolerance

As was done previously, we used the count of the trials that each species co-fed across conditions as response variable. In the original study, bonobos co-fed significantly more often than chimpanzees, however, we only found a trend in this direction ($t_{10.82} = 1.56$, $N = 100$, $p = 0.074$). When only the two clumped conditions were assessed, in the original study the difference between the species was especially pronounced. We obtained a marginally significant result indicating that bonobos and chimpanzees differed in their co-feeding tolerance in the clumped conditions ($t_{10.45} = 1.81$, $N = 98$, $p = 0.049$).

With respect to the apes' social behaviours during the co-feeding tests, similar to Hare2007, we found no aggression within bonobo pairs. Yet, contrary to Hare2007 who found no species difference, we found significantly higher rates of aggression in chimpanzee than bonobo pairs ($t_{88} = -3.02$, $N = 100$, $p = 0.002$). Furthermore, Hare2007 observed significantly more socio-sexual contact between bonobo than chimpanzee pairs, however, in our study, even though socio-sexual interactions were absent in chimpanzees, we did not find a significant difference between the species ($t_{10} = 1.35$, $N = 100$, $p = 0.104$). Similarly, Hare2007 found significantly more play between bonobo than chimpanzee pairs, yet we did not observe this statistical pattern ($t_{11.37} = 0.68$, $N = 100$, $p = 0.256$). The original study found no age difference between the two species and concluded that the observed species differences in co-feeding tolerance and behaviour could not be explained by age differences (see [4]). In line with Hare2007's findings, the bonobos and chimpanzees in our study were of similar ages ($t_{21.38} = -1.46$, $p = 0.159$).

### 3.1.2. Cooperation

Hare2007 observed that bonobos and chimpanzees cooperated similarly often when the food was *dispersed divisible* in 'Experiment 2 and 3'. We found the same result ($t_{10.07} = 0.06$, $N = 62$, $p = 0.478$). When the food was *clumped divisible*, the original authors noted that bonobos cooperated in significantly more trials than chimpanzees. Again, we found the same result ($t_{8.87} = 2.51$, $N = 62$, $p = 0.017$). Further, Hare2007 observed that bonobos shared food more often than chimpanzees when the food was monopolizable (*clumped divisible*), and also took fewer pieces per trial compared to chimpanzees. We found that four out of 53 tested chimpanzee dyads (1 f-f, 1 m-m, 2 f-m, 7.5%) shared food after successful cooperation in the *clumped divisible* condition. By contrast, two of our nine tested bonobo dyads (both f-f, 22.2%) shared food at least once. We ran a Fisher's exact test that showed no statistical difference between the two species with regard to the number of dyads that shared food in condition *clumped divisible* ($p = 0.206$). The percentage of trials during which food was shared in these dyads ranged from 17% to 67% in chimpanzees, and from 67% to 83% in bonobos. We also coded aggression, play, and socio-sexual contact during the cooperation experiment and found some differences to the co-feeding experiment. While aggression rates were significantly higher in chimpanzees than bonobos ($t_{52} = -2.85$, $N = 62$, $p = 0.003$), bonobos had significantly more socio-sexual contact than chimpanzees in the cooperation task ($t_{8.07} = 2.75$, $N = 62$, $p = 0.012$). In the cooperation experiment, chimpanzee dyads played significantly more often than bonobo dyads, which in fact did not play before or during any of the trials ($t_{52} = -3.07$, $N = 62$, $p = 0.002$).

**Table 3.** Comparison of results based on co-feeding data from the original Hare2007 study and the current replication. Both types of analyses are presented (i.e. the original *t*-tests and the currently used GLMMs). Those analyses for which we did not find congruent results are indicated in bold.

| topic | statistical test | Hare *et al.* [4] | current replication |
|---|---|---|---|
| co-feeding across conditions | *t*-test[a] | **bonobos > chimpanzees** **$t_{24} = 3.38$, $p = 0.002$** | trend, bonobos = chimpanzees $t_{10.82} = 1.56$, $p = 0.074$ |
| | GLMM | **bonobos > chimpanzees** **$\chi^2(1) = 5.78$, $p = 0.016$** | bonobos = chimpanzees $\chi^2(1) = 0.39$, $p = 0.533$ |
| co-feeding in clumped conditions | *t*-test[a] | bonobos > chimpanzees $t_{24} = 3.52$, $p < 0.001$ | bonobos > chimpanzees $t_{10.45} = 1.81$, $p = 0.049$ |
| | GLMM | non-sign. interaction, both co-fed similarly different between all three conditions *interaction:* $\chi^2(2) = 2.69$ $p = 0.260$ *main effect of condition:* $\chi^2(2) = 31.32$, $p = < 0.001$ | non-sign. interaction, both co-fed more in the dispersed than the two clumped conditions *interaction:* $\chi^2(2) = 0.27$, $p = 0.874$ *main effect of condition:* $\chi^2(2) = 30.89$, $p < 0.001$ |
| age of the partner | *t*-test[a] | bonobos = chimpanzees non-sign., $p > 0.200$ | bonobos = chimpanzees $t_{21.38} = -1.46$, $p = 0.159$ |
| | GLMM | influences co-feeding for both species P1: $\chi^2(1) = 3.76$, $p = 0.052$ **P2: $\chi^2(1) = 7.43$, $p = 0.006$** | influences co-feeding for both species **P1: $\chi^2(1) = 3.86$, $p = 0.049$** P2: $\chi^2(1) = 3.19$, $p = 0.074$ |
| affiliative behaviours | *t*-test[a] | **bonobos > chimpanzees** **soc.-sex: $t_9 = 2.51$, $p = 0.017$** **play: $t_{9.14} = 2.33$, $p = 0.022$** | bonobos = chimpanzees soc.-sex: $t_{10} = 1.35$, $p = 0.104$ play: $t_{11.37} = 0.68$, $p = 0.256$ |
| | GLMM | does not influence co-feeding for both species $\chi^2(1) = 0.001$, $p = 0.973$ | does not influence co-feeding for both species $\chi^2(1) = 0.68$, $p = 0.409$ |
| aggression | *t*-test[a] | bonobos = chimpanzees non-sign., Fig. 1 in Hare *et al.* [4] | **bonobos < chimpanzees** **$t_{88} = -3.02$, $p = 0.002$** |

[a]applied to replicate original statistical approach by Hare2007.

## 3.2. Step 2: new statistical analyses on new co-feeding and cooperation data

Next, we modelled the newly obtained co-feeding data (see electronic supplementary material, table S8 for comparison with our *t*-tests) and cooperation data (see electronic supplementary material, table S10 for comparison with our *t*-tests) with Generalized Linear Mixed Models (GLMM) to understand which factors influenced the likelihood to co-feed and cooperate.

### 3.2.1. Co-feeding tolerance

We assessed which factors influenced the likelihood that dyads co-fed, and the resulting full GLMM fitted the co-feeding data significantly better than a null model including only the control factors and random effects ($\chi^2 = 50.38$, d.f. = 14, $N = 587$, $p < 0.001$). Since none of the interactions were significant (see *Methods*), we omitted them to evaluate the main effects (see electronic supplementary material, table S7). Most importantly, we found no evidence for an interaction between species

**Table 4.** Comparison of results based on cooperation data from the original Hare2007 study and the current replication. Both types of analyses are presented (i.e. the original $t$-tests and the currently used GLMMs). Those analyses for which we did not find congruent results are indicated in bold.

| topic | statistical test | Hare et al. [4] | current replication |
|---|---|---|---|
| cooperation across conditions | $t$-test[a] | Exp. 2, dispersed div.:<br>bonobos = chimpanzees<br>$t_{10} = 0.660$, $p > 0.5$ | dispersed div.:<br>bonobos = chimpanzees<br>$t_{10.07} = 0.060$, $p = 0.478$ |
| | | Exp. 3, clumped div.:<br>bonobos > chimpanzees<br>R1: $t_{10} = 2.8$, $p < 0.01$[b]<br>R2: $t_{10} = 1.9$, $p < 0.05$[b] | clumped div.:<br>bonobos > chimpanzees<br>$t_{8.87} = 2.51$, $p = 0.017$ |
| | GLMM | Exp. 2:<br>marginally sign. model,<br>reduced model shows:<br>**chimpanzees > bonobos[c]**<br>**$\chi^2(1) = 6.79$, $p = 0.009$** | |
| | | Exp. 3:<br>sign. interaction<br>bonobos > chimpanzees in clumped div. but not dispersed div.<br>$\chi^2(1) = 9.59$, $p = 0.002$ | sign. interaction<br>bonobos > chimpanzees in clumped div. but not dispersed div.<br>$\chi^2(1) = 5.08$, $p = 0.024$ |
| co-feeding in previous experiment | GLMM | Exp. 2:<br>marginally sign. model,<br>reduced model shows:<br>influences cooperation across conditions[c]<br>$\chi^2(1) = 4.91$, $p = 0.027$ | influences cooperation across conditions<br>$\chi^2(1) = 5.67$, $p = 0.017$ |
| affiliative behaviours | $t$-test[a] | not reported | soc.-sex:<br>bonobos > chimpanzees<br>$t_{8.07} = 2.75$, $p = 0.012$ |
| | | | play:<br>bonobos < chimpanzees<br>$t_{52} = -3.07$, $p = 0.002$ |
| | GLMM | not included | does not influence cooperation for both species<br>$\chi^2(1) = 0.54$, $p = 0.462$ |
| aggression | $t$-test[a] | not reported | bonobos < chimpanzees<br>$t_{52} = -2.85$, $p = 0.003$ |

[a]applied to replicate original statistical approach by Hare2007.

[b]'Experiment 3' was split into two rounds, one with a same sex partner and one with an opposite sex partner.

[c]the full model was only marginally significantly different to a null model that only included the control factors and random intercept ($p = 0.075$), wherefore the results presented here need to be interpreted with caution.

and condition ($\chi^2 = 0.27$, d.f. = 2, $p = 0.874$). Furthermore, we found no evidence that either species was generally more likely to co-feed ($\chi^2 = 0.39$, d.f. = 1, $p = 0.533$; figure 3). However, the type of condition significantly influenced the likelihood of whether they co-fed ($\chi^2 = 30.89$, d.f. = 2, $p < 0.001$; figure 3).

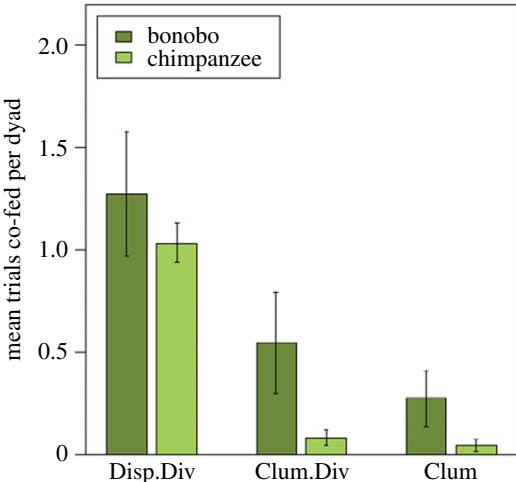

**Figure 3.** New co-feeding data with new statistical analyses. Mean number (±standard error) of trials that bonobo and chimpanzee dyads co-fed in the dispersed divisible, clumped divisible, and clumped condition. Each dyad received two trials per condition.

Both species were more likely to co-feed when the food was *dispersed divisible* compared to when it was *clumped divisible* or *clumped non-divisible*, but similarly likely in the *clumped divisible* and *clumped non-divisible* conditions.

With respect to the expressed social behaviours during the experiment, we did not find an effect of whether the partners engaged in affiliative behaviour before or during the trial on the likelihood to co-feed ($\chi^2 = 0.68$, d.f. = 1, $p = 0.409$). Interestingly, the age of the partner influenced the likelihood of whether apes co-fed. We used the age of both partners separately, but since our outcome is not directional the assignment of partner 1 and partner 2 is arbitrary. The effect for partner 1 was slightly below the 0.5 significance level ($\chi^2 = 3.86$, d.f. = 1, $p = 0.049$), while for partner 2 it was slightly above it ($\chi^2 = 3.19$, d.f. = 1, $p = 0.074$). For both the likelihood that the ape would co-feed increased with decreasing age of the partner.

### 3.2.2. Cooperation data

We assessed which factors influenced the likelihood that dyads cooperated, and the resulting full GLMM fitted the co-feeding data significantly better than a null model including only the control factors and random effects ($\chi^2 = 47.66$, d.f. = 11, $N = 737$, $p < 0.001$). Again, we excluded non-significant interactions (see *Methods*) to assess the lower interaction terms or main effects, resulting in a model that encompassed the two-way interaction between condition and species and the main effects for affiliative behaviours and number co-fed (see electronic supplementary material, table S9). Bonobos were more likely to cooperate than chimpanzees, but this difference was dependent on the condition ($\chi^2 = 5.08$, d.f. = 1, $p = 0.024$): while both species were similarly likely to cooperate in the *dispersed divisible* condition, chimpanzees but not bonobos were less likely to cooperate in the *clumped divisible* condition compared to the *dispersed divisible* condition (figure 4). Across both species, dyads with a higher number of trials of co-feeding during the previous experiment were more likely to cooperate ($\chi^2 = 5.67$, d.f. = 1, $p = 0.017$; figure 5). We did not find evidence for a significant influence of whether the partners had affiliative contact before or during a trial on the likelihood to cooperate ($\chi^2 = 0.54$, d.f. = 1, $p = 0.462$).

## 3.3. Step 3: new statistical analyses on original co-feeding and cooperation data

Lastly, we ran GLMMs on the original data of 'Experiments 1–3' provided in the electronic supplementary material of Hare2007 and mirrored the models that we used on the new data as much as possible.

### 3.3.1. Co-feeding data

First, we analysed the original co-feeding data ('Experiment 1' of Hare2007). The full GLMM (see electronic supplementary material, table S12 for comparison with original results) fitted the data significantly better than the corresponding null model only including the control factor and random

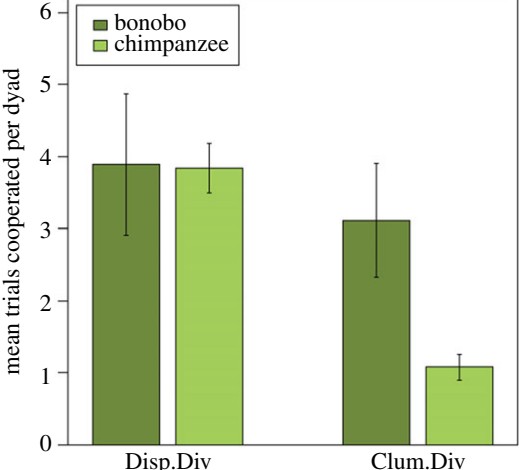

**Figure 4.** New cooperation data with new statistical analyses. Mean number (±standard errors) of trials that bonobo and chimpanzee dyads cooperated in the dispersed divisible and clumped divisible condition. Each dyad received six trials per condition.

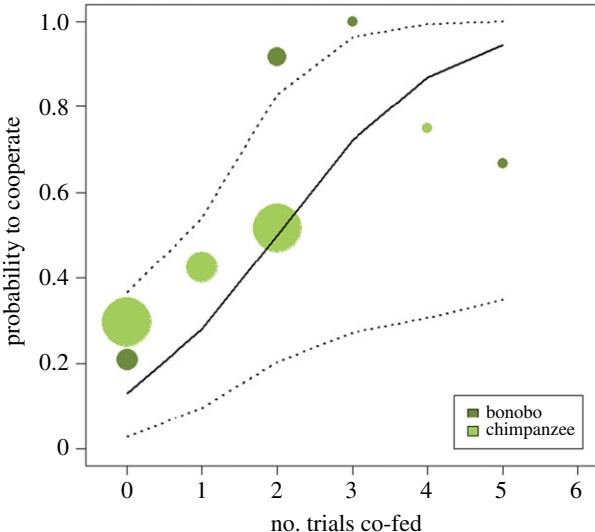

**Figure 5.** New cooperation data with new statistical analyses. Proportion of cooperation as a function of number of trials co-fed in the previous experiment, separately for chimpanzees and bonobos. The area of the dots scales with the number of observations (ranging from 12 to 261; total $N = 737$). The line shows the fitted model and the dotted lines its 95% confidence interval.

intercept ($\chi^2 = 49.32$, d.f. = 12, $N = 78$, $p < 0.001$). Since none of the interactions were significant (see *Methods*), we omitted them to evaluate the main effects (see electronic supplementary material, table S11). Contrasting our data (however, note the difference in response variables used), the main effect of species was significant ($\chi^2 = 5.78$, d.f. = 1, $p = 0.016$) with bonobos co-feeding more often than chimpanzees. Further, while exhibiting the same pattern, both species co-fed to a different degree across conditions ($\chi^2 = 31.32$, d.f. = 2, $p < 0.001$). Similar to our data, they co-fed more often when the food was *dispersed divisible* compared to when it was *clumped divisible* or *clumped non-divisible*. However, contrasting our results and the original *t*-tests, both species co-fed more when the food was *clumped divisible* versus *clumped non-divisible*. In the original data, we also did not find an effect of the number of affiliative behaviours that the two partners engaged in during the test ($\chi^2 = 0.001$, d.f. = 1, $p = 0.973$). Additionally, in the original data the age of the partner also influenced whether an ape co-fed: There was a trend for one of the arbitrarily set partners and a significant effect for the other (partner1: $\chi^2 = 3.76$, d.f. = 1, $p = 0.052$; partner2: $\chi^2 = 7.43$, d.f. = 1, $p = 0.006$). Hence, apes would co-feed more with decreasing partner's age.

### 3.3.2. Cooperation data

In 'Experiment 2', Hare2007 only tested whether apes would cooperate in the *dispersed divisible* condition. The full GLMM (see electronic supplementary material, table S15 for comparison with original results) was only marginally significantly different to a null model that only included the control factors and random intercept ($\chi^2 = 6.9$, d.f. = 3, $N = 24$, $p = 0.075$; see electronic supplementary material, table S13). However, while keeping in mind the marginal significance, we further assessed the main effects (see electronic supplementary material, table S14). A higher number of trials a dyad had co-fed in 'Experiment 1' was significantly related to more cooperation in 'Experiment 2' across species ($\chi^2 = 4.91$, d.f. = 1, $p = 0.027$). Moreover, chimpanzees cooperated more than bonobos in 'Experiment 2' ($\chi^2 = 6.79$, d.f. = 1, $p = 0.009$). These two results need to be interpreted with caution given the marginal significance of the model.

In 'Experiment 3', Hare2007 included a new set of dyads and tested whether apes would cooperate differently in the *dispersed divisible* compared to the *clumped divisible* condition. The full GLMM (see electronic supplementary material, table S17 for comparison with original results) was significantly different to a null model that only included the control factors and random intercepts ($\chi^2 = 17.28$, d.f. = 3, $N = 48$, $p < 0.001$; see electronic supplementary material, table S16). Corroborating our results and the *t*-tests, bonobos cooperated significantly more than chimpanzees when the food was *clumped divisible* compared to when it was *dispersed divisible* ($\chi^2 = 9.59$, d.f.= 1, $p = 0.002$).

## 4. Discussion

This study replicated a seminal study by Hare and colleagues ([4], henceforth: Hare2007) that compares performance of two ape species, bonobos and chimpanzees, in a co-feeding and a cooperation task. Hare2007 found bonobos to co-feed more often compared to chimpanzees and also to cooperate more often when the food was monopolizable, resulting in the conclusion that bonobos' heightened tolerance levels enabled them to cooperate even under potentially conflict-inducing conditions. By contrast, other studies found bonobos to be less tolerant or cooperative than chimpanzees [42–44,46]. By replicating this cornerstone experimental study, we aimed to understand whether current discrepant findings on Pan tolerance and cooperation may result from noise, methodological differences, or behavioural variability. Taken together, we found that bonobos and chimpanzees co-fed to a similar degree in our sample, contrasting the previous finding by Hare2007. However, in line with the original study, bonobos cooperated more than chimpanzees when the food was monopolizable, yet showed similar levels of cooperation when the food was sharable.

### 4.1. Co-feeding replicate

Our replication of the Hare2007 co-feeding task did not result in a replication of all outcomes. First, we performed the same statistical analyses as done by Hare2007 on the new data. Based on such analyses, even though we found a significant difference between the two species in the number of trials they co-fed on clumped food, we did not find that bonobos co-fed more often than chimpanzees in general (i.e. across all food distribution conditions). Furthermore, while chimpanzees also had more aggressive contact than bonobos in the current study, we did not replicate a difference in their play or socio-sexual contact during the co-feeding task.

Second, when analysing our co-feeding data with a statistical model that takes into account potentially influential variables (e.g. maternal relatedness, sex combination), we did not find a pronounced species difference in either condition. Both species co-fed most in the dispersed compared to the two clumped conditions. By contrast to Hare2007, we, therefore, did not find support that the two species co-fed to a different degree. The model showed no significant influence of socio-sexual contact and play on the likelihood to co-feed. Interestingly, our data indicated that the apes were more likely to co-feed with younger partners.

Lastly, we ran a model on the original co-feeding data of Hare2007 to explore whether currently used statistical models provide different outcomes. Corroborating the *t*-tests, the model showed that chimpanzees tested by Hare2007 co-fed less often than bonobos even when the effect of other factors (e.g. age, sex combination) was statistically considered. Contrasting the *t*-tests, the model showed no pronounced species difference in either condition and both species co-fed most when the food was dispersed divisible, followed by when it was clumped divisible, and when it was clumped. There was no effect of the number of affiliative behaviours on the frequency to co-feed, mirroring the outcome

obtained from our data. Interestingly, also apes of the original data co-fed more with younger partners. While we lack information on the maternal relationship between the dyads in the original data and therefore could not control for its potential influence in the model, we did control for its influence in our replication sample. Thus, bonobos and chimpanzees co-fed more when the partner was younger independent of relatedness, which is possibly due to higher perceived control over the outcome (i.e. resource-holding potential) or higher tolerance levels toward younger group members [56].

## 4.2. Cooperation replicate

Our outcomes on the cooperation experiment indicate that the replication corroborates the published Hare2007 data. First, we replicated the analyses done by Hare2007 and found mostly congruent results—while both species cooperated to a similar degree when the food was sharable, bonobos cooperated more often than chimpanzees when it was clumped and thus monopolizable. Further, chimpanzees had more aggressive contact than bonobos and, at least during the cooperative task, bonobos had more socio-sexual contact than chimpanzees. The only striking difference is that chimpanzees played more often than bonobos during our cooperation task, with the opposite result found by Hare2007 during the co-feeding task. This could point toward a different coping strategy to reduce tension or to avoid possible conflicts.

Similarly, the new statistical model used to analyse our cooperation data also corroborates the original Hare2007 results. We found no difference between the two species in the condition *dispersed divisible*, but bonobos cooperated more in the condition *clumped divisible* than chimpanzees. For both species, the likelihood to cooperate with the partner was influenced by the number of trials they co-fed in the previous experiment. Similar to the co-feeding task, socio-sexual contact and play did not influence the likelihood to cooperate. Thus, in both tasks, socio-sexual contact and play were most likely reactive (i.e. coping strategies) instead of proactive responses induced by the test.

Lastly, we ran two new statistical models on the original cooperation data of Hare2007. Re-analysing 'Experiment 2' (only condition *dispersed divisible* was tested), we found that chimpanzees cooperated more than bonobos, contrasting the original results obtained via *t*-tests (i.e. no species difference was found in Hare2007) and our own replication. This finding needs to be interpreted cautiously though, as the full-null model comparison only trended toward significance (see §3.3.2.). The original data also showed that co-feeding positively influenced how often dyads cooperated in 'Experiment 2'. However, again this finding needs to be interpreted cautiously (see §3.3.2.). We ran an additional model on the original cooperation data from 'Experiment 3', which resulted in the same outcome as the original *t*-tests and the results from our data: bonobos cooperated more often than chimpanzees when the food was *dispersed divisible* compared to *clumped divisible*. Thus, the new statistical analyses largely corroborate the published outcomes concerning cooperation.

## 4.3. General discussion

Here, we relate the outcomes of the co-feeding and cooperation experiments and point to future directions. Since we did not find strong support that bonobos are more likely to co-feed on food resources than chimpanzees, our findings call into question Hare2007's interpretation that at the species level bonobos cooperate to a higher degree because they are inherently more tolerant toward each other. Nevertheless, tolerance did play a role in whether dyads cooperated and those pairs that shared resources with each other in the co-feeding experiment were also more likely to cooperate (in line with [57]), but this effect was the same for both species. Our findings, therefore, indicate that differences in tolerance at group levels result from dyad characteristics, instead of a species-specific pattern. This study design facilitates the influence of tolerance on cooperation as the apes were tested without physical barriers that spatially separated them. Past literature already highlighted the influence of dyad characteristics on tolerance when partners were not spatially separated by physical barriers during cooperative tasks ([31,58,59]; also see same effect in other species: [60–62]).

At least in chimpanzees, tolerance cannot be regarded as a stable species-specific trait, but exhibits substantial within- and inter-group variation [45,59,63]. Whereas the same is likely true for bonobos [64], most *Pan* research focused on the effect of tolerance on cooperation only in chimpanzees (e.g. [31,58,59]) or studies compared several groups of chimpanzees to one group of bonobos [45,46]. Unfortunately, the current study is facing a similar limitation: due to safety reasons induced by the COVID-19 pandemic, we could not obtain data from multiple groups and therefore cannot gauge the extent of behavioural variability that influenced our outcome. However, the two sampled groups were

housed at the same facility, therefore ruling out differences in housing conditions and rearing background. More data are needed from different chimpanzees and bonobo groups to understand the extent of variation of tolerance levels in bonobo groups, derive potential species-specific traits, and draw general conclusions about the link between tolerance and cooperation in both species.

Tolerance differences between bonobos and chimpanzees may also result from the dyads selected for the experiments. Due to animal handling decisions, we could not test adult female–male bonobo dyads, but only two mothers with their offspring. These kinds of constraints are common when testing bonobos across zoos and, usually, testing is only allowed when high tolerance within a pair is ensured. Hence, bonobo pairings are often already pre-selected due to handling regulations, which suggests that studies testing dyads for tolerance levels might be biased. The same often does not hold true for chimpanzee testing, where regulations typically allow for a multitude of different pairings as long as aggression remains low. Especially when studying tolerance, biases in pre-selection of tested dyads should be reported whenever possible and it should be discussed whether such pre-selection might have influenced outcomes. Our data are influenced by a pre-selection and subsequent lack of adult mixed bonobo pairs as discussed above, meaning that tolerance in our sampled bonobo group might be overestimated compared to that of the sampled chimpanzees.

A further limitation of the current study is that we could not replicate the amount of food that was used in the original study as feeding regulations of the zoo forbade the use of such large amounts. Instead, we replicated what was originally done by Hare2007 for condition *clumped divisible* of the cooperation task, and for the remaining conditions and the co-feeding task applied the same logic of dividing the food as in the original design (e.g. food that was used in *clumped dispersed* was divided in half and placed at the outer sides). This resulted in consistent food amounts within and across experiments while still being based on one condition originally used by Hare2007. The discrepancy in co-feeding results between our and the original study might have resulted from the difference in food quantities used. One possibility here could be that bonobos but not chimpanzees share more food when the amount is relatively large. In Hare2007, the apes received 500 g of fruits in the condition *clumped divisible* instead of four 1.5 cm thick banana pieces in our study. In case species differences in tolerance explain the difference in behaviour, this suggests that tolerance around larger food amounts may break down in chimpanzees but not bonobos (however see potentially opposite effect in chimpanzees based on [57] Exp.1 compared to Exp.2). Future studies that compare tolerance levels of the two species could experimentally vary the amount of food they receive and inform when and whether tolerance breaks down depending on the amount of food presented to both species.

Even though both our sampled groups performed similarly on dyadic tolerance measures, we did find that bonobos were better able to maintain cooperation when the food was monopolizable, while performing similarly when the food was sharable. This study, therefore, adds to the cumulating body of literature that, in direct comparison, bonobos seem more adept at solving cooperative tasks than chimpanzees under potentially conflict-inducing situations [4,36,37]. While keeping in mind that samples of multiple groups are needed to rule out the effect of intra-species variation, greater cooperative ability in bonobos may be due to a greater ability or motivation to understand or attend to social cues. Research underlying this claim comes from studies assessing neural, behavioural, and hormonal differences between the two species. While bonobo males seem to be less conflict-oriented than chimpanzee males [65], differences in oxytocin reactivity could support general affiliative contact and an orientation toward the face and eyes of the partner in bonobos [66]. Eye contact and gaze following are crucial features to the development of human communication and understanding the intent of your partner [67,68], and have been found to be indeed increased in bonobos compared to chimpanzees [69–71]. Thus, increased eye contact in turn may increase the likelihood that bonobos understand social cues or intentions of their partner. Based on neural differences between the two species and relevant research in humans, bonobos may be better at socio-emotional processing than chimpanzees [72–74], and were found to perform better than chimpanzees on tasks involving theory of mind abilities [69,75], and be sensitive to violations of social expectations during aggressive conflicts [76]. Taken together, the picture emerges that bonobos may direct their attention more toward social cues of the partner, therefore possibly facilitating a greater understanding of the partner's communication or goals. This ability may enable bonobos to outperform chimpanzees during tasks that are potentially conflict inducing and require increased social coordination, hence, possibly explain why chimpanzees were able to cooperate similarly to bonobos when the risk of monopolization did not exist. However, since studies are missing that directly show bonobos to be better at perceiving partner's communication and intentions than chimpanzees, future research should tackle the connection between current lines of research. If future studies would indeed reveal that bonobos are better at perceiving partner's communication and intentions, our study would

support the notion that such heightened sensitivity to perceive social cues in turn translates to an increased ability to cooperate with one another.

Furthermore, we need to address why some studies show the reverse pattern arguing that chimpanzees are more tolerant and prosocial than bonobos [42–44,46]. As was proposed above, bonobos may show greater prosocial motivation in some contexts (i.e. cooperating to achieve a common goal) and chimpanzees in others (i.e. sharing resources). Closely replicating additional cornerstone studies on tolerance and cooperation in our closest living relatives are warranted to further our understanding of the contexts during which one or the other species might be better able to show their cooperative skills and to estimate the magnitude of group-level variation that both species might show.

Altogether, our study shows the value of replicating earlier studies. Our outcomes confirm that bonobos are more cooperative than chimpanzees specifically in a clumped setting, corroborating the earlier found species difference in this capacity. However, our new data show a lack of difference in tolerance between these two species. Instead of species differences, our results indicate that dyadic differences—possibly affected by group-specific social dynamics—may be crucial to understand how tolerance affects cooperation in the Pan species. This informs between and within species variation, and such knowledge will greatly aid in the search of proximate factors underlying tolerance and cooperation and those that have shaped it over evolutionary time.

## Preregistered report

This article received results-blind in-principle acceptance (IPA) at Royal Society Open Science. Following IPA, the accepted Stage 1 version of the manuscript, not including results and discussion, was preregistered on the OSF (https://osf.io/ejywz/). This preregistration was performed after data analysis.

Ethics. The Max Planck Institute for Evolutionary Anthropology, Leipzig, Germany approved this study. Research was non-invasive and strictly adhered to the German legal requirements. The study and the housing conditions of the apes complied with the ethical guidelines of the European and World Association of Zoos and Aquariums (EAZA and WAZA).

Data accessibility. Data are available from the Dryad Digital Repository: https://doi.org/10.5061/dryad.nzs7h44vj [77].
    Supplementary material is available online [78].

Authors' contributions. S.N.: conceptualization, data curation, formal analysis, investigation, methodology, project administration, resources, software, supervision, visualization, writing—original draft, writing—review and editing; E.H.S.: conceptualization, funding acquisition, methodology, project administration, resources, supervision, writing—original draft, writing—review and editing; E.J.C.L.: conceptualization, funding acquisition, methodology, project administration, resources, supervision, writing—original draft, writing—review and editing.

    All authors gave final approval for publication and agreed to be held accountable for the work performed therein.

Conflict of interest declaration. We declare we have no competing interests.

Funding. Open access funding provided by the Max Planck Society.

    This publication is part of the project 'The evolutionary roots of human social tolerance and cooperation' (with project number 401.18.04) of the research programme Replication Studies which is financed by the Dutch Research Council (NWO). E.J.C. van Leeuwen was funded by the European Union under ERC Starting grant no. 101042961 – CULT_ORIGINS. Views and opinions expressed are however those of the author(s) only and do not necessarily reflect those of the European Union or the European Research Council Executive Agency. Neither the European Union nor the granting authority can be held responsible for them.

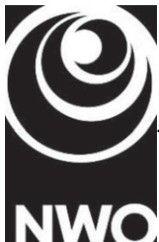

Acknowledgements. We are grateful to Daniel Haun and Hanna Petschauer for hosting and coordinating our research at the WKPRC, Paul Babucke for providing reliability coding, Sylvio Tüpke for the figures of the set-up, and the caretakers at WKPRC for enabling testing. Moreover, we thank Brian Hare for providing details on the original test protocol to facilitate our replication. Above all, we thank the chimpanzees and bonobos for their participation.

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
