## [Peer Review File · Royal Society Open Science]

Review History

RSOS-211106.R0 (Original submission)

Review form: Reviewer 1

Do you have any ethical concerns with this paper?

No

Have you any concerns about statistical analyses in this paper?

Yes

Recommendation?

Major revision

Comments to the Author(s)

See attached review (Appendix A).

Review form: Reviewer 2

Do you have any ethical concerns with this paper?

Yes

Have you any concerns about statistical analyses in this paper?

No

Recommendation?

Accept with minor revision

Comments to the Author(s)

The study makes valuable contribution to the literature on social tolerance and cooperation. The article is well written, clear, and concise. The authors provide a thorough review of relevant literature in *Pan spp.* and the existing discrepancies in results.

The authors provide a detailed description of their methods and analyses. In addition to directly replicating the original paper's analyses, the authors also updated their analyses to include GLMMs. Importantly, the authors extracted the data from the original study and reanalysed the data using GLMMs. This approach will provide valuable insight into analytical approaches typically used in these types of studies.

In principle, the findings of this study will help to resolve debate over social tolerance and cooperation in two closely related species which exhibit different social behaviours; study results will make a valuable contribution to literature investigating questions surrounding the evolutionary links between social tolerance and cooperative abilities.

I have selected 'accept with minor revision' due to one area of the manuscript that was unclear:

Line 104 states "due to special animal handling regulations for the bonobos, no adult mixed pairs could be tested, but only mothers with their male offspring."

However, it is unclear whether bonobo dyads were tested in only mother/ male offspring pairs for both co-feeding and cooperation trials. It is also unclear whether subadult bonobos were tested in unrelated pairs.

In their analyses of co-feeding, the authors included maternal relatedness as an explanatory variable but this variable was not included for the analyses of cooperation data. Line 237 states: "We did not include maternal relationship because none of the bonobo and only six of the chimpanzee dyads that we tested in the cooperation experiment were maternally related and a model that included the factor resulted in substantially inflated standard errors." This line indicates that bonobos were tested in non-related pairs for the cooperation trials.

Thus, I suggest the authors make a minor revision to clarify dyad composition for both trial types (co-feeding and cooperation). This can be easily accomplished by including a few lines in the methods section or a supplementary table that lists the dyads and the relationship between dyad members.

Decision letter (RSOS-211106.R0)

Dear Ms Nolte,

The Editors assigned to your Stage 1 Replication submission ("Does Tolerance Allow Bonobos to Outperform Chimpanzees on a Cooperative Task? A replication of Hare et al., 2007") have now received comments from reviewers. We would like you to revise your paper in accordance with the referee and editors suggestions which can be found below (not including confidential reports to the Editor). Please note this decision does not guarantee eventual acceptance.

Please submit a copy of your revised paper within three weeks (i.e. by the 06-Oct-2021). If deemed necessary by the Editors, your manuscript will be sent back to one or more of the original reviewers for assessment.

When submitting your revised manuscript, you must respond to the comments made by the referees and upload a file "Response to Referees" in the "File Upload" step. Please use this to document how you have responded to the comments, and the adjustments you have made. In order to expedite the processing of the revised manuscript, please be as specific as possible in your response.

Once again, thank you for submitting your manuscript to Royal Society Open Science and I look forward to receiving your revision. If you have any questions at all, please do not hesitate to get in touch. Full author guidelines may be found at <https://royalsocietypublishing.org/rsos/replication-studies#AuthorsGuidance>.

Kind regards,
Professor Chris Chambers
Royal Society Open Science
openscience@royalsociety.org

Associate Editor Comments to Author (Professor Chris Chambers):

Associate Editor: 1

Comments to the Author:

Two expert reviewers have now assessed the Stage 1 manuscript. As you will see, the reviews differ substantially in their assessments. Review 1 is critical of several aspects of the manuscript, but most notably whether the study deviates too much from the original design, and whether it lacks a sufficient sample size, to satisfy Stage 1 criterion #2. The tone of the review is blunt in places (e.g. I personally do not regard your discussion of reproducibility as "virtue signalling") but it is also constructive and helpful where it counts, providing a number of useful suggestions for addressing the concerns raised. Review 2 is more positive, recommending only minor revisions to include additional methodological details.

The main concern identified in Review 1 is whether the study deviates too greatly from the original study design to be considered under the replication article type. This is often a concern raised in Stage 1 replications, given the lack of close replications in many fields, and it can be a challenging, subjective judgment. I am therefore interested especially in how the authors respond to this point. In the event that this criterion remains unmet, but all other concerns are solved, it is

possible that we could eventually transfer the article from the Replications article type to the Research Article type. However, we can make that decision later after Stage 2, provided we get that far. For now, please revise and submit under the Replications article type.

Comments to Author:

Reviewer: 1

Comments to the Author(s)

See attached review

Reviewer: 2

Comments to the Author(s)

The study makes valuable contribution to the literature on social tolerance and cooperation. The article is well written, clear, and concise. The authors provide a thorough review of relevant literature in *Pan spp.* and the existing discrepancies in results.

The authors provide a detailed description of their methods and analyses. In addition to directly replicating the original paper's analyses, the authors also updated their analyses to include GLMMs. Importantly, the authors extracted the data from the original study and reanalysed the data using GLMMs. This approach will provide valuable insight into analytical approaches typically used in these types of studies.

In principle, the findings of this study will help to resolve debate over social tolerance and cooperation in two closely related species which exhibit different social behaviours; study results will make a valuable contribution to literature investigating questions surrounding the evolutionary links between social tolerance and cooperative abilities.

I have selected 'accept with minor revision' due to one area of the manuscript that was unclear:

Line 104 states "due to special animal handling regulations for the bonobos, no adult mixed pairs could be tested, but only mothers with their male offspring."

However, it is unclear whether bonobo dyads were tested in only mother/male offspring pairs for both co-feeding and cooperation trials. It is also unclear whether subadult bonobos were tested in unrelated pairs.

In their analyses of co-feeding, the authors included maternal relatedness as an explanatory variable but this variable was not included for the analyses of cooperation data. Line 237 states: "We did not include maternal relationship because none of the bonobo and only six of the chimpanzee dyads that we tested in the cooperation experiment were maternally related and a model that included the factor resulted in substantially inflated standard errors." This line indicates that bonobos were tested in non-related pairs for the cooperation trials.

Thus, I suggest the authors make a minor revision to clarify dyad composition for both trial types (co-feeding and cooperation). This can be easily accomplished by including a few lines in the methods section or a supplementary table that lists the dyads and the relationship between dyad members.

Author's Response to Decision Letter for (RSOS-211106.R0)

See Appendices B & C.

RSOS-211106.R1 (Revision)

Review form: Reviewer 1

Do you have any ethical concerns with this paper?

No

Have you any concerns about statistical analyses in this paper?

No

Recommendation?

Reject

Comments to the Author(s)

See attached file (Appendix D).

Review form: Reviewer 2

Do you have any ethical concerns with this paper?

No

Have you any concerns about statistical analyses in this paper?

I do not feel qualified to assess the statistics

Recommendation?

Accept with minor revision

Comments to the Author(s)

As stated in my previous review, in principle, the findings of this study will help resolve debate over social tolerance and cooperation in two closely related species that exhibit different social behaviors. The authors provide a detailed description of their methods and analyses and the study will make a valuable contribution to the literature. I recommend 'accept with minor revision' with the following suggestions for improvement:

1. Provide information (e.g. a supplementary table) on the specific behaviors coded as prosocial, playful, or aggressive in section 4.2.
2. As described in Hare et al. 2007 (see also Hare, Brian, and Michael Tomasello. "The emotional reactivity hypothesis and cognitive evolution." 2005, which I'm surprised is not included in the literature cited) success on cooperative problem solving tasks can be hindered by aggression. In Hare et al. 2007, aggression was found to be non-significant/non-existent. The current study does not include aggressive behavior in the new analyses of co-feeding or cooperation, and it is unclear from the provided information whether this is due to an absence of aggressive behavior during trials or whether the authors had other justifications for only including affiliative behavior in the new models. If, as in Hare 2007, aggressive behaviors were found to be NS or did not occur, this should be stated. Otherwise, a justification should be provided.

Decision letter (RSOS-211106.R1)

Dear Ms Nolte,

The Editors assigned to your Stage 1 Replication submission ("Does Tolerance Allow Bonobos to Outperform Chimpanzees on a Cooperative Task? A replication of Hare et al., 2007") have now received comments from reviewers. We would like you to revise your paper in accordance with the referee and editors suggestions which can be found below (not including confidential reports to the Editor). Please note this decision does not guarantee eventual acceptance.

Please submit a copy of your revised paper within three weeks (i.e. by the 01-Feb-2022). If deemed necessary by the Editors, your manuscript will be sent back to one or more of the original reviewers for assessment.

Once again, thank you for submitting your manuscript to Royal Society Open Science and I look forward to receiving your revision. If you have any questions at all, please do not hesitate to get in touch. Full guidance for authors is also available at <https://royalsocietypublishing.org/rsos/replication-studies#AuthorsGuidance>.

Kind regards,
Professor Chris Chambers
Royal Society Open Science
openscience@royalsociety.org

on behalf of Professor Chris Chambers (Registered Reports Editor, Royal Society Open Science)
openscience@royalsociety.org

Associate Editor Comments to Author (Professor Chris Chambers):

Associate Editor: 1

Comments to the Author:

The two original reviewers kindly evaluated the revised manuscript. Reviewer 2 is now satisfied and recommends in-principle acceptance following some minor revisions to provide additional information. Reviewer 1 again provides the more critical assesment, judging that neither of the primary Stage 1 criteria have been met. The reviewer provides a very helpful and detailed assessment, noting major concerns with the suitability of the testing environment, deviations from the original study, and appropriateness of the literature review and framing.

As in the first round, the issue of deviations does potentially push this submission out of frame for the Replications article type, but as I noted in the previous action letter, this is not necessarily a blocker as the article could be published under the Research Article type (following Stage 2 acceptance as a Replication), provided all other issues were resolved. I am also mindful of the challenges faced in this field in performing very close replications, which makes it even more important to eliminate publication bias.

Reviewer 1 helpfully provides a table summarising various differences in methodology between the current study and Hare et al. 2007, complementing similar tables that you provided in your previous response to reviewers. I think a comprehensive table showing all such differences will be an important addition to a revised manuscript, so please include a version of this in the main text. Given the range of differences, I think it would be also be prudent to label this study as a "conceptual" or "indirect" replication (e.g. in the title and elsewhere), as suggested by the reviewer.

These differences aside, there a number of additional issues that will need to be fully addressed from Review 1, including concerns regarding the testing environment (which the reviewer argues requires too much compromise on scientific validity), the level of balanced consideration of literature, and the accuracy of the figure depicting the test environment (see the reviewer's point headed "Misrepresentative figure").

I am mindful of overburdening reviewers and will reach a final Stage 1 decision following the next revision. Given the severity of the issues raised, I cannot promise that this decision will be favourable so please ensure that you do everything possibly to comprehensively address the reviewers' comments.

Comments to Author:

Reviewer: 1

Comments to the Author(s)

See attached file

Reviewer: 2

Comments to the Author(s)

As stated in my previous review, in principle, the findings of this study will help resolve debate over social tolerance and cooperation in two closely related species that exhibit different social behaviors. The authors provide a detailed description of their methods and analyses and the study will make a valuable contribution to the literature. I recommend 'accept with minor revision' with the following suggestions for improvement:

1. Provide information (e.g. a supplementary table) on the specific behaviors coded as prosocial, playful, or aggressive in section 4.2.
2. As described in Hare et al. 2007 (see also Hare, Brian, and Michael Tomasello. "The emotional reactivity hypothesis and cognitive evolution." 2005, which I'm surprised is not included in the literature cited) success on cooperative problem solving tasks can be hindered by aggression. In Hare et al. 2007, aggression was found to be non-significant/non-existent. The current study does not include aggressive behavior in the new analyses of co-feeding or cooperation, and it is unclear from the provided information whether this is due to an absence of aggressive behavior during trials or whether the authors had other justifications for only including affiliative behavior in the new models. If, as in Hare 2007, aggressive behaviors were found to be NS or did not occur, this should be stated. Otherwise, a justification should be provided.

Author's Response to Decision Letter for (RSOS-211106.R1)

See Appendix E.

RSOS-220194.R0

Review form: Reviewer 1

Do you have any ethical concerns with this paper?

No

Have you any concerns about statistical analyses in this paper?

No

Recommendation?

Accept with minor revision

Comments to the Author(s)

See attached (Appendix F).

Decision letter (RSOS-220194.R0)

Dear Ms Nolte,

On behalf of the Editors, I am pleased to inform you that your Manuscript RSOS-220194 entitled "Does Tolerance Allow Bonobos to Outperform Chimpanzees on a Cooperative Task? A conceptual replication of Hare et al., 2007" has been accepted in principle for publication in Royal Society Open Science subject to minor revision in accordance with the referee and editor suggestions. Please find their comments at the end of this email.

The reviewers and handling editors have recommended publication, but also suggest some minor revisions to your manuscript. Therefore, I invite you to respond to the comments and revise your manuscript.

Please you submit the revised version of your manuscript within 21 days (i.e. by the 11-May-2022). If you do not think you will be able to meet this date please let me know immediately.

To revise your manuscript, log into <https://mc.manuscriptcentral.com/rsos> and enter your Author Centre, where you will find your manuscript title listed under "Manuscripts with Decisions". Under "Actions," click on "Create a Revision." You will be unable to make your

revisions on the originally submitted version of the manuscript. Instead, revise your manuscript and upload a new version through your Author Centre.

Full author guidelines can be found here <https://royalsocietypublishing.org/rsos/replication-studies#AuthorsGuidance>.

Kind regards,
Professor Chris Chambers
Royal Society Open Science
openscience@royalsociety.org

on behalf of Professor Chris Chambers (Registered Reports Editor, Royal Society Open Science)
openscience@royalsociety.org

Associate Editor Comments to Author (Professor Chris Chambers):

I realise this final phase of the Stage 1 review process has taken longer than expected, and I am grateful for your ongoing patience. Following your recent revision, I decided to return the manuscript to Reviewer 1, with the specific aim to resolve (as much as possible) disagreements about the differences in your study and Hare et al. (2007). As you will see, the reviewer has now responded systematically to various sections of the tables, as well as offering an evaluation (and in some cases re-evaluation) of existing issues.

I have decided that your submission will be suitable for in-principle acceptance as a Stage 1 Replication, so this final revision and response has two specific purposes: (1) to respond to the reviewer's points (which could include reiterating your previous responses where the matter has been raised before) and make any further revisions where appropriate; and (2) to determine whether there is now sufficient clarity concerning the differences between your study and Hare et al. 2007 that you are able to include versions of Tables 1a, 1b and 1c from your previous response letter in the revised manuscript. Even when there are uncertainties, I feel that this table provides vital information and context for readers that should not be restricted to the review history.

I will leave this final Stage 1 review and your response to consider. Following your next revision, I will immediately issue in-principle acceptance at desk without further in-depth review.

Reviewers' comments to Author:

Reviewer: 1

Comments to the Author(s)

Attached file: Bonobo_replication_review_3.pdf

See attached

Author's Response to Decision Letter for (RSOS-220194.R0)

See Appendix G.

Decision letter (RSOS-220194.R1)

Dear Ms Nolte

On behalf of the Editor, I am pleased to inform you that your Manuscript RSOS-220194.R1 entitled "Does Tolerance Allow Bonobos to Outperform Chimpanzees on a Cooperative Task? A conceptual replication of Hare et al., 2007" has been accepted in principle for publication in Royal Society Open Science. The reviewers' and editors' comments are included at the end of this email.

You may now progress to Stage 2 and complete the study as approved.

Please note that you must now register your approved protocol on the Open Science Framework (<https://osf.io/rr>), using the 'Submit your approved Registered Report' option and then the 'Registered Report Protocol Preregistration' option. Please use the Registered Report option even though your article is being accepted as a Stage 1 Replication. Further into the registration process, in the Journal Title field enter 'Royal Society Open Science (Replication article type, Results-Blind track)'. Please note that a time-stamped, independent registration of the protocol is mandatory under journal policy, and manuscripts that do not conform to this requirement cannot be considered at Stage 2. The protocol should be registered unchanged from its current approved state. Please include a URL to the protocol in your Stage 2 manuscript, and because you submitted via the Results-Blind track please note in the manuscript that the pre-registration was performed after data analysis (e.g. 'This article received results-blind in-principle acceptance (IPA) at Royal Society Open Science. Following IPA, the accepted Stage 1 version of the manuscript, not including results and discussion, was preregistered on the OSF (URL). This preregistration was performed after data analysis.')

Following completion, we invite you to resubmit your paper for peer review as a Stage 2 Replication. Please note that your manuscript can still be rejected for publication at Stage 2 if the Editors consider any of the following conditions to be met:

- The Introduction and methods deviated from the approved Stage 1 submission (required).
- The authors' conclusions were not considered justified given the data.

We encourage you to read the complete guidelines for authors concerning Stage 2 submissions at: <https://royalsocietypublishing.org/rsos/replication-studies#AuthorsGuidance>. Please especially note the requirements for data sharing and that withdrawing your manuscript will result in publication of a Withdrawn Registration.

Once again, thank you for submitting your manuscript to Royal Society Open Science and I look forward to receiving your Stage 2 submission. If you have any questions at all, please do not hesitate to get in touch. We look forward to hearing from you shortly with the anticipated submission date for your stage two manuscript.

Kind regards,
Professor Chris Chambers
Royal Society Open Science
openscience@royalsociety.org

Author's Response to Decision Letter for (RSOS-220194.R1)

See Appendix H.

Decision letter (RSOS-220194.R2)

Dear Ms Nolte:

I am pleased to inform you that your manuscript entitled "Does Tolerance Allow Bonobos to Outperform Chimpanzees on a Cooperative Task? A conceptual replication of Hare et al., 2007" is now accepted for publication in Royal Society Open Science.

Please remember to make any data sets or code libraries 'live' prior to publication, and update any links as needed when you receive a proof to check - for instance, from a private 'for review' URL to a publicly accessible 'for publication' URL. It is also good practice to add data sets, code and other digital materials to your reference list.

Royal Society Open Science is a fully open access journal. A payment may be due before your article is published. Our partner Copyright Clearance Center's RightsLink for Scientific Communications will contact the corresponding author about your open access options from the email domain @copyright.com (if you have any queries regarding fees, please see <https://royalsocietypublishing.org/rsos/charges> or contact authorfees@royalsociety.org).

on behalf of Professor Professor Chris Chambers (Subject Editor)

Follow Royal Society Publishing on Twitter: @RSocPublishing
Follow Royal Society Publishing on Facebook:
<https://www.facebook.com/RoyalSocietyPublishing/>
Read Royal Society Publishing's blog:
<https://royalsociety.org/blog/blogsearchpage/?category=Publishing>

Appendix A

Royal Society Open Science: Does tolerance allow bonobos to outperform chimpanzees on a cooperative task? A replication of Hare et al 2007

The authors present an introduction and method section that is meant to replicate a study on chimpanzee versus bonobo cofeeding and cooperation done by Hare et al. (2007). To do so, they report methods for a cofeeding task experiment with 16 chimps and 9 bonobos (89 unique chimp and 11 unique bonobo dyads) and then a loose string task experiment with 11 chimpanzees and 7 bonobos (53 unique chimp and 9 unique bonobo dyads*). During both experiments the authors report they also recorded aggressive and sociosexual behaviors. They then examined species differences with Welch t-tests. To examine what additional factors might explain cofeeding/cooperation they then created GLMM models for using variety of variables to analyze the cofeeding data, the cooperation data, and Hare et al (2007)'s original data. Below are our major concerns with the introduction and methods that we hope will be helpful in revising the manuscript:

- 1) Remove reproducibility section: This section eats up space that should be used to more carefully review the growing literature on chimpanzee and bonobo sharing that the authors currently do a relatively poor job reviewing (see below). Also it reads as if replication is some novel idea while methods books have emphasized the importance of replication in animal studies for decades (e.g. Martin and Bateson, 1992). The literature is not perfect but it is replete with examples of failed and successful replications. No need to virtue signal instead of thoroughly reviewing existing relevant replications that are currently missing from the review.
- 2) Improve scholarship: The literature cited is woefully unrepresentative of the current state of the field. This is particularly surprising since other papers by the first author are very well cited and balanced. Examples of missing citations include: published replications of the co-feeding experiment reported in Hare et al 2007 including physiological explanations for the differences observed; multiple replicated experimental studies and a field study showing bonobos voluntarily share food with non-group members; experiments showing bonobos prefer to help and share with non group mates and other studies showing how bonobos prefer to actively share food over tools. Moreover, the authors cast the evidence as if there is confusion in the literature by using studies that fail to include any condition for bonobos to share w/ non-group members (e.g. Bullinger et al) or include a tiny sample of bonobos (6-8 individuals) in comparison to dozens of chimpanzees (Jeaggi et al; Cronin et al). Also included is field work claiming bonobos are more afraid of snakes b/c they had less information even though there are several demonstrations that bonobos are highly reactive in comparison to chimpanzees...i.e. they probably seemed more nervous b/c bonobos are more nervous! Other work on bonobo vocal behavior suggests important species differences but is left out. None of the limitations of any of the studies included are noted leading to a false equivalency. We can all look at the evidence and come to different interpretations and hypotheses but this review is currently just way too far off where the literature is.

3) Not a replication: The authors state: “Here, we aimed to replicate a seminal study by Hare and colleagues (2007, *Cur. Biol.* 17, 619– 623. doi: 10.1016/j.cub.2007.02.040), who found that bonobos co-fed more often than chimpanzees on both monopolizable and sharable food resources and additionally outperformed chimpanzees on a cooperative task when the food was monopolizable....” The authors have badly missed their mark. They should drop the replication phrase in the title and recast this work as an original but related test that replicates some aspects of the previous work. This study is so different in enough important ways from the original study that this is a new methodology – not remotely a faithful replication. That doesn’t mean the study is not valuable but it is definitely not a replication even using generous definitions. Here are some of the more obvious differences that the authors should highlight in a discussion:

- 1) Different subjects: The current study used way fewer individuals, with different experimental history, ages, and origin than the original study. With so little power to detect a difference how is this a replication in the spirit of reproducibility? The human studies in the vein have equal or larger samples and expand heterogeneity of participants – here we move in the opposite direction (especially for power!).
 - a. Low power: The current study included <50% the sample size of bonobos and only 50% the sample of chimpanzees in the co-feeding task from Exp 1 of Hare et al. (Hare et al. used 32 chimpanzees and 20 bonobos for their cofeeding experiment). The current work includes <60% the sample of bonobos in the cooperation test of Exp 2-3 in Hare et al. (Hare et al included 12 bonobos for their cooperation task). There is no indication that the chimpanzees in the current cooperation study were the most tolerant dyads (Hare et al selected dyads that were among the most tolerant dyads out of the dozens possible to include).
 - b. Different experimental histories: Hare et al tested chimpanzees that were largely experimentally naïve (having only been in a handful of studies and recently demonstrated an understanding of the cooperative pulling task) and the bonobos were completely naïve. Many of the chimpanzees and bonobos in the current work have been tested almost daily for decades - including dozens of cooperation tasks. We are provided with no information of this history.
 - c. Much older animals: Subjects mean age in current study is >2x the age of the subjects in Hare et al 2007. There were adult subjects in Hare et al 2007 but the mean age was in the older juvenile age range while here it is older adult range. Other studies have shown juveniles are more tolerant than adults. Hare et al was able to match the age and sex of subjects across species but that was not possible here given sample size limits.

- d. Different origins: All the chimpanzees and bonobos tested in the current study are captive born zoo living while Hare et al tested wild born sanctuary living apes (i.e. living in outdoor forest enclosure ~10-20x the size of the zoo apes and large multemale-female mixed groups with dozens of individuals in their groups – combined this allows for fission fusion, exercise (climbing/travel) and ad libitum browsing on low quality herbs more similar to what is found in wild populations; “meals” are an artificial invention of Western managers and may lead to very different behavior around food / food sharing).

2) Different procedure:

- a. Food rewards: Food was distributed for co-feeding and cooperation similarly in relative terms but in absolutely much smaller amounts than Hare et al. In both the cofeeding and dispersed-divisible cooperation experiments Hare et al used 0.5 kg to 1.5 kg of bananas (~ 4 to 10 bananas) versus the current study using 3 to 6 cm slices of a banana (~1/4 to 1/2 banana). This difference alone nullifies this as a replication if we are trying to look at link between feeding tolerance and cooperation! The current study basically uses a completely novel set of feeding conditions.
- b. No reaching: The subjects in the current study must work through small mesh holes in the walls to get food and the cooperation rope. The sanctuary set-up used by Hare et al allowed subjects to reach both of their entire arms out b/c of the lack of metal mesh common in zoos. They also threw the rope to the two waiting subjects in Exp 2 and 3 but the mesh prevents this method and will increase/exaggerate failure to coordinate simply b/c individuals are nervous around the experimenters or are not attending simultaneously.
- c. Missing measures?: There is no mention of whether they measured what percent of food was monopolized in the cooperation tasks as reported in Hare et al.

- 3) Different Design: Hare et al. 2007 first assigned dyads who were tested in the cofeeding (6 trials only) and the initial cooperation experiment (6 trials only). One year after this first experiment a subset of the most tolerant chimpanzee dyads were compared to age-sex matched bonobos. The chimpanzees had participated in several studies showing they understood to coordinate roles in this task but the bonobos were otherwise still completely naïve to the task. The bonobos were given warm up trials to make sure each dyad could solve the task. Once they completed the 3 trial warm up each dyad from each species was given 6 cooperation trials in with highly sharable food and 6 with highly monopolizable food. During this entire time they remained within the same dyad. Only in experiment 3 for 6 trials were subjects repaired and analyzed completely separately.

In the current study the experimenters opted to drop the entire first experiment as well as the warm up for the bonobos (it seems anyway – apologies if we misunderstood). Due to the reduced sample available for testing it seems the experiments were forced to re-pair each subject multiple times with multiple different partners to maximize trial numbers that each subject participated (as opposed to keeping independent dyads as in the previous study) – this is fundamentally a different design that raises questions about the independence of observations (and analytic issues) due to re-pairing – again more reason to say this is a novel experiment. Definitely not a replication.

- 4) Analytic questions: GLMM appears to include interactions but not necessarily main effects. Main effects of any interaction should be included in a model even if they are insignificant. GLMM is very complex and a little hard to follow. Generally, would recommend a rewrite of this section to be more clear. It would be nice if the authors included more information on what method they chose to order their removal of non-significant variables. The order in which predictors are removed can define what overall predictors end up significant in a final model.

Appendix B

Dear Reviewer(s)

Thank you for your response; we address all raised concerns below.

1) Remove reproducibility section: This section eats up space that should be used to more carefully review the growing literature on chimpanzee and bonobo sharing that the authors currently do a relatively poor job reviewing (see below). Also it reads as if replication is some novel idea while methods books have emphasized the importance of replication in animal studies for decades (e.g. Martin and Bateson, 1992). The literature is not perfect but it is replete with examples of failed and successful replications. No need to virtue signal instead of thoroughly reviewing existing relevant replications that are currently missing from the review.

- We address the first concern regarding reviewing the literature below.
- Concerning the value of replications in science: Indeed, the importance has been highlighted previously and we have now shortened the paragraph and added the suggested citation (see line 65). However, to our knowledge, no studies have been done so far on the topic of bonobo and chimpanzee cooperation and social tolerance that report a replication of original measurements. Even though there are several studies that report on tolerance and/or cooperation *in either one species* (this is a really broad field, as the terms tolerance and cooperation can be interpreted quite liberally), in our opinion, a valid comparison between the species can only be made by subjecting both species to the exact same conditions, like in Hare et al., 2007. This is why we steer clear from single-species studies, review the literature on direct comparisons between the species, and aimed to replicate Hare et al., 2007 to the best of our abilities (see below for more details on that).

2) Improve scholarship: The literature cited is woefully unrepresentative of the current state of the field. This is particularly surprising since other papers by the first author are very well cited and balanced. Examples of missing citations include: published replications of the co-feeding experiment reported in Hare et al 2007 including physiological explanations for the differences observed; multiple replicated experimental studies and a field study showing bonobos voluntarily share food with non-group members; experiments showing bonobos prefer to help and share with non group mates and other studies showing how bonobos prefer to actively share food over tools. Moreover, the authors cast the evidence as if there is confusion in the literature by using studies that fail to include any condition for bonobos to share w/ non-group members (e.g. Bullinger et al) or include a tiny sample of bonobos (6-8 individuals) in comparison to dozens of chimpanzees (Jeaggi et al; Cronin et al). Also included is field work claiming bonobos are more afraid of snakes b/c they had less information even though there are several demonstrations that bonobos are highly reactive in comparison to chimpanzees...i.e. they probably seemed more nervous b/c bonobos are more nervous! Other work on bonobo vocal behavior suggests important species differences but is left out. None of the limitations of any of the studies included are noted leading to a false equivalency. We can all look at the evidence and come to different interpretations and hypotheses but this review is currently just way too far off where the literature is.

- The current study focuses on a direct comparison of tolerance and cooperation in chimpanzees and bonobos. Therefore, we focus our introduction on papers that also provide a direct comparison. We made several choices for the literature

reported. First, to keep the introduction concise, we chose not to elaborate on papers on tolerance / cooperation reporting on one species, since from single species studies it is difficult to deduce species differences. To make this clearer to the reader, we now included such a paragraph in introduction (please see line 44-51). Second, we focus on within-group processes, since between-group tolerance is altogether a different topic. Lastly, we recognize that chimpanzees and bonobos may differ in numerous aspects (e.g., nervousness), but we focus the introduction on the topic at hand: within-group tolerance and cooperation. We will answer each of the concerns raised separately.

- **“published replications of the co-feeding experiment reported in Hare et al 2007 including physiological explanations for the differences observed“**: In this light, we only found one study that reported physiological explanations for potential species differences (Wobber et al. 2010). Indeed, the authors used the same methods as in the co-feeding task of Hare et al., 2007. However, the authors do not report whether the two species differ with regard to how often they co-fed, but report on their physiological responses to the experiment that may explain behavioural differences between males of both species. Given that we did not find reports of which species co-fed more frequently, we do not think that the publication can be used as a replication reporting on tolerance. However, we discuss this study in our Discussion to explain species differences.
- **“multiple replicated experimental studies and a field study showing bonobos voluntarily share food with non-group members”**: There are indeed several studies, e.g. Tan & Hare 2013; Tan, Ariely & Hare, 2017; Fruth & Hohmann, 2018, reporting on between-group tolerance. However, these studies are not directly relevant for our study: first, they concern *between*-group tolerance, while our study focusses on *within*-group tolerance, and second, quite essentially, they do not compare chimpanzees and bonobos.
- **“experiments showing bonobos prefer to help and share with non group mates” and “using studies that fail to include any condition for bonobos to share w/ non-group members (e.g. Bullinger et al)”**: See above, this is interesting, but not pertaining to the goal of our study.
- **“other studies showing how bonobos prefer to actively share food over tools”**: Idem, these studies are only conducted with bonobos and lack the comparison to chimpanzees. As stated above, we however now included a paragraph that references some of these single species studies (line 44-51).
- **“or include a tiny sample of bonobos (6-8 individuals) in comparison to dozens of chimpanzees (Jaeggi et al; Cronin et al)”**: We do not deem it appropriate to ignore studies with a relatively small data set. Moreover, in comparative studies often relatively small numbers of bonobos are studied. This also pertains to the mentioned studies of Jaeggi and Cronin. We address these limitations in the Discussion.
- **“Also included is field work claiming bonobos are more afraid of snakes b/c they had less information even though there are several demonstrations that bonobos are highly reactive in comparison to chimpanzees...i.e. they probably seemed more nervous b/c bonobos are more nervous!”**: Notwithstanding the relevance of this study for information transfer, we agree with the reviewer that including this reference is exceeding the scope of the current paper since it does

not specifically focus on social tolerance and/or goal-oriented cooperation. We have therefore omitted the reference from our ms.

- **“Other work on bonobo vocal behavior suggests important species differences but is left out.”**: See above, this is interesting, but not pertaining to the goal of our study.

3) Not a replication: The authors state: “Here, we aimed to replicate a seminal study by Hare and colleagues (2007, *Cur. Biol.* 17, 619– 623. doi: 10.1016/j.cub.2007.02.040), who found that bonobos co-fed more often than chimpanzees on both monopolizable and sharable food resources and additionally outperformed chimpanzees on a cooperative task when the food was monopolizable....” The authors have badly missed their mark. They should drop the replication phrase in the title and recast this work as an original but related test that replicates some aspects of the previous work. This study is so different in enough important ways from the original study that this is a new methodology – not remotely a faithful replication. That doesn’t mean the study is not valuable but it is definitely not a replication even using generous definitions.

- The aim of the study by Hare et al (2007) was to compare tolerance and cooperation of chimpanzees and bonobos, focusing on species differences. Our study aims to replicate the Hare et al. (2007) methods as closely as possible (n.b., our study was specifically funded in a Replication Grant scheme). Such comparisons require that the two species are compared, yet for generalizability to the species level the individuals do not have to be exactly the same, and it requires that possible confounds such as individual experience and age are controlled for. We do this either by design (e.g., experience) or in our statistical analyses (e.g. age). Next, the relevant methods should be replicated as closely as possible, which we did with the exception that the *amount* of food provided in three of the four original conditions was not allowed by the current management standards in the zoo (WKPRC). Therefore, we used the amount of food of the fourth original condition (the least amount) throughout all of the studies and conditions (i.e., thereby ascertaining internal consistency and thus comparability). This is the only deviation from the original study, and we mention this in our introduction and also address it in our discussion. Moreover, we replicated the relevant experiments, i.e., 1 and 3. We did not replicate experiment 2 as the original authors only tested one condition but also integrated it in experiment 3, which we did replicate. Overall, in all aspects, except for the amount of food provided, we consider our study a pure replication of the Hare et al. (2007) study. Below, we provide a more detailed account.

Here are some of the more obvious differences that the authors should highlight in a discussion:

- 1) Different subjects: The current study used way fewer individuals, with different experimental history, ages, and origin than the original study. With so little power to detect a difference how is this a replication in the spirit of reproducibility? The human studies in the vein have equal or larger samples and expand heterogeneity of participants – here we move in the opposite direction (especially for power!).**

a. Low power: The current study included <50% the sample size of bonobos and only 50% the sample of chimpanzees in the co-feeding task from Exp 1 of Hare et al. (Hare et al. used 32 chimpanzees and 20 bonobos for their cofeeding experiment). The current work includes <60% the sample of bonobos in the cooperation test of Exp 2-3 in Hare et al. (Hare et al included 12 bonobos for their cooperation task). There is no indication that the chimpanzees in the current cooperation study were the most tolerant dyads (Hare et al selected dyads that were among the most tolerant dyads out of the dozens possible to include).

→ Indeed, the sampled groups encompass fewer individuals than those in the original study, yet a similar or larger number of dyads. Across all three experiments, Hare et al. used 20 bonobos and 35 chimpanzees that when summing across experiments contributed to 22 unique bonobo and 28 unique chimpanzee dyads. However, given the split of experiments, it is not possible to analyze the data within-subject. We tested 9 bonobos and 16 chimpanzees that contributed to 11 unique bonobo and 89 unique chimpanzee dyads for the co-feeding and 9 unique bonobo and 53 unique chimpanzee dyads for the cooperation task, across experiments we have 20 bonobo and 142 chimpanzee dyads. Since we tested each dyad in both experiments, we use a within-subject design that allows to control for influential dyad characteristics. While being aware of the fact that we could only test substantially fewer bonobos than chimpanzees and discussing these limitations in the discussion, we think the data are valuable in three respects. First, both groups were housed at the same facility and we can therefore rule out husbandry related factors such as diet and general treatment. Comparing species that are housed at different facilities always comes with the drawback of uncertainty to what extent husbandry differences influenced results. Second, instead of testing those dyads that are particularly tolerant, we tested all possible dyads and report on those that we could not include due to high intolerance levels. This gives a clearer picture of the species' general tolerance measures. Finally, we included all dyads in all tests which means that each dyad contributes data to every aspect of the experiment.

	Experiment 1 & 2, nbr unique individuals	Experiment 1 & 2, tested dyads	Experiment 3, nbr unique individuals	Experiment 3, tested dyads
Hare et al 2007	20 bonobos, 32 chimpanzees	10 bonobos, 16 chimpanzees	subset of exp. 1 & 2 plus 3 additional chimps	12 bonobos, 12 chimpanzees (six of each received both conditions, six received condition clumped only)
	Co-feeding, nbr unique individuals	Co-feeding, tested dyads	Cooperation, nbr unique individuals	Cooperation, tested dyads
Current study	9 bonobos, 16 chimpanzees	11 bonobos, 89 chimpanzees	same dyad composition	9 bonobos, 53 chimpanzees

b. Different experimental histories: Hare et al tested chimpanzees that were largely experimentally naïve (having only been in a handful of studies and recently demonstrated an understanding of the cooperative pulling task) and the bonobos were completely naïve. Many of the chimpanzees and bonobos in the current work have been tested almost daily for decades – including dozens of cooperation tasks. We are provided with no information of this history.

→ We actually think it is an advantage of this study that it does not replicate the demographics of the original study since we would like to understand generalizable species typical levels of tolerance and cooperative abilities and not those of a specific set of animals. We address the different demographics in the discussion. With regard to naïve versus experienced in participating in cognitive and cooperative tests, the influence should be minor when it comes to being tolerant of other individuals. We do not know of any study that showed tolerance levels will rise when being subjected to the same test over a long period of time. Even if that would be true, this is not the case for our sample because 1) the test was conducted once per dyad and due to the keepers' decisions any kind of dyadic co-feeding tests are rarely allowed anymore at WKPRC, and 2) there was nearly no testing at WKPRC since about two years due to changes in directorship of the MPI department. As for reporting information on the experience of the subjects, please see line 104 where we did state: “and have extensive experience with various cognitive and behavioural tests”.

c. Much older animals: Subjects mean age in current study is >2x the age of the subjects in Hare et al 2007. There were adult subjects in Hare et al 2007 but the mean age was in the older juvenile age range while here it is older adult range. Other studies have shown juveniles are more tolerant than adults. Hare et al was able to match the age and sex of subjects across species but that was not possible here given sample size limits.

→ Again, we do not think it would have been goal oriented if we would have also tested only young individuals. We covered a broader range and did include the age of both partners as control predictors in the model to account for their influence. If outcomes would have been sensitive to such minor deviations, then we would need to discuss why. Moreover, the original study speaks of species differences, not differences between juvenile/sub-adult bonobos and chimpanzees.

d. Different origins: All the chimpanzees and bonobos tested in the current study are captive born zoo living while Hare et al tested wild born sanctuary living apes (i.e. living in outdoor forest enclosure ~10-20x the size of the zoo apes and large multemale-female mixed groups with dozens of individuals in their groups – combined this allows for fission fusion, exercise (climbing/travel) and ad libitum browsing on low quality herbs more similar to what is found in wild populations; “meals” are an artificial invention of Western managers and may lead to very different behavior around food / food sharing).

→ This is arguing that none of the published studies using zoo-living chimpanzees and bonobos are valid with regard to arguing about their behaviour and cognition. Indeed, zoo-living Pan species are facing different conditions than wild or

sanctuary living groups, however, the effects are yet unclear. Understanding differences in behaviour and cognition based on housing conditions will further help us to gauge the behavioural and cognitive flexibility and factors influencing its expression. Here we purposefully tested two groups that were housed similarly to rule out such effects.

2) Different procedure:

a. Food rewards: Food was distributed for co-feeding and cooperation similarly in relative terms but in absolutely much smaller amounts than Hare et al. In both the cofeeding and dispersed-divisible cooperation experiments Hare et al used 0.5 kg to 1.5 kg of bananas (~ 4 to 10 bananas) versus the current study using 3 to 6 cm slices of a banana (~1/4 to 1/2 banana). This difference alone nullifies this as a replication if we are trying to look at link between feeding tolerance and cooperation! The current study basically uses a completely novel set of feeding conditions.

→ Indeed, this is the largest difference between the two studies. We attempted to replicate the methods as closely as possible, however, we were not allowed to give such large amounts of food as was done in three of the four replicated conditions of the Hare et al study. It is unclear how the results may have been influenced by the food amount and in case that is possible at a facility, it would be very valuable to test both species with varying food amounts. To stay as close to the original study as possible, we took the amount of food used in the Hare et al cooperation task (clumped-divisible) and used this specific amount of food in all tasks. Thus, we replicated the Hare study at this point. Additionally, we thought it would be helpful to use the same food amounts in both tests to enable comparison across experiments and across conditions within experiments. In conjunction, while we do see the difference, we believe that comparing the two studies is still valid.

	Dispersed-divisible	Clumped-divisible	Clumped
Hare et al. Co-feeding	250g of sliced fruit at either end	500g of sliced fruit in the centre	two equally sized pieces of fruit in the centre
Current study Co-feeding	two 1.5cm thick pieces of banana at either end	four 1.5cm thick pieces of banana in the centre	two 3cm thick pieces of banana in the centre
Hare et al. Cooperation	250g of sliced fruit at either end	four 1.5cm thick pieces of fruit in the centre	Not included
Current study Cooperation	two 1.5cm thick pieces of banana at either end	four 1.5cm thick pieces of banana in the centre	Not included

b. No reaching: The subjects in the current study must work through small mesh holes in the walls to get food and the cooperation rope. The sanctuary set-up used by Hare et al allowed subjects to reach both of their entire arms out b/c of the lack of metal mesh common in zoos. They also threw the rope to the two waiting subjects in Exp 2 and 3 but the mesh prevents this method and will increase/exaggerate failure to coordinate simply b/c individuals are nervous around the experimenters or are not attending simultaneously.

→ Indeed, the mesh did not allow individuals to stick their arm through the mesh, we now included such a remark on line 158-160. However, both individuals had the same chance of reaching and obtaining food and the set up was the same for both species. We therefore think it is possible to compare the reaction of both species. In the light of replication, we agree that tolerance and cooperation may be influenced by such differences, however, the extent is unknown. Therefore, our study adds to the cumulating evidence and discusses results from other groups that faced other housing conditions than the ones tested originally.

c. Missing measures?: There is no mention of whether they measured what percent of food was monopolized in the cooperation tasks as reported in Hare et al.

→ We will indeed provide such a measure in the result section. Thank you for identifying the missing piece of information.

3) Different Design: Hare et al. 2007 first assigned dyads who were tested in the cofeeding (6 trials only) and the initial cooperation experiment (6 trials only). One year after this first experiment a subset of the most tolerant chimpanzees dyads were compared to age-sex matched bonobos. The chimpanzees had participated in several studies showing they understood to coordinate roles in this task but the bonobos were otherwise still completely naïve to the task. The bonobos were given warm up trials to make sure each dyad could solve the task. Once they completed the 3 trial warm up each dyad from each species was given 6 cooperation trials in with highly sharable food and 6 with highly monopolizable food. During this entiretime they remained within the same dyad. Only in experiment 3 for 6 trials were subjects repaired and analyzed completely separately. In the current study the experimenters opted to drop the entire first experiment as well as the warm up for the bonobos (it seems anyway – apologies if we misunderstood). Due to the reduced sample available for testing it seems the experiments were forced to re-pair each subject multiple times with multiple different partners to maximize trial numbers that each subject participated (as opposed to keeping independent dyads as in the previous study) – this is fundamentally a different design that raises questions about the independence of observations (and analytic issues) due to repairing – again more reason to say this is a novel experiment. Definitely not a replication.

→ We did include the first experiment and replicated the methods as closely as possible. We did not include the second experiment given that the used condition was included in the third experiment but with re-paired dyads. Thus, we combined

the methods applied for experiment 2 and 3 since this allowed us to compare the co-feeding scores of each dyad with the cooperation scores for both conditions. We provide this rationale.

	Experiment 1 / Co-feeding	Experiment 2 / Cooperation d.d	Experiment 3 / Cooperation both
Hare et al 2007	6 trials per dyad, 2 trials per three conditions	6 trials per same dyad as used in exp.1, only dispersed divisible (sharable condition)	new dyads based on subset of exp. 1 & 2 plus 3 additional chimps, both conditions tested per dyad
Current study	6 trials per dyad, 2 trials per three conditions	Not conducted and instead chance to compare both coop. conditions to co-feeding	same dyads as in co-feeding, both conditions tested per dyad

- ➔ We did not include a warm-up for the bonobos since they were similarly knowledgeable as the tested chimpanzees. For both groups, we ensured that both partners understood the contingencies of the apparatus. Only those apes that showed they understood how to use the apparatus (please see details in the methods) were tested with a partner in order to know that they actually understand the role of their partner and do not manipulate the rope out of other motives. This is mentioned in the methods section.
- ➔ We decided to re-pair the apes to understand their tolerance levels and motivation to cooperate between multiple members of the group. We also report on all dyads that were not possible to test due to their intolerance. Pre-selecting dyads based on their tolerance would have only allowed us to understand the behaviour of a specific sample of both species but not to infer more generalizable behaviour. To avoid pseudo-replication, we included the random effects of the identity of both partners and the identity of the dyad. This allows for interpreting results independently of such influences. Overall, we argue that this does not harm the replication: if there are true species differences, then also when re-pairing subject should the effects emerge; they are all still the same species.

4) Analytic questions: GLMM appears to include interactions but not necessarily main effects. Main effects of any interaction should be included in a model even if they are insignificant. GLMM is very complex and a little hard to follow. Generally, would recommend a rewrite of this section to be more clear. It would be nice if the authors included more information on what method they chose to order their removal of non-significant variables. The order in which predictors are removed can define what overall predictors end up significant in a final model.

- ➔ We followed the widely used procedure of excluding all interactions from the model that are non-significant and re-running the model without them. Given that one is not allowed to interpret main effects when the model includes higher order interaction terms, we followed this procedure in order to gauge whether the main effects themselves significantly influenced the response. Non-significant main effects were never excluded from the model. In case interactions were significant, we did not remove them. Of course, all lower order terms are always included in the model (this is reported on line 317, 348, 370, and 387) and we now included a clear statement of such on line 287. Tables with results of the models that give a visual overview of the model predictors and their influence are provided.

Appendix C

Dear Reviewer,

Thank you very much for your feedback and description of the study.

Both comments were very helpful, and we adapted the method section accordingly.

Thank you!

All the best

Suska Nolte, Liesbeth Sterck, Edwin van Leeuwen

Reviewer: 2

Comments to the Author(s)

The study makes valuable contribution to the literature on social tolerance and cooperation. The article is well written, clear, and concise. The authors provide a thorough review of relevant literature in Pan spp. and the existing discrepancies in results.

The authors provide a detailed description of their methods and analyses. In addition to directly replicating the original paper's analyses, the authors also updated their analyses to include GLMMs. Importantly, the authors extracted the data from the original study and reanalysed the data using GLMMs. This approach will provide valuable insight into analytical approaches typically used in these types of studies.

In principle, the findings of this study will help to resolve debate over social tolerance and cooperation in two closely related species which exhibit different social behaviours; study results will make a valuable contribution to literature investigating questions surrounding the evolutionary links between social tolerance and cooperative abilities.

I have selected 'accept with minor revision' due to one area of the manuscript that was unclear:

Line 104 states "due to special animal handling regulations for the bonobos, no adult mixed pairs could be tested, but only mothers with their male offspring."

However, it is unclear whether bonobo dyads were tested in only mother/male offspring pairs for both co-feeding and cooperation trials. It is also unclear whether subadult bonobos were tested in unrelated pairs.

→ Indeed, we see how this part could be confusing to the reader and adapted it slightly. It now includes "adult" and "juvenile sons" whenever needed (line 108-125).

In their analyses of co-feeding, the authors included maternal relatedness as an explanatory variable but this variable was not included for the analyses of cooperation data. Line 237 states: "We did not include maternal relationship because none of the bonobo and only six of the chimpanzee dyads that we tested in the cooperation experiment were maternally related and a model that included the factor resulted in substantially inflated standard errors." This line indicates that bonobos were tested

in non-related pairs for the cooperation trials. Thus, I suggest the authors make a minor revision to clarify dyad composition for both trial types (co-feeding and cooperation). This can be easily accomplished by including a few lines in the methods section or a supplementary table that lists the dyads and the relationship between dyad members.

- ➔ Yes, this information is helpful to the reader. Please see line 115-116 and 123-125 for added information.

Appendix D

I remain with major reservations regarding this paper, but before I detail those a couple of points. First, I think this study is ultimately publishable and represents a contribution to the literature with different framing. I also applaud the desire to replicate even though I do not think this effort can be characterized that way (faithful replication in this case will likely require sanctuary work). Second, I obviously have only seen the introduction and methods and have no idea what the authors have found / will find since I have not seen the results. But my concerns apply regardless of the findings. All my comments are simply about the framing and comparability of the methods. After carefully reading the response and revision it is clearer this study should not be characterized as a methodological replication of previous methods. I certainly think it can be framed as a conceptual replication (i.e. designed to test the same ideas) but too many critical variables are just too different (see Table 1). I hope the authors take the thoughts below and the time put into it as a good faith effort to help them and everyone get the most out of the work they are trying to publish. I will be curious to see the findings with these new methods regardless of what they are.

Suboptimal: I say suboptimal because the authors of the original Leipzig team from Hare et al 2007 decided **not** to conduct this work at the WKPRC even though the same exact set up and animals were available to them over a decade earlier. The original team realized conducting this kind of work at the WKPRC required far too many design compromises that the current authors have again faced (small samples ($n < 10$), too little flexibility in experimental set up (no reaching), fear of caretakers by the bonobos, and required dietary restrictions (preventing use of enough food to allow for robust cofeeding/sharing). And this realization was earned. Melis et al 2006 (*Anim Behaviour*) conducted and published an initial pilot study in this same facility with some of the same subjects the authors are working with. Due to the constraints above experienced during that pilot the team largely abandoned the zoo for this type of work. That paper and all that teams papers that followed on feeding tolerance and instrumental cooperation were conducted in African sanctuaries because they allowed these constraints to be escaped. Another way to say this is – Why would the previous team go to all the trouble to work in Uganda, Congo-Brazza and D.R.C. if they could just do the same thing in LEJ where they already lived? The LEJ building is well suited for lots of studies but this type is not one of them.

Poor Scholarship: Unfortunately, their scholarship is not there yet. A balanced introduction would present evidence for and against species differences in tolerance and cooperation between the two species and would include both captive and wild observations. The authors suggest this is not necessary because they are focused on Hare et al 2007 and studies that involve direct comparisons – but they then skip over Hare & Kwetuenda 2010 *Curr Bio*; Tan & Hare, 2013 *PLoS One*; Tan et al 2019 *Scientific Reports* that present a type of tolerance and sharing with non-group members in bonobos that 1) has never observed in chimpanzees but has since been observed in wild bonobos (e.g. Fruth & Hohman, 2018, *Human Nature*; also see Yamamoto & Furiuchi 2017) and 2) could not even be tested in chimps due to safety issues. Based on their revision, it also seems the authors are asking the audience to ignore all the other work that built toward Hare et al 2007 and then followed it in support (i.e. Melis et al, 2006 a;b; Hare et al, 2007, Wobber et al, 2010 a;b). They justify this by saying they are only attempting to

replicate Hare et al 2007 and only cite studies that provide quantitative comparisons. I hope the authors will reconsider in future versions. Here is why:

A series of studies were completed at Ngamba island (Melis et al, 2006 a;b; 2009) before Hare et al 2007 was piloted. These initial studies provided the initial link b/w tolerance and cooperation using a sample of 36 chimpanzees (i.e. making it clear robust social groups (N=20-30) were needed to have enough power; this relationship has been replicated in a host of other species – see references for examples in rooks, ravens, macaques and marmosets). They also taught us about the amount of food that was needed to see robust and spontaneous cooperation using the 3 conditions – typically several entire bananas cut up (~500g) in multiple dishes (i.e. there is significantly less tolerance w/ less food). We also learned that performance in the causal understanding pre-test in Melis et al 2006 was not related to cooperation success (i.e. making rope training of subjects in future work unnecessary). We also discovered the importance of open walls that the apes could reach through (reducing the impact of individual differences in self-control in waiting for others to grab the rope). Entire buildings in Uganda, DRC and Congo Brazzaville were designed and renovated to allow for this type of work (opposed to just suffering with the fixed design at WKPRC). Based on all this experience we designed and conducted Hare et al 2007. The study was built on many others and that phylogeny of a method may matter if the intent is to robustly replicate.

Some of the review borders on caricature. Hare et al 2007 was designed to test the hunting hypothesis that predicted more cooperation in chimpanzees than bonobos when working to obtain food since chimpanzees cooperatively hunt much more than bonobos. The initial two experiments show that the bonobos and chimpanzees perform similarly even though the chimpanzees had more experience with the experimental task (rejecting the hunting hypothesis). However, the authors fail to mention this framing. Instead, the authors appear to cite recent examples of hunting in bonobos as if that means the hunting behavior b/w the two species is recently discovered and similar. First, bonobo hunting observations were published in the 80s and 90s. Second, their hunting behavior could not be more different quantitatively or qualitatively. In a single hunt the Ngogo community killed more monkeys than bonobos were observed to kill in seven years of research at Lui Kitale. Bonobo hunting is extremely rare, even rarer if targeted at monkeys, and has been observed to be led by females not males. There is also no evidence for male impact hunting in bonobos as observed in East African chimpanzees, or the extinction of an entire monkey population due to bonobo hunting as observed at Ngogo, or male bonobos collaborating to hunt down rival males from other groups. Data on the proportion of calories bonobos obtain from hunting is miniscule compared to that observed in chimpanzees. The false equivalences throughout the intro are like saying “both bonobos and chimpanzees use tools in the wild and captivity” to suggest they are the same – even though wild bonobos have never been observed to use tools for extractive foraging across all field sites and decades of work. It is easy to obscure robust differences without giving the literature its due.

The authors currently characterize wild and captive work on chimpanzees and bonobos as if it is confusing and inconsistent. Dyadic experimental tests of tolerance in both species show that bonobos co-feed more readily than chimpanzees. Dyadic tests are especially powerful b/c they remove the influence of dependent rank via kin and friends as well as sex and age effects. The “confusion” enters when group tests of co-feeding are included with all these

variables uncontrolled. Perhaps they have more validity for revealing group level tolerance – for example even chimpanzees have shown tremendous tolerance in group cooperation settings (Suchak et al, 2016 PNAS), but within dyadic tests of co-feeding I am not aware of inconsistency (although see MacLean & Hare, 2013 J Comp Psych for nonfeeding dyadic tolerance measures).

Previous Replications: Both the species difference in dyadic feeding tolerance and cooperation have already replicated in actual faithful methodological replications but the authors again fail to outline this in detail. Experiment 1 of Hare et al 2007 (conducted at Ngamba (*P. swainfurthi*) and Lola ya Bonobo) was replicated in method and results by Wobber et al 2010a Current Bio – a paper the authors seem unaware of. Wobber et al, 2010a extended Exp 1 from Hare et al 2007 by again showing that bonobos are more food tolerant than chimpanzees. It used a large sample (24 bonobos and 30 chimpanzees *P. troglodytes*) that included a new set of subjects not tested in Hare et al 2007 (100% new chimpanzees of different subspecies at Tchimpounga sanctuary and ~50% of the bonobos from Lola ya Bonobo). The relatively large and sex- age balanced sample had enough resolution to see the species difference in the development of feeding intolerance in chimpanzees but not bonobos. Wobber et al 2010b PNAS then showed from these same sharing tests the species difference in physiological response that went along with the replicated species differences in sharing/cofeeding, sociosexual behavior and play. In terms of cooperation, again Exp 1 and 2 of Hare et al 2007 demonstrate two times (with ~ yr b/w each experiment) bonobos cooperate at similar levels with chimpanzees with sharable food amounts (not tested in the authors study). Finally, Hare et al, 2007 showed in experiment 3 round A that the bonobos outperformed the chimpanzees with monopolizable amounts of food and then replicate this finding in the round B. So - everything the authors claim has not been replicated previously has already been faithfully replicated twice (this also includes socio-sexual and play measures). To be fair to this work it should be acknowledged and explained in detail.

Bonobo constraints: Other large comparisons between dozens of chimpanzees and bonobos have revealed differences in social inhibition (Wobber et al, 2010a Curr Bio) and a willingness to share human artifacts (Krupenye et al 2018 Proc Royal Society). To successfully compare the impact of tolerance on the cooperative ability of the two species a set up needed to allow subjects to reach out directly for the cooperation rope. This effectively took away the inhibition constraint since a subject could immediately move forward and easily grab the rope w/ no need for dexterity or waiting on the partner. The design of the current experiment requires the bonobos / chimpanzees to put their fingers through the mesh to obtain food and ropes – requiring much more patience/persistence – psychological traits that likely differ b/w the two species (Rosati et al, 2007; Curr Bio). The Hirata string pulling tasks was also abandoned for use w/ bonobos after Hare et al 2007 for good reason – bonobos, unlike chimpanzees, are unwilling to return human artifacts like ropes or toys to experimenters (see Krupenye et al, 2018). The door opening paradigm was more effective (Hare & Kwetuenda, 2010, Tan and Hare, 2013; Tan et al, 2019) because the peg to open the door could be secured (instead of the bonobos trying to steal the rope all the time). The door opening work supports Hare et al 2007 showing types of food sharing and tolerance that does not exist in chimpanzees (i.e. especially with strangers). This series of papers replicates with a new paradigm the relationship between high feeding

tolerance in bonobos and cooperation (releasing another bonobo to co-feed). It directly replicates the hypothesized relationship initially established in Hare et al, 2007. Citing this work and its implications should only strengthen the current paper not distract as the authors suggest.

Major method differences: The authors have conducted a different but related experiment. Table 1 list almost 20 differences that can be gleaned from the methods sections. Some of these are really major differences by any estimation (half the bonobo sample of Hare et al, need for training, other constraints detailed above, etc). Some are more interesting / present opportunity (i.e. this may be first dyadic tolerance test of *P. verus* which is thought to show more bonobo-like gregariousness and feeding tolerance in the wild than other chimpanzee subspecies; also Hare et al 2007 focused on juveniles while this sample is adult). Really the only thing that is replicated is species and rope pulling. The paper can only be characterized as a conceptual replication as a result. An actual replication – similar to efforts in the human literature require: increasing sample to have more power not less and following procedures and designs exactly as described. This work, I am sorry to say, does not meet those very basic requirements.

Misrepresentative figure: The figure does not accurately represent the physical arrangement in the sleeping rooms at WKPRC - unless there have been major renovations. The mesh panels are much smaller and mesh does not go to the floor. Scale should be added and the actual dimensions of mesh / support beams / plexiglass represented.

Kaigaishi, Y., Nakamichi, M., & Yamada, K. (2019). High but not low tolerance populations of Japanese macaques solve a novel cooperative task. *Primates*, 60(5), 421-430.

Seed, A. M., Clayton, N. S., & Emery, N. J. (2008). Cooperative problem solving in rooks (*Corvus frugilegus*). *Proceedings of the Royal Society B: Biological Sciences*, 275(1641), 1421-1429.

Massen, J. J., Ritter, C., & Bugnyar, T. (2015). Tolerance and reward equity predict cooperation in ravens (*Corvus corax*). *Scientific reports*, 5(1), 1-11.

Martin, J. S., Koski, S. E., Bugnyar, T., Jaeggi, A. V., & Massen, J. J. (2021). Prosociality, social tolerance and partner choice facilitate mutually beneficial cooperation in common marmosets, *Callithrix jacchus*. *Animal Behaviour*, 173, 115-136.

Table 1. Some of the main differences in the methodology between the target experiment and the author’s current experiment. Apologies if I have misunderstood any details but I think the point will remain – this is a different method.

	Hare et al, 2007	Current Experiment
Subjects		
Subject pool	Subjects self-select to participate from population of ~50 bonobos and 39 chimpanzees (subspecies P. swainfurthi)	All available subjects trained for testing; trained to separate and participate (subspecies P. verus)
Experimental history	Bonobos experimentally naïve; chimpanzees only previously tested in exact same pulling task: Melis et al (2006a;b) and Melis et al 2009).	Both species daily testing for almost two decades on hundreds of training and cognitive tasks – including dozens of cooperation tasks.
Environment	Relatively large mix age – sex groups (20-40 individuals); free range 5-40 hectares primary tropical forest in range country representing microcosm of wild habitat. Possible intergroup interaction (+ & -) through fence.	Small bonobo group size of foraging party (~10) both species live in small indoor enclosure 6+ months per year and only have access to <1 hectare and artificial climbing structures (high tension in months of crowding). No intergroup encounters for bonobos (see Wobber et al 2011, PLoS One for evidence of higher aberrant behavior rate in Leipzig than sanctuaries as result).
Feeding regime	Ad libitum browse always available to subjects during day (similar to wild). No regurgitation observed.	Food only available during meals and experiments – very little low-quality browse available = > aberrant regurgitation
Mean Age	Exp 3: Bonobo: Juvenile (7.9 yrs) Chimpanzee: Juvenile (7.5 years)	Bonobo: Adult (>18 yrs) Chimpanzee: Adult (>25 yrs)
Dyad Composition		
Kinship controlled w/in Dyads	Yes - no kin tested together	No - some pairs related
Age matched Dyads	Yes – within and across species (all subjects in dyad 0-3 yrs apart in age)	No – dyads of different ages repaired
Sex balanced Dyads	Yes – within and across species MM FF MF basically equal	No – dyads of difference sexes repaired throughout – although oddly no F-F pairings (suggests some serious group instability or caretaking / management issues).
Dyads created through mixing subjects	No - Pairs of dyads tested throughout each 6 trial experimental session and unit of analysis	Yes – dyads created by repairing of subjects within the same experiment – raising questions of independence
Set up		

Cooperation platforms	3.4 meters wide (feeding dishes 2.7 cm apart)	2.7 meters wide (feeding dishes each 27cm long making them 2.16 cm apart) – subjects required to work and feed in closer proximity.
Test location	Open air rooms (friends - allies potentially visible easy to hear)	Closed concrete walls (cannot see/hear friends / allies). Much smaller test rooms.
Experimenters move subjects	Yes – this allowed reduced subject anxiety using only positive reinforcement during test (critical for reactive bonobos).	No – in LEJ only caretakers move subjects and bonobos in particular have long history of anxiety toward their caretakers. Restrictions on their ability to test mix sex pairs as a result.
Design		
Individual Rope Pulling Training	No - they spontaneously cooperated for large amounts of food in Exp 1 and 2.	Yes - 5 individual test sessions to train subjects to pull rope / tray.
Cooperation Pretest	Yes - all pairs of both species demonstrate success cooperating for highly sharable food before critical test (6 trials)	No – no introduction to dyadic cooperation task with highly sharable food before critical test
Cooperation Warm up	3 trials success criteria before experiment	No warm-up
“clean dyads” in test trials (exp 3 A & B)	Yes - 6 test trials in session A and B = 12 trials total per subject / dyad. 6 clean dyads per species in A & B.	No - With only 7 testable bonobos it would only be possible to test 3 clean dyads – so repairing and repeated testing of same subjects necessary.
Procedure		
Experimenter calls subjects name and throws rope	Yes – this allowed to control for species differences in self-control by getting rope ends to both subjects simultaneously.	No – subjects must inhibit pulling until partner can get access to rope that human threads through small mesh hole - does not control for species differences in social inhibition.
Test Food	Introduced to pulling task with large amounts of food (.5 kg of banana each trial) then key tolerance test with small amount of food introduced	Only tested with small amounts of food

Appendix E

Response to Reviewer 1

I remain with major reservations regarding this paper, but before I detail those a couple of points. First, I think this study is ultimately publishable and represents a contribution to the literature with different framing. I also applaud the desire to replicate even though I do not think this effort can be characterized that way (faithful replication in this case will likely require sanctuary work). Second, I obviously have only seen the introduction and methods and have no idea what the authors have found / will find since I have not seen the results. But my concerns apply regardless of the findings. All my comments are simply about the framing and comparability of the methods. After carefully reading the response and revision it is clearer this study should not be characterized as a methodological replication of previous methods. I certainly think it can be framed as a conceptual replication (i.e. designed to test the same ideas) but too many critical variables are just too different (see Table 1). I hope the authors take the thoughts below and the time put into it as a good faith effort to help them and everyone get the most out of the work they are trying to publish. I will be curious to see the findings with these new methods regardless of what they are.

Dear reviewer, we are very thankful for your suggestions and sincerely appreciate the time and effort you have been willing to put into improving the work. We follow the suggestion that our study is a *conceptual* replication, and present our study as such. Please find below our specific responses to your comments.

Suboptimal: I say suboptimal because the authors of the original Leipzig team from Hare et al 2007 decided **not** to conduct this work at the WKPRC even though the same exact set up and animals were available to them over a decade earlier. The original team realized conducting this kind of work at the WKPRC required far too many design compromises that the current authors have again faced (small samples ($n < 10$), too little flexibility in experimental set up (no reaching), fear of caretakers by the bonobos, and required dietary restrictions (preventing use of enough food to allow for robust cofeeding/sharing). And this realization was earned. Melis et al 2006 (*Anim Behaviour*) conducted and published an initial pilot study in this same facility with some of the same subjects the authors are working with. Due to the constraints above experienced during that pilot the team largely abandoned the zoo for this type of

work. That paper and all that teams papers that followed on feeding tolerance and instrumental cooperation were conducted in African sanctuaries because they allowed these constraints to be escaped. Another way to say this is – Why would the previous team go to all the trouble to work in Uganda, Congo-Brazza and D.R.C. if they could just do the same thing in LEJ where they already lived? The LEJ building is well suited for lots of studies but this type is not one of them.

→ We are partially sympathetic to the reviewer’s concerns regarding the limitations of the used test-location, but we do not agree that these limitations preclude the possibility to learn about social tolerance and cooperation of bonobos and chimpanzees. Most importantly, we have subjected both species to the same test-setup, which, in our view, is a big advantage of the chosen test-location (this same location was also used for a subset of bonobos (exp 1 & 2 & 3) and chimpanzees (exp 3) in Hare et al., 2007). The impossibilities of “reaching” and providing *large* amounts of food are indeed differences between the original study and ours – and we now see that our efforts are therefore best described as a *conceptual* replication (see our changed title). In our view, though, these differences do not deem our study findings invalid. Both species have experienced the same conditions (e.g., no reaching and smaller amounts of food) which could in principle be viewed as an improvement over the original study in which the authors tested bonobos and chimpanzees in different test-locations. Moreover, we would hope that the concluded behavioural tendencies in the original study should be robust against small alterations of the test setup details. We address other points raised here below.

Poor Scholarship: Unfortunately, their scholarship is not there yet. A balanced introduction would present evidence for and against species differences in tolerance and cooperation between the two species and would include both captive and wild observations. The authors suggest this is not necessary because they are focused on Hare et al 2007 and studies that involve direct comparisons – but they then skip over Hare & Kwetuenda 2010 Curr Bio; Tan & Hare, 2013 PLoS One; Tan et al 2019 Scientific Reports that present a type of tolerance and sharing with non-group members in bonobos that 1) has never observed in chimpanzees but has since been observed in wild bonobos (e.g. Fruth & Hohman, 2018, Human Nature; also see

Yamamoto & Furiuchi 2017) and 2) could not even be tested in chimps due to safety issues. Based on their revision, it also seems the authors are asking the audience to ignore all the other work that built toward Hare et al 2007 and then followed it in support (i.e. Melis et al, 2006 a;b; Hare et al, 2007, Wobber et al, 2010 a;b). They justify this by saying they are only attempting to replicate Hare et al 2007 and only cite studies that provide quantitative comparisons. I hope the authors will reconsider in future versions. Here is why: A series of studies were completed at Ngamba island (Melis et al, 2006 a;b; 2009) before Hare et al 2007 was piloted.

- We thank the reviewer for their thorough evaluation of our written scholarship. We have not intended to convey the message to the readers that they should ignore literature. Instead, we aimed to highlight the extra relevance of studies in which both species were tested in the very same setup, as we know that different studies unavoidably come with different parameters. Although we did dedicate a separate paragraph to review both wild and captive research and cited all papers referred to as missing here (except for two), we have now elaborated on both paragraphs including the additional two studies mentioned by the reviewer. Therefore, we now include research on bonobos sharing with outgroup members in the wild and captivity. We agree that this information could be useful to further understand bonobo behaviour especially in comparison to chimpanzees' behaviour. Please see below the added text pieces:
- Line 44-49: “In stark contrast to rather aggressive and lethal intergroup encounters in chimpanzees, bonobos even share food with members of neighbouring communities (Samuni et al., 2020; Fruth & Hohmann, 2018). Thus, in the wild, chimpanzees and bonobos both engage in collective behaviours, however, they differ in that chimpanzees exhibit more varied behaviours in aggressive and competitive contexts, while bonobos share food with individuals outside their own group, engage in less collaborative hunting, and do not engage in lethal territorial defence.”
 - Line 52-57: “Based on such research we know that chimpanzees can instrumentally help humans and conspecifics (Yamamoto et al. 2009, 2012; Melis et al. 2011; Warneken et al., 2007), provide benefits to a conspecific (Horner et al., 2011), and understand the role of their partner

during cooperation (Melis et al., 2006b). Captive bonobos have been observed to instrumentally help conspecifics (Krupenye & Hare, 2018), share food with other group members (Hare & Kwetuenda, 2010), and, corroborating research from the wild, also help obtain and share food with outgroup members (Tan et al., 2017; Tan & Hare, 2013).”

→ As for the “the other work that built toward Hare et al 2007 and then followed it in support (i.e. Melis et al, 2006 a;b; Hare et al, 2007, Wobber et al, 2010 a;b)”, specifically, we note that Melis et al. 2006a and Hare et al. 2007 used the same data for the chimpanzees and only acquired new data on the bonobos in Hare et al. 2007. This means that there is no published work that built toward Hare et al 2007 experiment 1 and 2. If indeed pilot studies were conducted, neither Melis et al., nor Hare et al. conveyed this in their publications. Finally, the work that “followed it in support” (i.e., Wobber et al. 2010ab) is discussed in our manuscript (see our more detailed comments below).

Melis, A. P., Hare, B., & Tomasello, M. (2006). Engineering cooperation in chimpanzees: tolerance constraints on cooperation. *Animal Behaviour*, 72(2), 275-286.

Melis, A. P., Hare, B., & Tomasello, M. (2006). Chimpanzees recruit the best collaborators. *Science*, 311(5765), 1297-1300.

Wobber, V., Wrangham, R., & Hare, B. (2010). Bonobos exhibit delayed development of social behavior and cognition relative to chimpanzees. *Current Biology*, 20(3), 226-230.

Wobber, V., Hare, B., Maboto, J., Lipson, S., Wrangham, R., & Ellison, P. T. (2010). Differential changes in steroid hormones before competition in bonobos and chimpanzees. *Proceedings of the National Academy of Sciences*, 107(28), 12457-12462.

These initial studies provided the initial link b/w tolerance and cooperation using a sample of 36 chimpanzees (i.e. making it clear robust social groups (N=20- 30) were needed to have enough power; this relationship has been replicated in a host of other species – see references for examples in rooks, ravens, macaques and marmosets). They also taught us about the amount of food that was needed to see robust and spontaneous cooperation using the 3 conditions – typically several **entire** bananas cut up (~500g) in multiple dishes (i.e. there is significantly less tolerance w/ less food).

→ We have not been able to find any statistical analyses and published information regarding these aspects in any of the stated papers. All we have been able to do is work within the contemporary diet protocols (i.e., food amounts in apes are restricted nowadays, more so than 15 years ago) and, most importantly, keep the amounts exactly equal for both species. Furthermore, we did replicate the food amount of the “most critical” condition of the cooperation test and adapted the remaining food amounts based on this, allowing for comparability across conditions and experiments.

We also learned that performance in the causal understanding pre-test in Melis et al 2006 was not related to cooperation success (i.e. making rope training of subjects in future work unnecessary).

→ In our study, we wished to ascertain that all the tested apes were equally proficient at the contingencies of the rope-pulling task. Similar to what was decided in Melis et al. 2009 (p. 382: "two adult females did not participate in Melis et al., 2006a and, thus, were given similar experience as the rest of the subjects before starting the present experiment."), we wanted to make sure that all individuals in our study had the same basic knowledge of the apparatus, independent of their previous knowledge and especially since all apes were not tested for nearly two years prior to the start of our study. If anything, the solitary training would have no effect on cooperation levels, based on Melis et al., 2006a.

Melis, A. P., Hare, B., & Tomasello, M. (2009). Chimpanzees coordinate in a negotiation game. *Evolution and Human Behavior*, 30(6), 381-392.

We also discovered the importance of open walls that the apes could reach through (reducing the impact of individual differences in self-control in waiting for others to grab the rope). Entire buildings in Uganda, DRC and Congo Brazzaville were designed and renovated to allow for this type of work (opposed to just suffering with the fixed design at WKPRC). Based on all this experience we designed and conducted

Hare et al 2007. The study was built on many others and that phylogeny of a method may matter if the intent is to robustly replicate.

→ The information provided by the respective papers (Melis/Hare and colleagues) shows that the chimpanzee data of all experiments other than the third in Hare et al. 2007 were used from the initial study of this series. If previous studies were performed other than experiment 1 of Melis et al. 2006a, then they were not reported. Based on published work, no information is conveyed about the effect of open walls that apes can reach through. Moreover, Hare et al. 2007 did test individuals at WKPRC without discussing possible differences. We can only use the information provided to us and try our best to replicate and discuss the current study based on that information.

Some of the review borders on caricature. Hare et al 2007 was designed to test the hunting hypothesis that predicted more cooperation in chimpanzees than bonobos when working to obtain food since chimpanzees cooperatively hunt much more than bonobos. The initial two experiments show that the bonobos and chimpanzees perform similarly even though the chimpanzees had more experience with the experimental task (rejecting the hunting hypothesis). However, the authors fail to mention this framing.

→ The framing of the hunting hypothesis is not discussed. Our work did not set out to test new or old hypotheses but to understand inconsistencies in published work.

Instead, the authors appear to cite recent examples of hunting in bonobos as if that means the hunting behavior b/w the two species is recently discovered and similar. First, bonobo hunting observations were published in the 80s and 90s. Second, their hunting behavior could not be more different quantitatively or qualitatively. In a single hunt the Ngogo community killed more monkeys than bonobos were observed to kill in seven years of research at Lui Kitale. Bonobo hunting is extremely rare, even rarer if targeted at monkeys, and has been observed to be led by females not males. There is also no evidence for male impact hunting in bonobos as observed in East African chimpanzees, or the extinction of an entire monkey population due to bonobo hunting

as observed at Ngogo, or male bonobos collaborating to hunt down rival males from other groups. Data on the proportion of calories bonobos obtain from hunting is miniscule compared to that observed in chimpanzees. The false equivalences throughout the intro are like saying “both bonobos and chimpanzees use tools in the wild and captivity” to suggest they are the same – even though wild bonobos have never been observed to use tools for extractive foraging across all field sites and decades of work. It is easy to obscure robust differences without giving the literature its due.

→ We do not cite papers that involve solitary hunting behaviours and also never stated that the two species would hunt to the same degree. However, we acknowledge that this might not have been conveyed clearly enough in our introduction and so we have adjusted it accordingly:

- Line 38-41: “Even though wild bonobos are rarely seen to perform actions such as boundary patrols and collective territorial defence (Furuichi, 2011; Hohmann & Fruth, 2002; Wilson et al., 2014), there is a growing body of evidence that they are capable of group hunting, albeit do so much more infrequent than chimpanzees (Surbeck & Hohmann, 2008; Samuni et al., 2020; Wakefield et al., 2019).”

The authors currently characterize wild and captive work on chimpanzees and bonobos as if it is confusing and inconsistent. Dyadic experimental tests of tolerance in both species show that bonobos co-feed more readily than chimpanzees. Dyadic tests are especially powerful b/c they remove the influence of dependent rank via kin and friends as well as sex and age effects. The “confusion” enters when group tests of co-feeding are included with all these variables uncontrolled. Perhaps they have more validity for revealing group level tolerance – for example even chimpanzees have shown tremendous tolerance in group cooperation settings (Suchak et al, 2016 PNAS), but within dyadic tests of co-feeding I am not aware of inconsistency (although see MacLean & Hare, 2013 J Comp Psych for nonfeeding dyadic tolerance measures).

→ Thank you for pointing out the potential difference between dyadic versus group tests. To our knowledge, the only dyadic experimental tests that compare co-

feeding tolerance levels of both species is Hare et al 2007 (finding bonobos to be more tolerant) and Tagliatalata et al 2020 (finding chimpanzees to be more tolerant). We included group settings that compare the tolerance levels of both species to give a more complete picture and ultimately understand what influences tolerance and what are general species patterns. We adapted the introduction accordingly and thank you again for the input:

- Line 73-75: “Importantly, however, all except one study (Tagliatalata et al., 2020) that found chimpanzees to be more prosocial or tolerant than bonobos employed group instead of dyadic tests. Thus, it is unclear to what extent group factors influence such differences.”

→ In our view, it is not possible to directly compare the results of two studies that use vastly different designs and measurements, as such methodological differences can influence responses (e.g. Tan et al. 2015; House et al. 2014). Therefore, to study species differences most validly, we need to look at those studies that actually compare the two species with the same paradigm. The “one-species studies” can still provide insightful information into the respective species alone, which we (now) also provide.

Tan, J., Kwtuenda, S., & Hare, B. (2015). Preference or paradigm? Bonobos show no evidence of other-regard in the standard prosocial choice task. In *Bonobo Cognition and Behaviour* (pp. 275-298). Brill.

House, B. R., Silk, J. B., Lambeth, S. P., & Schapiro, S. J. (2014). Task design influences prosociality in captive chimpanzees (*Pan troglodytes*). *PLoS one*, 9(9), e103422.

Previous Replications: Both the species difference in dyadic feeding tolerance and cooperation have already replicated in actual faithful methodological replications but the authors again fail to outline this in detail. Experiment 1 of Hare et al 2007 (conducted at Ngamba (*P. swainfurthi*) and Lola ya Bonobo) was replicated in method and results by Wobber et al 2010a Current Bio – a paper the authors seem unaware of. Wobber et al, 2010a extended Exp 1 from Hare et al 2007 by again showing that bonobos are more food tolerant than chimpanzees. It used a large sample (24 bonobos and 30 chimpanzees *P. troglodytes*) that included a new set of subjects not tested in Hare et al 2007 (100% new chimpanzees of different subspecies at Tchimpounga

sanctuary and ~50% of the bonobos from Lola ya Bonobo). The relatively large and sex- age balanced sample had enough resolution to see the species difference in the development of feeding intolerance in chimpanzees but not bonobos. Wobber et al 2010b PNAS then showed from these same sharing tests the species difference in physiological response that went along with the replicated species differences in sharing/cofeeding, sociosexual behavior and play.

→ Thank you. We have indeed been aware of these two studies, but given that they themselves indicate to *not* be faithful replications of Hare et al. 2007, we did not want to put these studies forth as previous replications. For instance, Wobber et al. (2010a) placed food inside (instead of outside) the testing rooms, spaced out food instead of clumping it in the “clumped” condition, and changed the number of trials. That said, we have now added the results of this study to our introduction. With respect to the other potential previous replication (Wobber et al., 2010b), they do not report any results regarding the difference in co-feeding scores between the two species. As indicated previously, however, this study is valuable in its own right and we use it to discuss our results.

In terms of cooperation, again Exp 1 and 2 of Hare et al 2007 demonstrate two times (with ~ yr b/w each experiment) bonobos cooperate at similar levels with chimpanzees with sharable food amounts (not tested in the authors study).

→ Thank you, yet Hare et al. 2007 Exp 1 only tests the co-feeding conditions, while Exp 2 only tests the dispersed cooperation condition. Here, we believe the reviewer means Exp 2 and Exp 3, which indeed both test for cooperation with ~ 1 yr between experiments. This is indeed the very first replication of the findings that bonobos cooperate at similar levels to chimpanzees with sharable food amounts and we refer to this (see text piece copied below). Our study, however, extends beyond this replication by also testing co-feeding patterns and cooperation for clumped resources:

- Line 152-159: “In “Experiment 2”, the original authors conducted the cooperation test with shareable food distribution only and replicated both the methods and results in “Experiment 3” while additionally incorporating a condition with monopolizable food distribution. We

therefore did not conduct their first cooperation test (i.e. “Experiment 2”), but replicated Experiment 3 that encompassed both conditions. Hare2007 used a within-subject design for comparing the co-feeding (“Experiment 1”) and first cooperation test (“Experiment 2”), but a between-subject design for the second cooperation test (“Experiment 3”). We replicated the within-subject design, which enabled us to also compare co-feeding and cooperation with monopolizable food distribution.”

Finally, Hare et al, 2007 showed in experiment 3 round A that the bonobos outperformed the chimpanzees with monopolizable amounts of food and then replicate this finding in the round B. So - everything the authors claim has not been replicated previously has already been faithfully replicated twice (this also includes sociosexual and play measures). To be fair to this work it should be acknowledged and explained in detail.

- ➔ In Round 1 of experiment 3, opposite-sex dyads were tested and subsequently in Round 2, the same subjects were re-combined into same-sex dyads. We do not agree with the reviewer’s proposition that creating new dyads within the same experiment should be considered a replication of results.
- ➔ Furthermore, in Round 1 of their study, dyads were tested in both the *dispersed* and *clumped* condition. In Round 2, the new constellation of dyads was tested only in the *clumped* condition. Instead of also testing the *dispersed* condition with the new dyads, the researchers used the previous scores of both partners and averaged them. Thus, if in Round 1 dyad AB cooperated on 6 trials and dyad CD on 0 trials, then in Round 2 the new dyad BC would receive a cooperation score of 3, ignoring the fact that averaging like this will most likely not lead to a true score. To avoid this possibly influential data-handling technique, we decided to replicate both conditions with each dyad while using comparable food amounts between conditions.

Bonobo constraints: Other large comparisons between dozens of chimpanzees and bonobos have revealed differences in social inhibition (Wobber et al, 2010a Curr Bio) and a willingness to share human artifacts (Krupenye et al 2018 Proc Royal Society). To successfully compare the impact of tolerance on the cooperative ability of the two species a set up needed to allow subjects to reach out directly for the cooperation rope. This effectively took away the inhibition constraint since a subject could immediately move forward and easily grab the rope w/ no need for dexterity or waiting on the partner. The design of the current experiment requires the bonobos / chimpanzees to put their fingers through the mesh to obtain food and ropes – requiring much more patience/persistence – psychological traits that likely differ b/w the two species (Rosati et al, 2007; Curr Bio).

→ We believe this remark is based on a misreading of our methods. As we explain in our Methods, both experimenters placed the rope into the cage simultaneously or after 20 seconds of having their names called. Thus, we used the exact same procedure as in Hare et al. with the minor difference that instead of *throwing* the rope into the hands of the apes, we *placed* the rope into the hands of the apes. In case the individuals would not arrive upon their name being called, the rope remained immediately available to them as in Hare et al. 2007. Therefore, at no point needed the apes to grab through the narrower mesh to retrieve the rope, making the argument of bonobos' inhibition constraint invalid.

The Hirata string pulling tasks was also abandoned for use w/ bonobos after Hare et al 2007 for good reason – bonobos, unlike chimpanzees, are unwilling to return human artifacts like ropes or toys to experimenters (see Krupenye et al, 2018).

→ We never experienced bonobos to be unwilling to return human artifacts, neither in this study, nor in Nolte et al. 2021 (using the same bonobo population) where they had to return up to three objects after every trial. More importantly, our study was never hindered because bonobos wanted to play with the rope for long periods or would not give it back. We have now added this information to the manuscript:

- Line 180-184: “Different to Hare2007, bonobos did not play with the rope, except for one session during which one female took the rope three

times to play with it after her partner refused to cooperate. She gave back the rope twice after one and once after three minutes, and we could continue testing the remaining trials. Therefore, testing was never hindered by play behaviour involving the rope.”

The door opening paradigm was more effective (Hare & Kwetuenda, 2010, Tan and Hare, 2013; Tan et al, 2019) because the peg to open the door could be secured (instead of the bonobos trying to steal the rope all the time). The door opening work supports Hare et al 2007 showing types of food sharing and tolerance that does not exist in chimpanzees (i.e. especially with strangers). This series of papers replicates with a new paradigm the relationship between high feeding tolerance in bonobos and cooperation (releasing another bonobo to co-feed). It directly replicates the hypothesized relationship initially established in Hare et al, 2007. Citing this work and its implications should only strengthen the current paper not distract as the authors suggest.

→ We do acknowledge that other paradigms are also interesting to shed light on the respective research question. As conveyed above, we have now included information on species differences in sharing with strangers in our manuscript (Lines 44-46 and 56-57).

Major method differences: The authors have conducted a different but related experiment. Table 1 list almost 20 differences that can be gleaned from the methods sections. Some of these are really major differences by any estimation (half the bonobo sample of Hare et al, need for training, other constraints detailed above, etc). Some are more interesting / resent opportunity (i.e. this may be first dyadic tolerance test of *P. verus* which is thought to show more bonobo-like gregariousness and feeding tolerance in the wild than other chimpanzee subspecies; also Hare et al 2007 focused on juveniles while this sample is adult). Really the only thing that is replicated is species and rope pulling. The paper can only be characterized as a conceptual replication as a result. An actual replication – similar to efforts in the human literature require: increasing sample to have more power not less and

following procedures and designs exactly as described. This work, I am sorry to say, does not meet those very basic requirements.

→ Thank you for drafting this table and giving background information. In light of the reviewer's remarks, we agree that our study can be better represented as a *conceptual* replication and have adapted our title accordingly. Moreover, we have now incorporated a Table with study differences in our Supplementary Materials (i.e., we chose for the ESM given the size and dissections into levels of relevance, see below). We adjusted the table provided by the reviewer to reflect more accurately our study details, and divided the differences into:

- Methodological differences (Table 1a), 4 arguments
- Population and Sampling differences (Table 1b), 8 arguments

Whereas the differences depicted in Table 1a render our study a *conceptual* rather than a faithful replication, the differences depicted in Table 1b will aid to our understanding regarding the generalizability of the original findings. In our view, **four differences (Table 1a) are valid concerns towards a faithful replication**. We clearly state them in the manuscript. For these reasons, we agree with a framing in terms of a *conceptual* replication. Including the streamlined information drafted in Table 1ab to our ESM, we allow the readers to judge the level of replication for themselves. In addition to referring to the respective table per discussed argument, we have now included:

- Line 185-189: "All distinctions in methodology between the current and original study are outlined in the supplementary material (Table 2a), based on which the current study represents a conceptual rather than faithful replication of Hare2007. Furthermore, we provide details on the population and sampling differences between the two studies that can aid to our understanding of the generalizability of species differences (ESM, Table 2b)."
- Lastly, we have constructed a Table 1c (8 arguments) specifically for the rebuttal of reviewer #1's concerns. In this Table 1c, we have listed what the suggested as differences that we cannot verify based on published work. Hence, we choose not to incorporate Table 1c in our manuscript.

Explanation for reading the following tables: We have contrasted the original Hare et al., 2007 study with our study in the columns. Per suggested difference, we have 1) provided reviewer #1's description (in grey), 2) our suggested streamlined version (in green), and 3) our comments regarding the respective difference (in white).

Table 1a. Methodological differences between the original and the current study.

		Hare et al., 2007	Current study
Cooperation platforms	Reviewer	Tray length is 3.4 meters wide (feeding dishes 2.7 cm apart)	2.7 meters wide (feeding dishes each 27cm long making them 2.16 cm apart) – subjects required to work and feed in closer proximity.
	Streamlined for publication	Tray length is 3.4 meters wide (feeding dishes 2.7 m apart). ⁽¹⁾	Tray length is 2.7 meters wide (feeding dishes 2.2 m apart) – subjects required to co-feed and cooperate in closer proximity.
	Comments	We wonder how the data at WKPRC were obtained since no indication is given that methods were adapted. We did not mention this out of respect to the original authors.	
Food amount	Reviewer	Introduced to pulling task with large amounts of food (.5 kg of banana each trial) then key tolerance test with small amount of food introduced	Only tested with small amounts of food
	Streamlined for publication	Co-feeding task tested with large amounts of food (0.5 kg of banana each trial). Cooperation task tested with large amounts of food in the dispersed condition and small amounts of food in the clumped condition (0.5 kg of banana each trial versus four 1.5cm pieces).	Only tested with small amounts of food (four 1.5cm pieces of banana each trial, replicating what was originally used in the clumped condition) in both experiments .
	Comments	Changing the amount of food mid-study could induce noise. We wonder why the clumped divisible condition was tested with 0.5kg food in the co-feeding task but with 1.5cm	We faithfully replicated the “key” (according to the reviewer) condition clumped divisible of the cooperation task. To enable us to actually compare results across

pieces in the cooperation task?
No reason is provided in the original publication.

conditions, we used the same amount of food in all other conditions while adhering to the logic of dividing the food as in Hare et al. 2007

Cooperation "warm up"	Reviewer	3 trials success criteria before experiment	No warm-up
	Streamlined for publication	“Three-trials-success” criteria of cooperating in condition dispersed divisible (up to 13 trials per dyad) provided to bonobos at both facilities and chimpanzees at the WKPRC before conducting cooperation experiment that included condition dispersed divisible and clumped (i.e. Exp.3).	No dyadic “warm-up”, only solitary pre-test trials for rope-pulling understanding provided to both species. Only partners used in our study that understood the mechanism (six consecutive successful trials within the same session during which the rope ends were spaced by a distance of at least 1m).
	Comments	Hare et al. Suppl. S2: „Finally, the bonobos (and Leipzig chimpanzees) were given warm-up trials because, unlike the Ngamba chimpanzees, they were largely naive to the task. All of the bonobo pairs were given warm-up cooperation trials in which both trays were baited with sharable food (dispersed divisible) until they were able to successfully retrieve the food for three trials in a row. Two pairs of bonobos made no mistakes in their first session, so this session was simply used as their first test session, whereas the other pairs were all proficient within 13 trials.“ We did not want to highlight this, but the authors basically trained their specific constellation of bonobo dyads and three WKPRC chimp dyads to perform well in the dispersed division task before starting the experiment 3.	For obvious reasons we did not perform such “warm up” cooperation trials (i.e., in our view, this could bias the results of the study, especially given that the majority of bonobos were “warmed-up”, but the majority of chimpanzees was not), but only ensured that each ape we tested in the cooperation task knew how the mechanism of the apparatus worked via solitary training.

Accessibility of food	Reviewer	Added by us	Added by us
	Streamlined for publication	Width of mesh allowed apes to reach for food by sticking their arm through the mesh. ⁽¹⁾	Width of mesh is narrower: Apes can reach for food by sticking their fingers through the mesh.

(1) No published information provided for WKPRC apes

Table 1b. Population and sampling differences between the original and the current study.

		Hare et al, 2007	Current study
Chimpanzee subjects	Reviewer	subspecies P. schweinfurthii	subspecies P. verus
	Streamlined for publication	Subspecies P. schweinfurthii at Ngamba and P. verus at WKPRC.	Subspecies P. verus and four females are verus-schweinfurthii hybrids.
Sample procedure	Reviewer	Subjects self-select to participate from population of ~50 bonobos and 39 chimpanzees	All available subjects trained for testing; trained to separate and participate
	Streamlined for publication	Exp. 1 & 2: Lola ya Bonobo : Bonobos selected from several populations of 12–20 individuals. WKPRC : Bonobos lived together in social group. Ngamba Island : Chimpanzees selected from one population of 39 individuals. Exp. 3: Most tolerant chimpanzee dyads selected of previous population and additionally three tolerant dyads at WKPRC selected. Bonobo pairs of previous population were selected to match the age and sex of the most tolerant chimpanzee pairs.	WKPRC : Both species sampled at same facility. Bonobos were selected from population of 12 individuals, and chimpanzees from population of 19 individuals. In both experiments, as many dyad combinations were sampled that were allowed by the current handling regime (see main manuscript for details). Data collection was immediately stopped if either partner showed signs of fear towards being released into the same room with the other (see main manuscript for details).
	Comments	No published information on self-selection provided in Hare et al. 2007 or Melis et al. 2006a.	

		Population sizes adapted based on Hare et al. 2007.	
Experimental history	Reviewer	Bonobos experimentally naïve; chimpanzees only previously tested in exact same pulling task: Melis et al (2006a;b) and Melis et al 2009).	Both species daily testing for almost two decades on hundreds of training and cognitive tasks – including dozens of cooperation tasks.
	Streamlined for publication	Bonobos and chimpanzees experimentally naïve in Exp. 1 (co-feeding task) and Exp. 2 (cooperation task, condition dispersed divisible).	Both species trained for participating in testing and being separated with experience across various cooperation tasks.
	Comments	The information provided by the reviewer is not correct (based on published information and outlined previously).	
Environment	Reviewer	Relatively large mix age – sex groups (20-40 individuals); free range 5-40 hectares primary tropical forest in range country representing microcosm of wild habitat. Possible intergroup interaction (+ & -) through fence.	Small bonobo group size of foraging party (~10) both species live in small indoor enclosure 6+ months per year and only have access to <1 hectare and artificial climbing structures (high tension in months of crowding). No intergroup encounters for bonobos (see Wobber et al 2011, PLoS One for evidence of higher aberrant behavior rate in Leipzig than sanctuaries as result).
	Streamlined for publication	Sanctuaries: 5-40 hectares of primary tropical forest. WKPRC: Bonobos have access to indoor and outdoor enclosure (2600 m2). No information provided on chimpanzees (however see “Current study”).	Both species live in indoor enclosure (430 m2 chimpanzees, 256 m2 bonobos) and have access to outdoor enclosure (4000 m2 chimpanzees, 2300 m2 bonobos) during warm months.
	Comments	No published information available in Hare et al. 2007 on possible intergroup interaction through fence.	
Mean age and range	Reviewer	Exp 3: Bonobo: Juvenile (7.9 yrs) Chimpanzee: Juvenile (7.5	Bonobo: Adult (>18 yrs) Chimpanzee: Adult (>25 yrs)

		years)	
	Streamlined for publication	Exp. 1 & 2: Bonobo: 9.6 years (range: 5 – 22); Chimpanzee: 11.7 years (range: 4 – 21)	Co-feeding experiment: Bonobo: 18.9 years (range 6.2 – 36.5) Chimpanzee: 25.9 years (range 3.2 – 43.5)
		Exp 3, Round 1: Bonobo: 7.9 years (range: 5 – 20) Chimpanzee: 7.5 years (range: 5 – 12)	Cooperation experiment (subsample based on passing solitary training): Bonobo: 22.5 years (range 13.6 – 36.5) Chimpanzee: 30.4 years (range 15.1 – 43.5)
			Age is incorporated in the statistical models.
Kinship	Reviewer	Yes - no kin tested together	No - some pairs related
	Streamlined for publication	Selection of only non-kin dyads.	Some dyads related; incorporated in the statistical models.
Test location	Reviewer	Open air rooms (friends - allies potentially visible easy to hear)	Closed concrete walls (cannot see/hear friends / allies). Much smaller test rooms.
	Streamlined for publication	15 m2 testing room and open space in sanctuaries, 9 m2 testing room at WKPRC.	7 m2 testing room for bonobos with available adjacent rooms of 6,3 m2 to left and of 11,3 m2 to right. 13,32 m2 testing room for chimps with available adjacent rooms of 6,8 m2 to left and right.
	Comments	No published information on whether	Friends and allies can be easily heard through the walls.
Dyad composition	Reviewer	No - Pairs of dyads tested throughout each 6 trial experimental session and unit of analysis	Yes – dyads created by repairing of subjects within the same experiment – raising questions of independence
	Streamlined for publication	In Exp. 3 (cooperation task) same partners used in two unique dyads.	To increase effective sample size, we tested apes in multiple dyads.

Comments	Information provided by the reviewer is not correct, so we adjusted it to be more specific.	Within subject design used (all pairs first tested in the co-feeding and subsequent cooperation task). Random effects of dyad identity, partner 1, and partner 2 and corresponding random slope effects used in model to ensure independence of results.
-----------------	---	--

The following table will not be part of our submission given the inaccurate or unknown information provided by reviewer #1.

Table 1c. Information not true based on publication or no published information available.

		Hare et al, 2007	Current study
Feeding regime	Reviewer	Ad libitum browse always available to subjects during day (similar to wild). No regurgitation observed.	Food only available during meals and experiments – very little low quality browse available = > aberrant regurgitation
	Comments	No published information available in Hare et al. 2007 or Melis et al. 2006a	
Age matched Dyads	Reviewer	Yes – within and across species (all subjects in dyad 0-3 yrs apart in age)	No – dyads of different ages paired.
	Comments	Several dyads included with huge age difference tested in exp. 1 and 2.	As stated above in Table 1b, the age of both partners is included in the analyses.
Sex balanced Dyads	Reviewer	Yes – within and across species MM/FF/MF dyads basically equally present	No – dyads of different sexes re-paired throughout – although oddly no F-F pairings (suggests some serious group instability or caretaking / management issues).
	Comments	Not true based on published information.	Several F-F pairings were included in both species. The effect of dyad sex estimated with statistical

“clean dyads” in test trials (exp 3 A & B)	Reviewer	Yes - 6 test trials in session A and B = 12 trials total per subject / dyad. 6 clean dyads per species in A & B.	analyses No - With only 7 testable bonobos it would only be possible to test 3 clean dyads – so re-pairing and repeated testing of same subjects necessary.
	Comments	Not true based on published information. See adapted information of this above (“Dyad composition”)	We ensured independence of results in our analyses (see above “Dyad composition”)
Experimenters move subjects	Reviewer	Yes – this allowed reduced subject anxiety using only positive reinforcement during test (critical for reactive bonobos).	No – in LEJ only caretakers move subjects and bonobos in particular have long history of anxiety toward their caretakers. Restrictions on their ability to test mix sex pairs as a result.
	Comments		Though we cannot speak for the experience of this research team 15+ years ago, but at the time of this study, bonobos were certainly not afraid of their own keepers. Restrictions to test mixed sex pairs resulted from tension towards one particular male and are thoroughly discussed in the discussion.
Individual Rope Pulling Training	Reviewer	No – they spontaneously cooperated for large amounts of food in Exp 1 and 2.	Yes - 5 individual test sessions to train subjects to pull rope / tray.
	Comments	Experiment 1 did not test cooperation. As for experiment 2, individual training was administered: Hare et al 2007 Suppl. S1: „Before participating in the cooperation test, each subject was individually introduced to the cooperation apparatus. All subjects participated in one session of six trials. The majority of subjects quickly learned to use the rope	We wanted to make sure that all individuals in our study had the same basic knowledge of the apparatus, independent of their previous knowledge and especially since all apes were not tested for nearly two years prior to the start of our study. If anything, the solitary training would have no

		to pull in the food, although some did not use the rope successfully in this initial introduction.“	effect on cooperation levels, based on Melis et al., 2006a.
Cooperation Pretest	Reviewer	Yes - all pairs of both species demonstrate success cooperating for highly sharable food before critical test (6 trials).	No – no introduction to dyadic cooperation task with highly sharable food before critical test.
	Comments	Are you referring to the condition dispersed divisible? This is not a cooperation pretest and was never discussed as such, but an important condition to argue about cooperation.	We incorporated the dispersed divisible condition the same way as Hare et al. did.
Experimenter calls subjects name and throws rope	Reviewer	Yes – this allowed to control for species differences in self-control by getting rope ends to both subjects simultaneously.	No – subjects must inhibit pulling until partner can get access to rope that human threads through small mesh hole - does not control for species differences in social inhibition.
	Comments		Both experimenters placed the rope into the cage simultaneously, so no social inhibition needed.

Misrepresentative figure: The figure does not accurately represent the physical arrangement in the sleeping rooms at WKPRC - unless there have been major renovations. The mesh panels are much smaller and mesh does not go to the floor. Scale should be added and the actual dimensions of mesh / support beams / plexiglass represented.

→ Thank you. In our previous manuscript we used a reduced depiction since we set out to test at several different facilities and merely wanted to convey the general gist. We have now adapted the figure (including scale and dimensions in the figure caption) to show the structure of the mesh and how we managed to raise the platform in order to test at WKPRC. Please see below.

Figure 1. In the cooperation task, the two apes had to pull at both ends of the rope to pull in the platform. In the co-feeding task, no rope was used and, instead, two experimenters pushed the platform towards the mesh. The depiction of the cage mesh is representing the actual scale, the width of the platform was 2.7m and its initial distance to the mesh was 1m.

Response to Reviewer 2

Dear reviewer,

Thanks a lot for your additional and helpful input. Please see below our response.

We wish you all the best and thank you for improving this manuscript.

Suska Nolte, Liesbeth Sterck, Edwin van Leeuwen

I recommend 'accept with minor revision' with the following suggestions for improvement:

1. Provide information (e.g. a supplementary table) on the specific behaviors coded as prosocial, playful, or aggressive in section 4.2.

Indeed, such a table would improve the paper and we now provide it in the supplementary material, as suggested (ESM Table 5).

2. As described in Hare et al. 2007 (see also Hare, Brian, and Michael Tomasello. "The emotional reactivity hypothesis and cognitive evolution." 2005, which I'm surprised is not included in the literature cited) success on cooperative problem solving tasks can be hindered by aggression. In Hare et al. 2007, aggression was found to be non-significant/non-existent. The current study does not include aggressive behavior in the new analyses of co-feeding or cooperation, and it is unclear from the provided information whether this is due to an absence of aggressive behavior during trials or whether the authors had other justifications for only including affiliative behavior in the new models. If, as in Hare 2007, aggressive behaviors were found to be NS or did not occur, this should be stated. Otherwise, a justification should be provided.

Thank you for your comment. In the cooperation task, we only observed aggression during a total of ten trials, which precluded an assessment of the potentially hindering effect on cooperation. To assess species differences in aggression, we did replicate the original analyses and compared the frequency of aggression between the two species using t-tests. The above is now made transparent in the Methods describing the GLMM to analyse the cooperation data, and the paper is cited:

- Line 339-342: “We could not estimate the potentially hindering effect of aggressive interactions on cooperation (Hare et al., 2005) given that we only observed ten trials (out of 629 total trials) of aggressive interactions across eight chimpanzee dyads and zero cases across all bonobo dyads. Thus, results would not be robust due to a lack in sufficient data points.”

Appendix F

Review of revised paper “Does Tolerance Allow Bonobos to Outperform Chimpanzees on a Cooperative Task? A conceptual replication of Hare et al., 2007”

I really appreciate the work the authors have put into this revision. Clearly a lot of work so I feel for them. They have also in many ways been very responsive to suggestions and critiques. I applaud them for this. I will be curious to see what the authors discovered using their new method of testing co-feeding and cooperation in bonobos and chimpanzees. I also appreciate their detailed responses to my table of method differences. However, unfortunately, I do think at this point we will probably just have to agree to disagree on a number of points. Below are my main concerns that remain (i.e. in many ways repeating what I wrote previously).

Summary of main method concerns:

- *Underpowered:* The authors only had access to a relatively small number of adult apes resulting in less powerful statistical approach (i.e. many studies including the one they are modeling their own after have had larger samples and more power). This is particularly acute in the case of the bonobos. Replications should have larger samples / more power not less.
- *Novel apparatus:* I am glad to see the authors provided a scaled figure of their novel test apparatus in their revision. It does reveal however, that instead of having the apparatus on ground, as in all previous studies, the new apparatus is elevated. This introduces gravity bias as a potential confound in performance. Perhaps even more important, by providing a figure in the original that was meant to “convey the general gist” (as the authors put it) the authors would have left other researchers unable to faithfully replicate their new method. This is an odd oversight in a study initially claiming to be a methodological replication and given the authors grumblings about what they perceive as missing details in Hare et al. My interpretation is the replication is really hard and even when people are doing their very best to provide precise methods some details get left out by accident / oversight. That should be expected and authors replicating studies should contact the original authors and give lots of time in study plans to coordinate (i.e. months). It is why the many primates, birds and dogs effort are allowing / putting so much time in to coordinating. I would argue the idea you can faithfully replicate an animal cognition study based on published methods alone – without significant consultation – is not our best bet as a field.
- *Statistical control only:* Due to the limits of their sample size the authors must rely on statistical estimates instead of *a priori* design features to control for the impact of dyad age-sex and repeated testing. They then seem to suggest in their review responses that statistical estimates are apples to apples with *a priori* control via sampling/design features (as used in Hare et al 2007). For example, I don’t think anyone would argue for a design that statistically controls for sex effects over a sampling design that tests equal number of males and females. Likewise, no one would prefer a design to test for spontaneous problem solving that relies on resampling the same small group of individuals repeatedly over testing a large sample once or in a small set of trials.
- *Not enough food:* The authors, constrained by the test site, were unable to provide large quantities of food to initiate cooperation before reducing food in a test of constraints on food sharing / cooperation.
- *Arbitrary norm:* The authors appear to be operating as if there is some agreed rule that only information cited in the paper being replicated is relevant to the evaluation of a replication. I have no idea what this is based on, but it is unclear how this is helpful. For example, Wobber and Hare, 2011 PLoS One show that >85% LEJ chimpanzees show coprophagy compared to 4% of sanctuary chimpanzees (apologies I had referred to regurgitation in table of previous review). This same paper discusses the relation of the lack of browse and the large amount of indoor time

the LEJ chimpanzees must endure as it relates to their aberrant behaviors. There is also work showing that forcing apes into smaller spaces for long periods cause tensions in the group they have to work to overcome (e.g. Aureli & De Waal, 1997). This effect may be more pronounced in some species than others (i.e. so having bonobos in the “same” environment instead of an ecologically appropriate may actually artificially cause big differences in social behavior). As stated in my previous review, this is highly relevant to the feeding regime used in LEJ and its influence on tolerance / aggression of the apes there, yet the authors basically brush this off in their response suggesting that because nothing is published on this in the paper, they are replicating this is not relevant.

Aureli, F. and De Waal, F.B., 1997. Inhibition of social behavior in chimpanzees under high-density conditions. *American Journal of Primatology*, 41(3), pp.213-228.

- *Previous replications:* There are existing published papers that provide replications but the authors apparently do not recognize them as such. Wobber et al 2010 *Curr Bio* very closely follows the sharing methods used in Hare et al 2007 (and clearly outline the 2-3 method differences). The biggest differences, as the authors correctly point out, is the placement of the food inside versus outside the test room. However, the authors here have acknowledged in their own experiment subjects could not reach their arms through the mesh unlike Hare et al 2007. So...why their current work is more of a replication of the co-feeding tests than Wobber et al is not? This seems pretty arbitrary. Wobber et al also do provide a test with a larger sample of subjects than Hare et al 2007, data evaluating both the species differences and age differences in co-feeding (in a supplemental section labeled “Comparison of food sharing with Hare et al 2007”). In addition, Experiment 3 of Hare et al, 2007 provides two separate tests of cooperation (round 1 and round 2). These two tests were separated in time and included a different combination of subjects (i.e. novel dyad combinations). That is a within study replication and methodologically faithful.
- *Experimentally experienced vs. experimentally naïve subjects:* The authors seem to discount this huge method difference by stating that the apes in their study had not been in any research for two years. First, the LEJ apes are the most experimentally experienced apes in the history of the world with the exception perhaps of the PRI apes. Second, the LEJ chimpanzees had demonstrated that apes can remember methodological details for years (Lewis et al, 2019) So the idea that a two year delay in testing somehow means these apes are naïve or comparable to the sanctuary apes is a pretty big oversight to not consider. It would be on the authors to show that in the case of cooperation your subjects are unaffected and behave like naïve apes if you want to make that claim.

Lewis, Berntsen and Call (2019) Long-term memory of past events in great apes. *Current Directions in Psychological Science* 28, 117-123.

Smaller details:

- The authors argue that the introduction of the pulling task to the bonobos in Hare et al 2007 is evidence the bonobos were “trained”. This is not the characterization used in Hare et al b/c the bonobos quickly met the criteria making few if any mistakes. Training is typically used to characterize efforts that require conditioning and dozens and hundreds of mistakes occur before an animal can meet a criteria.
- Regarding my comments on the authors characterizations of the two species hunting behavior in the introduction. This is from the original submission with the bolded text most relevant to my response:

“Wild populations of chimpanzees exhibit collective behaviours such as boundary patrols, territorial defence, group hunting and subsequent meat sharing (e.g. Boesch & Boesch, 1989; Mitani & Watts, 1999; Watts & Mitani, 2001). They inform others of danger via alarm calls, while bonobos seem less motivated to do so (Girard-Buttoz et al., 2020). Even though wild bonobos are rarely seen to perform actions such as boundary patrols and collective territorial defence (Furuichi, 2011; Surbeck & Hohmann, 2008; Wilson et al., 2014), **there is a growing body of evidence that they do perform group hunting and also share meat and fruits** (Hohmann & Fruth, 1996, 2008; Yamamoto & Furuichi, 2017; Samuni et al., 2020; Wakefield et al., 2019). Both species form coalitions and support their partners during fights, which usually occurs among males in chimpanzees and among females in bonobos (Boesch & Boesch, 1989; Hohmann & Fruth, 1996; Mitani & Watts, 1999; Tokuyama & Furuichi, 2016; Yamamoto & Furuichi, 2017). Thus, in the wild, chimpanzees and bonobos both engage in collective behaviours, with chimpanzees exhibiting more varied behaviours in aggressive and competitive contexts.”

The highlighted text can be read to mean that bonobo hunting is similar to chimpanzee hunting. This just needs to be clarified. They are both qualitatively and quantitatively very different as I explained in my last review.

- In their introduction of intergroup relations in bonobos the authors may also enjoy and want to share with their readers (Cheng et al 2022)

<https://www.sciencedirect.com/science/article/abs/pii/S0003347222000550?via%3Dihub>

Brandt, M.J., IJzerman, H., Dijksterhuis, A., Farach, F.J., Geller, J., Giner-Sorolla, R., Grange, J.A., Perugini, M., Spies, J.R. and Van't Veer, A., 2014. The replication recipe: What makes for a convincing replication?. *Journal of Experimental Social Psychology*, 50, pp.217-224.

- See green text below. Again, this highlights a major method difference. As for what is normal, all ape facility I have worked with and continue to work with allow for larger amounts of fruit in the diet and during experiments. This is really idiosyncratic to this zoo and its managers – and it is they that have no statistical analysis that apes should be fed this way. All evidence suggest ad libitum browse will relieve the stress (coprophagy, regurgitation, rocking) and diarrhea that these apes constantly suffer from in LEJ. Restricting fruit is treating the symptom not the cause.

regarding these aspects in any of the stated papers. All we have been able to do is work within the contemporary diet protocols (i.e., food amounts in apes are restricted nowadays, more so than 15 years ago) and, most importantly, keep

- The authors revise the table I provided for publication. That is really cool move. I would argue though that many of the variables they select out as “inaccurate” or not in published paper are not so black and white. See below.

		Hare et al, 2007	Current study
Feeding regime	Reviewer	Ad libitum browse always available to subjects during day (similar to wild). No regurgitation observed.	Food only available during meals and experiments – very little low quality browse available => aberrant regurgitation
	Comments	No published information available in Hare et al. 2007 or Melis et al. 2006a	

See Wobber et al 2010 that directly compares the coprophagy rates of Leipzig chimps (some tested in the current study) to those living at the sanctuary used in Wobber et al 2010 to compare sharing rates in bonobos and chimpanzees.

As stated above, the authors seem to be operating under some unwritten rule that they are not responsible for being aware of larger literature that is relevant to their replication. It seems they believe that if the information is not in the actual paper being replicated, they can somehow disregard. I have no idea what this is based on but it does not really seem justified as far as I can tell. If there is some citation that makes the claim that replications can ignore anything about reality outside the paper being replicated, please do pass that on in your response.

Age matched Dyads	Reviewer	Yes – within and across species (all subjects in dyad 0-3 yrs apart in age)	No – dyads of different ages paired.
	Comments	Several dyads included with huge age difference tested in exp. 1 and 2.	As stated above in Table 1b, the age of both partners is included in the analyses.

Figure S4 from Hare et al, 2007 shows that all dyads in Experiment 3 were within 0-3 years of each other (with the exception of one bonobo pair from Leipzig Zoo Joey-Yasa). Given the constraints of the demography of the small populations at the Leipzig zoo, the authors are left using statistical methods.

Sex balanced Dyads	Reviewer	Yes – within and across species MM/FF/MF dyads basically equally present	No – dyads of different sexes re-paired throughout – although oddly no F-F pairings (suggests some serious group instability or caretaking / management issues).
	Comments	Not true based on published information.	Several F-F pairings were included in both species. The effect of dyad sex estimated with statistical

Again, in Table S4 there are 6 M-F bonobo dyads compared to 5 M-F; 1 M-M chimpanzee dyads in Round 1 and in the 2nd replication round 3 F-F, 2 M-M, 1 M-F bonobo dyads compared to 3 F-F, 3 M-M chimpanzee dyads. Again, the authors of the current paper can only statically estimate.

“clean dyads” in test trials (exp 3 A & B)	Reviewer	Yes - 6 test trials in session A and B = 12 trials total per subject / dyad. 6 clean dyads per species in A & B.	No - With only 7 testable bonobos it would only be possible to test 3 clean dyads – so re-pairing and repeated testing of same subjects necessary.
	Comments	Not true based on published information. See adapted information of this above (“Dyad composition”)	We ensured independence of results in our analyses (see above “Dyad composition”)

unclear how authors arrive to this being “not true”. Interesting. Regardless they only tested 7 bonobos.

**Experimenters
move subjects**

Reviewer

Yes – this allowed reduced subject anxiety using only positive reinforcement during test (critical for reactive bonobos).

No – in LEJ only caretakers move subjects and bonobos in particular have long history of anxiety toward their caretakers. Restrictions on their ability to test mix sex pairs as a result.

Comments

Though we cannot speak for the experience of this research team 15+ years ago, but at the time of this study, bonobos were certainly not afraid of their own keepers. Restrictions to test mixed sex pairs resulted from tension towards one particular male and are thoroughly discussed in the discussion.

Even if the authors are correct here it still means that in Hare et al 2007 the sanctuary subjects were moved and work with experimenters but in Leipzig they are moved by caretakers. Just another method differences very important to the subjects.

Individual Rope Pulling Training	Reviewer	No – they spontaneously cooperated for large amounts of food in Exp 1 and 2.	Yes - 5 individual test sessions to train subjects to pull rope / tray.
	Comments	Experiment 1 did not test cooperation. As for experiment 2, individual training was administered: Hare et al 2007 Suppl. S1: „Before participating in the cooperation test, each subject was individually introduced to the cooperation apparatus. All subjects participated in one session of six trials. The majority of subjects quickly learned to use the rope	We wanted to make sure that all individuals in our study had the same basic knowledge of the apparatus, independent of their previous knowledge and especially since all apes were not tested for nearly two years prior to the start of our study. If anything, the solitary training would have no

<https://mc.manuscriptcentral.com/rsos>

to pull in the food, although some did not use the rope successfully in this initial introduction.“

effect on cooperation levels, based on Melis et al., 2006a.

I am not sure why they see my statements as incorrect. Their comments simply reaffirm what I said. The two studies used different methods here. This is a significant methodological difference.

Cooperation Pretest	Reviewer	introduction. Yes - all pairs of both species demonstrate success cooperating for highly sharable food before critical test (6 trials).	No – no introduction to dyadic cooperation task with highly sharable food before critical test.
	Comments	Are you referring to the condition dispersed divisible? This is not a cooperation pretest and was never discussed as such, but an important condition to argue about cooperation.	We incorporated the dispersed divisible condition the same way as Hare et al. did.

The authors have repeatedly said they were not allowed to use large amounts of food necessary to conduct the dispersed divisible condition that was conducted in Hare et al 2007. This method difference remains.

Experimenter calls subjects name and throws rope	Reviewer	Yes – this allowed to control for species differences in self-control by getting rope ends to both subjects simultaneously.	No – subjects must inhibit pulling until partner can get access to rope that human threads through small mesh hole - does not control for species differences in social inhibition.
	Comments		Both experimenters placed the rope into the cage simultaneously, so no social inhibition needed.

This method difference still stands. It would be particularly important for the authors to demonstrate “that no social inhibition needed” perhaps through supplemental coding. There are a number of studies showing the two species differ in inhibition. This could advantage the chimpanzees in particular and the authors may only have an inhibition test not one evaluating cooperative ability.

1 Dear Editor,

2
3 Thank you for your favourable evaluation of our work. We appreciate your thorough approach, including
4 the acknowledgement of our continued attempts to report our study as accurate and transparently as
5 possible. Please find our final responses to Reviewer 1 below and the finalized Table with differences
6 between the two studies at the end.

7
8 Thank you.

9 Sincerely,

10 Suska Nolte, Liesbeth Sterck, Edwin van Leeuwen

11
12
13
14 *Underpowered:* The authors only had access to a relatively small number of adult apes resulting in less
15 powerful statistical approach (i.e. many studies including the one they are modeling their own after have
16 had larger samples and more power). This is particularly acute in the case of the bonobos. Replications
17 should have larger samples / more power not less.

18
19 Indeed, the sampled groups encompass fewer individuals than those in the original study, yet a similar
20 and larger number of dyads, which is the level where the true power of the analysis lies. Since we tested
21 each dyad in both experiments - using a within-subject design - we control for influential dyad
22 characteristics that otherwise can drive results. Further, instead of using t-tests, we apply statistical
23 methods that 1) incorporate several explanatory variables and 2) control for the influence of individual
24 apes and dyad combinations, resulting in a more powerful statistical approach.

25
26
27 *Novel apparatus:* I am glad to see the authors provided a scaled figure of their novel test apparatus in
28 their revision. It does reveal however, that instead of having the apparatus on ground, as in all previous
29 studies, the new apparatus is elevated. This introduces gravity bias as a potential confound in performance.
30 Perhaps even more important, by providing a figure in the original that was meant to “convey the general
31 gist” (as the authors put it) the authors would have left other researchers unable to faithfully replicate their
32 new method. This is an odd oversight in a study initially claiming to be a methodological replication and
33 given the authors grumblings about what they perceive as missing details in Hare et al. My interpretation
34 is the replication is really hard and even when people are doing their very best to provide precise methods
35 some details get left out by accident / oversight. That should be expected and authors replicating studies
36 should contact the original authors and give lots of time in study plans to coordinate (i.e. months). It is
37 why the many primates, birds and dogs effort are allowing / putting so much time in to coordinating. I
38 would argue the idea you can faithfully replicate an animal cognition study based on published methods
39 alone – without significant consultation - is not our best bet as a field.

40
41 We fail to see how a gravity bias is involved in our study.

42
43 A *gravity bias* (Hood, 1995, 1998) refers to the inability of young children and many nonhuman
44 animals (e.g. Hood et al., 1999; Tomonaga et al., 2007) to understand that the trajectory of a

45 falling object can be diverted, e.g. because the tube into which a ball was dropped is curved. Thus,
 46 even when facing counter evidence, they search horizontally underneath the location of insertion
 47 expecting objects to fall in a straight line. Once a transparent instead of opaque tube is used, the
 48 error fades.

49
 50 Hood, B. M. (1995). Gravity rules for 2-to 4-year olds?. *Cognitive Development*, 10(4), 577-598.

51 Hood, B. M. (1998). Gravity does rule for falling events. *Developmental Science*, 1(1), 59-63.

52 Hood, B. M., Hauser, M. D., Anderson, L., & Santos, L. (1999). Gravity biases in a non-human
 53 primate?. *Developmental Science*, 2(1), 35-41.

54 Tomonaga, M., Imura, T., Mizuno, Y., & Tanaka, M. (2007). Gravity bias in young and adult
 55 chimpanzees (*Pan troglodytes*): Tests with a modified opaque-tubes task. *Developmental science*,
 56 10(3), 411-421.

57
 58 Our study merely made use of a small foundation on which the sliding tray was located in order to align
 59 the tray with the apes' cages. Our study uses neither opaque mechanisms, nor falling objects. Moreover,
 60 the apparatus was pulled horizontally, which defies any reference to a gravity bias. Lastly, with respect to
 61 the reviewer's comment regarding putting time and effort into replications, we wish to highlight for the
 62 record that we *did* contact the original first author and put a lot of time into planning our study.

63
 64
 65 *Statistical control only:* Due to the limits of their sample size the authors must rely on statistical estimates
 66 instead of *a priori* design features to control for the impact of dyad age-sex and repeated testing. They
 67 then seem to suggest in their review responses that statistical estimates are apples to apples with *a priori*
 68 control via sampling/design features (as used in Hare et al 2007). For example, I don't think anyone
 69 would argue for a design that statistically controls for sex effects over a sampling design that tests equal
 70 number of males and females. Likewise, no one would prefer a design to test for spontaneous problem
 71 solving that relies on resampling the same small group of individuals repeatedly over testing a large
 72 sample once or in a small set of trials.

73
 74 We do not suggest anything like that, we merely convey that statistically controlling for repeated testing
 75 is a powerful and necessary technique to reduce noise in factor estimations like the effect of individual
 76 age and sex in a dyad. If it would be possible to include the exact same age per individual per species in
 77 each experiment and condition, then accounting for age would not be important. Similarly, if the exact
 78 same number of sex combinations per species would be included in each experiment and each condition,
 79 accounting for sex would become redundant. However, in practice (also in the original study) this is often
 80 not possible. Thus, statistical methods are necessary to control for such deviations. This does not mean
 81 that a balanced sample should not be the goal whenever possible. Moreover, bigger populations from
 82 which samples can be drawn are always better, but not always realistically available. Yet, this would be
 83 discarding half the literature in our field. Finally, we wish to emphasize that we set out to understand
 84 general patterns and underlying influential factors of cooperation and tolerance by testing as many dyad
 85 constellations as possible, and not just the most tolerant partners like in the original study.

88 *Not enough food:* The authors, constrained by the test site, were unable to provide large quantities of food
 89 to initiate cooperation before reducing food in a test of constraints on food sharing / cooperation.
 90

91 We have responded to this aspect repeatedly. We suffice here by saying that i) we worked under the
 92 contemporary diet guidelines of the ape facility, ii) we chose to use the exact same amount of food as in
 93 the *clumped divisible* condition of the original authors' cooperation task, and iii) we kept this amount
 94 consistent across conditions, thereby making conditions comparable within our study.
 95

96
 97 *Arbitrary norm:* The authors appear to be operating as if there is some agreed rule that only information
 98 cited in the paper being replicated is relevant to the evaluation of a replication. I have no idea what this is
 99 based on, but it is unclear how this is helpful. For example, Wobber and Hare, 2011 PLoS One show
 100 that >85% LEJ chimpanzees show coprophagy compared to 4% of sanctuary chimpanzees (apologies I
 101 had referred to regurgitation in table of previous review). This same paper discusses the relation of the
 102 lack of browse and the large amount of indoor time the LEJ chimpanzees must endure as it relates to their
 103 aberrant behaviors. There is also work showing that forcing apes into smaller spaces for long periods
 104 cause tensions in the group they have to work to overcome (e.g. Aureli & De Waal, 1997). This effect
 105 may be more pronounced in some species than others (i.e. so having bonobos in the "same" environment
 106 instead of an ecologically appropriate may actually artificially cause big differences in social behavior).
 107 As stated in my previous review, this is highly relevant to the feeding regime used in LEJ and its
 108 influence on tolerance / aggression of the apes there, yet the authors basically brush this off in their
 109 response suggesting that because nothing is published on this in the paper, they are replicating this is not
 110 relevant.

111 Aureli, F. and De Waal, F.B., 1997. Inhibition of social behavior in chimpanzees under high-density
 112 conditions. *American Journal of Primatology*, 41(3), pp.213-228.
 113

114 We do not wield this assumption. In our original submission we have aimed to select those studies that
 115 truly report a species comparison within the same study design for the reason that comparing across
 116 single-species studies comes with many confounding variables. In our resubmission, we have additionally
 117 incorporated all of the reviewer's suggestions of studies that may bear relevance to our research question,
 118 despite the mentioned lack of validity. For instance, where the reviewer writes "This effect may be more
 119 pronounced in some species than others", based on a study in only one of the two species (chimpanzees:
 120 Aureli & de Waal, 1997), we see this as (a potentially endless) speculation with many ramifications that
 121 are not conducive to the empirical assessment of the similarities and differences between chimpanzees
 122 and bonobos.
 123

124
 125 *Previous replications:* There are existing published papers that provide replications but the authors
 126 apparently do not recognize them as such. Wobber et al 2010 *Curr Bio* very closely follows the sharing
 127 methods used in Hare et al 2007 (and clearly outline the 2-3 method differences). The biggest differences,
 128 as the authors correctly point out, is the placement of the food inside versus outside the test room.
 129 However, the authors here have acknowledged in their own experiment subjects could not reach their
 130 arms through the mesh unlike Hare et al 2007. So...why their current work is more of a replication of the
 131 co-feeding tests than Wobber et al is not? This seems pretty arbitrary. Wobber et al also do provide a test

with a larger sample of subjects than Hare et al 2007, data evaluating both the species differences and age differences in co-feeding (in a supplemental section labeled “Comparison of food sharing with Hare et al 2007”).

We do recognize and highly value the study by Wobber et al, 2010, and, as stated earlier, their respective results in term of co-feeding differences are included in the introduction of our manuscript. We do not dedicate a separate paragraph to this study in terms of its replication value since 1) it does not include both experiments, and 2) the authors themselves explicitly state that they did *not* faithfully replicate the original study for several reasons. The best we can do is to acknowledge methodological differences (see our Table), highlight and discuss them, and present our study as “conceptual” replication, which we now all do.

In addition, Experiment 3 of Hare et al, 2007 provides two separate tests of cooperation (round 1 and round 2). These two tests were separated in time and included a different combination of subjects (i.e. novel dyad combinations). That is a within study replication and methodologically faithful.

We have responded to this comment in detail before and reiterate here:

Stating that Hare et al 2007 Round 1 and 2 can be seen as a replication that we should discuss as such is a far stretch to what was done. In Round 1, dyads were tested in both the dispersed and clumped condition. In Round 2, a new combination of partners was tested only in the clumped condition. Instead of testing the dispersed condition, the scores of both partners they achieved with the previous partner were averaged. Thus, if AB had a score of 6 and CD had a score of 0, then the data provided and analysed for BC was 3, ignoring the fact that averaging like this will most likely not lead to a true score. As for the condition dispersed divisible, we do not agree with the reviewer’s proposition that recombining partners within the same experiment should be considered a replication of results.

Nonetheless, we do discuss that the original study includes a replication of their own results, given that a year passed in between Experiment 2 and 3 (not between round 1 and 2 of experiment 3). We address it as follows in our manuscript: “In “Experiment 2”, the original authors conducted the cooperation test with shareable food distribution only, and replicated both the methods and results in “Experiment 3”, while additionally incorporating a condition with monopolizable food distribution.”

Experimentally experienced vs. experimentally naïve subjects: The authors seem to discount this huge method difference by stating that the apes in their study had not been in any research for two years. First, the LEJ apes are the most experimentally experienced apes in the history of the world with the exception perhaps of the PRI apes. Second, the LEJ chimpanzees had demonstrated that apes can remember methodological details for years (Lewis et al, 2019) So the idea that a two year delay in testing somehow means these apes are naïve or comparable to the sanctuary apes is a pretty big oversight to not consider. It would be on the authors to show that in the case of cooperation your subjects are unaffected and behave like naïve apes if you want to make that claim.

175 Lewis, Berntsen and Call (2019) Long-term memory of past events in great apes. *Current Directions in*
 176 *Psychological Science* 28, 117-123.

177
 178 Nowhere do we make the claim that the tested apes in our study never participated in cognitive tests. Both
 179 species in our study have had experience with behavioural testing and we address this explicitly (see line
 180 121 and Table 1b Experimental History).

184 **Smaller Details**

185 The authors argue that the introduction of the pulling task to the bonobos in Hare et al 2007 is evidence
 186 the bonobos were “trained”. This is not the characterization used in Hare et al b/c the bonobos quickly
 187 met the criteria making few if any mistakes. Training is typically used to characterize efforts that require
 188 conditioning and dozens and hundreds of mistakes occur before an animal can meet a criteria.

189
 190 Hare et al. Suppl. S2: “Two pairs of bonobos made no mistakes in their first session, so this session was
 191 simply used as their first test session, whereas the other pairs were all proficient within 13 trials.” Instead
 192 of a mere “warm-up”, this can be plausibly seen as a training phase of up to 13 trials for most of the tested
 193 dyads. The “warm-up” was not applied individually but to those dyads that subsequently provided data,
 194 with the data immediately used when the dyad made no mistakes (see citation above). Note that the same
 195 was *not* done for the chimpanzees since they had already ‘worked’ with the paradigm, which in our
 196 perspective causes a potential difference between the two species tested. In our study, we had the exact
 197 same criteria for training applied to both species, and training was administered individually to ensure that
 198 apes learned how to pull in the apparatus but not how to cooperate with the given partner.

199
 200
 201 Regarding my comments on the authors characterizations of the two species hunting behavior in the
 202 introduction. This is from the original submission with the bolded text most relevant to my response:
 203 “Wild populations of chimpanzees exhibit collective behaviours such as boundary patrols, territorial
 204 defence, group hunting and subsequent meat sharing (e.g. Boesch & Boesch, 1989; Mitani & Watts, 1999;
 205 Watts & Mitani, 2001). They inform others of danger via alarm calls, while bonobos seem less motivated
 206 to do so (Girard-Buttoz et al., 2020). Even though wild bonobos are rarely seen to perform actions such as
 207 boundary patrols and collective territorial defence (Furuichi, 2011; Surbeck & Hohmann, 2008; Wilson et
 208 al., 2014), **there is a growing body of evidence that they do perform group hunting and also share**
 209 **meat and fruits** (Hohmann & Fruth, 1996, 2008; Yamamoto & Furuichi, 2017; Samuni et al., 2020;
 210 Wakefield et al., 2019). Both species form coalitions and support their partners during fights, which
 211 usually occurs among males in chimpanzees and among females in bonobos (Boesch & Boesch, 1989;
 212 Hohmann & Fruth, 1996; Mitani & Watts, 1999; Tokuyama & Furuichi, 2016; Yamamoto & Furuichi,
 213 2017). Thus, in the wild, chimpanzees and bonobos both engage in collective behaviours, with
 214 chimpanzees exhibiting more varied behaviours in aggressive and competitive contexts.”

215 The highlighted text can be read to mean that bonobo hunting is similar to chimpanzee hunting. This just
 216 needs to be clarified. They are both qualitatively and quantitatively very different as I explained in my last
 217 review.

219 We are glad this has been resolved now after the second revision. This now reads, line 40: "...there is a
220 growing body of evidence that they are capable of group hunting, albeit do so much more
221 infrequent than chimpanzees (Samuni et al., 2020; Surbeck & Hohmann, 2008; Wakefield et al.,
222 2019)."

225 In their introduction of intergroup relations in bonobos the authors may also enjoy and want to share with
226 their readers (Cheng et al 2022)

227 <https://www.sciencedirect.com/science/article/abs/pii/S0003347222000550?via%3Dihub>

229 Yes, we agree, the new publication (published in May 2022) by Cheng et al. is very interesting and we
230 now include it in the introduction of our manuscript, see line 45-46: "In stark contrast to rather
231 aggressive and lethal intergroup encounters in chimpanzees, bonobo females (Tokuyama et al.,
232 2019; Furuichi, 2011) and males (Cheng et al., 2022) actively maintain tolerant intergroup
233 encounters and even share food with members of neighbouring communities (Fruth & Hohmann,
234 2018; Samuni et al., 2020)."

237 See green text below. Again, this highlights a major method difference. As for what is normal, all ape
238 facility I have worked with and continue to work with allow for larger amounts of fruit in the diet and
239 during experiments. This is really idiosyncratic to this zoo and its managers – and it is they that have no
240 statistical analysis that apes should be fed this way. All evidence suggest ad libitum browse will relieve
241 the stress (coprophagy, regurgitation, rocking) and diarrhea that these apes constantly suffer from in LEJ.
242 Restricting fruit is treating the symptom not the cause.

regarding these aspects in any of the stated papers. All we have been able to do
is work within the contemporary diet protocols (i.e., food amounts in apes are
restricted nowadays, more so than 15 years ago) and, most importantly, keep

243 We unfortunately cannot decide on the husbandry style at the locations we test at.

245 Just as a personal exchange: many of the behaviours such as rocking when in stress or bored etc once seen
246 in the LEJ individuals have considerably reduced over the past 10 years that I (SN) had a chance to
247 observe them.

Table

		Hare et al, 2007	Current study
Feeding regime	Reviewer	Ad libitum browse always available to subjects during day (similar to wild). No regurgitation observed.	Food only available during meals and experiments – very little low quality browse available => aberrant regurgitation
	Comments	No published information available in Hare et al. 2007 or Melis et al. 2006a	

See Wobber et al 2010 that directly compares the coprophagy rates of Leipzig chimps (some tested in the current study) to those living at the sanctuary used in Wobber et al 2010 to compare sharing rates in bonobos and chimpanzees.

As stated above, the authors seem to be operating under some unwritten rule that they are not responsible for being aware of larger literature that is relevant to their replication. It seems they believe that if the information is not in the actual paper being replicated, they can somehow disregard. I have no idea what this is based on but it does not really seem justified as far as I can tell. If there is some citation that makes the claim that replications can ignore anything about reality outside the paper being replicated, please do pass that on in your response.

We do not believe such a thing and feel wrongly interpreted by the reviewer. We have repeatedly stated our justification for including studies in our theoretical framework. In this particular case, we do not see how coprophagy rates of a few individuals from more than 10 years ago can tell us anything meaningful about co-feeding and cooperation rates in our current study.

Age matched Dyads	Reviewer	Yes – within and across species (all subjects in dyad 0-3 yrs apart in age)	No – dyads of different ages paired.
	Comments	Several dyads included with huge age difference tested in exp. 1 and 2.	As stated above in Table 1b, the age of both partners is included in the analyses.

Figure S4 from Hare et al, 2007 shows that all dyads in Experiment 3 were within 0-3 years of each other (with the exception of one bonobo pair from Leipzig Zoo Joey-Yasa). Given the constraints of the demography of the small populations at the Leipzig zoo, the authors are left using statistical methods.

We extract the following information from the original study by Hare et al 2007:

In experiment 1 & 2: Three dyads with age difference 9 to 14 years.

In experiment 3: Two dyads with age difference 11 to 20 years.

As written above, realistically, it is often not possible to use precise age-matching in experiments with great apes – there is just too much natural variability. Not accounting for this variability by means of statistical procedures yields spurious results, so this is a shortcoming of the original study, if anything.

Based on the incorrectness of the information, we did and do not see it a fitting element in the Table.

Sex balanced Dyads	Reviewer	Yes – within and across species MM/FF/MF dyads basically equally present	No – dyads of different sexes re-paired throughout – although oddly no F-F pairings (suggests some serious group instability or caretaking / management issues).
	Comments	Not true based on published information.	Several F-F pairings were included in both species. The effect of dyad sex estimated with statistical

Again, in Table S4 there are 6 M-F bonobo dyads compared to 5 M-F; 1 M-M chimpanzee dyads in Round 1 and in the 2nd replication round 3 F-F, 2 M-M, 1 M-F bonobo dyads compared to 3 F-F, 3 M-M chimpanzee dyads. Again, the authors of the current paper can only statically estimate.

Whereas the Reviewer’s remark in the Table “MM/FF/MF dyads basically equally present” is a subjective statement based on the numbers given. We repeat that we do have several FF dyads of both species included in our study, which contrasts the Reviewer’s comment that ‘oddly no F-F pairings...’ were used in our study. We conclude that in the original study, the numbers are not balanced across species, sex-combinations or experiments (see below for details). Moreover, we conclude that in our study, we have sex-dyad pairings in all combinations, acknowledge and report that the numbers are not equal, and use the appropriate statistical procedures to estimate the potential effects of these unequal numbers. Given that readers have immediate access to all this information, we do not see it a fitting element in the Table.

Experiment 1&2:

Bonobo data includes 4 M-M, 4 F-F, 2 M-F

Chimp data includes 4 M-M, 6 F-F, 6 M-F

Experiment 3:

Bonobo data includes 7 M-F, 3 F-F, 2 M-M

Chimp data includes 5 M-F; 3 F-F, 4 M-M

“clean dyads” in test trials (exp 3 A & B)	Reviewer	Yes - 6 test trials in session A and B = 12 trials total per subject / dyad. 6 clean dyads per species in A & B.	No - With only 7 testable bonobos it would only be possible to test 3 clean dyads – so re-pairing and repeated testing of same subjects necessary.
	Comments	Not true based on published information. See adapted information of this above (“Dyad composition”)	We ensured independence of results in our analyses (see above “Dyad composition”)

unclear how authors arrive to this being “not true”. Interesting. Regardless they only tested 7 bonobos.

Please see below Table 4 of the supplementary material provided by Hare et al 2007 with highlighted individuals that were only tested in either round 1 or round 2. The rest of the dyads was recombined, like we did. Thus, the dyads were not clean, as the reviewer states, either with regard to using a between or within subject design. Hence, we do not incorporate this element in our Table.

Table S4. The Name, Group, Sex of Dyad, Estimated Ages in Rounds 1 and 2 of Experiment 3

Round 1			
Bonobo		Group	Sex
1	Bandundu-Kikwit	Lola	M-F
2	Lukuya-Lipopo	Lola	M-F
3	Kikongo-Lisalle	Lola	M-F
4	Noiki-Matadi	Lola	M-F
5	Bsende-Likasi	Lola	M-F
6	Joey-Yasa	Leipzig	M-F
		Mean	
Chimpanzee			
1	Umugenzi-Bili	Ngamba	M-F
2	Baluku-Ndakyra	Ngamba	M-F
3	Bwambale-Nani	Ngamba	M-F
4	Indi-Pasa	Ngamba	M-F
5	Okech-Nakuu	Ngamba	M-F
6	Frodo-Patrick	Leipzig	M-M
		Mean	
Round 2			
Bonobo			
1	Bandundu-Lukaya	Lola	F-F
2	Kisanto-Noiki	Lola	F-F
4	Lisalle-Likasi	Lola	F-F
3	Kikwit-Lipopo	Lola	M-M
5	Kikongo-Bsende	Lola	M-M
6	Yasa-Kuno	Leipzig	M-F
		Mean	
Chimpanzee			
2	Ndakyra-Pasa	Ngamba	F-F
3	Nani-Nakuu	Ngamba	F-F
4	Bili-Yoyo	Ngamba	F-F
1	Umugenzi-Baluku	Ngamba	M-M
5	Okech-Bwambale	Ngamba	M-M
6	Robert-Patrick	Leipzig	M-M
		Mean	

An asterisk indicates the estimated average performance of dyad member from round 1 during efforts to obtain shara

Experimenters move subjects	Reviewer	Yes – this allowed reduced subject anxiety using only positive reinforcement during test (critical for reactive bonobos).	No – in LEJ only caretakers move subjects and bonobos in particular have long history of anxiety toward their caretakers. Restrictions on their ability to test mix sex pairs as a result.
--	-----------------	---	--

Comments

Though we cannot speak for the experience of this research team 15+ years ago, but at the time of this study, bonobos were certainly not afraid of their own keepers. Restrictions to test mixed sex pairs resulted from tension towards one particular male and are thoroughly discussed in the discussion.

334 Even if the authors are correct here it still means that in Hare et al 2007 the sanctuary subjects were
 335 moved and work with experimenters but in Leipzig they are moved by caretakers. Just another method
 336 differences very important to the subjects.

337

338 By this standard, no replication would ever be possible. We feel that the Reviewer is unjustly rude
 339 towards the caring staff of the LEJ facility. We reiterate that the keepers in our study did a great job, that
 340 the bonobos nor the chimpanzees were afraid, and moreover, that any potential effect of the persons
 341 shifting the apes (caretakers vs experimenters) could go in either direction.

Individual Rope Pulling Training	Reviewer	No – they spontaneously cooperated for large amounts of food in Exp 1 and 2.	Yes - 5 individual test sessions to train subjects to pull rope / tray.
	Comments	Experiment 1 did not test cooperation. As for experiment 2, individual training was administered: Hare et al 2007 Suppl. S1: „Before participating in the cooperation test, each subject was individually introduced to the cooperation apparatus. All subjects participated in one session of six trials. The majority of subjects quickly learned to use the rope	We wanted to make sure that all individuals in our study had the same basic knowledge of the apparatus, independent of their previous knowledge and especially since all apes were not tested for nearly two years prior to the start of our study. If anything, the solitary training would have no

<https://mc.manuscriptcentral.com/rsos>

Royal Society Open Science: For review only

Pa

to pull in the food, although some did not use the rope successfully in this initial introduction.“

effect on cooperation levels, based on Melis et al., 2006a.

I am not sure why they see my statements as incorrect. Their comments simply reaffirm what I said. The two studies used different methods here. This is a significant methodological difference.

The initial remark by the reviewer stated that no training was administered in Hare et al 2007. However, according to information provided in Suppl. S1, the authors did incorporate solitary training before the first cooperation test (Experiment 2). Similarly, we also incorporated solitary training before the first cooperation test. The only difference is that Hare et al 2007 administered one session with six trials and all apes were included in the cooperation test regardless of their ability to successfully manipulate the apparatus. We included up to five sessions (many passed during the first session) with multiple trials to enable a learning curve and ensure that apes knew how the apparatus works before testing their cooperation skills with it. We excluded all apes that did not meet the criteria of six successful consecutive trials. Out of 21 apes, only three chimpanzees did not pass training (all details are provided in our supplementary material). We now adapted the table and split the information previously provided under “Cooperation warm up” into two:

Familiarization Rope & Individual Training	Familiarization Rope: Bonobos received time to play with rope to reduce play behaviour afterwards Training: One session of six trials Criteria to pass into test: none, all tested	Familiarization Rope: None administered, no play behaviour Training: Up to five sessions with multiple trials Criteria to pass into test: six successful consecutive trials in one session
Cooperation "warm up"	"Three-trials-success" criteria of successful cooperation with the given partner in condition dispersed divisible (up to 13 trials per dyad) provided to bonobos at both facilities and chimpanzees at the WKPRC before cooperation experiment (Exp. 3) that included condition dispersed divisible and clumped.	No dyadic "warm-up" to either species, only solitary pre-test trials for rope-pulling understanding provided to both species (see above)

Cooperation Pretest	Reviewer	Introduction.	Yes - all pairs of both species demonstrate success cooperating for highly sharable food before critical test (6 trials).	No – no introduction to dyadic cooperation task with highly sharable food before critical test.
	Comments		Are you referring to the condition dispersed divisible? This is not a cooperation pretest and was never discussed as such, but an important condition to argue about cooperation.	We incorporated the dispersed divisible condition the same way as Hare et al. did.

369 The authors have repeatedly said they were not allowed to use large amounts of food necessary to conduct
370 the dispersed divisible condition that was conducted in Hare et al 2007. This method difference remains.

371
372 The reviewer is now referring to the condition *dispersed divisible* as a pretest of the condition *clumped*
373 *divisible*. This is a new label, and we have incorporated both these conditions as Hare et al. did. The fact
374 that food amounts were changed was already highlighted under methodological differences in the table.

Experimenter calls subjects name and throws rope	Reviewer	Yes – this allowed to control for species differences in self-control by getting rope ends to both subjects simultaneously.	No – subjects must inhibit pulling until partner can get access to rope that human threads through small mesh hole - does not control for species differences in social inhibition.
Comments		Both experimenters placed the rope into the cage simultaneously, so no social inhibition needed.	

This method difference still stands. It would be particularly important for the authors to demonstrate “that no social inhibition needed” perhaps through supplemental coding. There are a number of studies showing the two species differ in inhibition. This could advantage the chimpanzees in particular and the authors may only have an inhibition test not one evaluating cooperative ability.

We have clearly stated that the ropes were simultaneously presented to both apes with the help of an assistant. So, at the exact same time, two experiments provided one rope end to either side of the cage. Hence, the apes had access to both rope ends at the exact same time as in Hare et al 2007. This means that ‘subjects must inhibit pulling until partner can get access to rope...’ is simply not true. Our method covered this.

This will be included in the final manuscript:

Table 1a. Methodological differences between the original and the current study.

	Hare et al., 2007	Current study
Cooperation platforms	Tray length is 3.4 m wide (feeding dishes 2.7 m apart). ⁽¹⁾	Tray length is 2.7 m wide (feeding dishes 2.2 m apart) – subjects required to co-feed and cooperate in closer proximity.
Food amount	Co-feeding task tested with large amounts of food (0.5 kg of banana each trial). Cooperation task tested with large amounts of food in the dispersed condition and small amounts of food in the clumped condition (0.5 kg of banana each trial versus four 1.5 cm pieces).	Only tested with small amounts of food (four 1.5 cm pieces of banana each trial, replicating what was originally used in the clumped condition) in both experiments . For additional information see ESM Table 2.
Familiarization Rope & Individual Training	Familiarization Rope: Bonobos received time to play with rope to reduce play behaviour afterwards Training: One session of six trials Criteria to pass into test: none, all tested	Familiarization Rope: None administered, no play behaviour Training: Up to five sessions with multiple trials Criteria to pass into test: six successful consecutive trials in one session
Cooperation "warm up"	“Three-trials-success” criteria of successful cooperation with the given partner in condition dispersed divisible (up to 13 trials per dyad) provided to bonobos at both facilities and chimpanzees at WKPRC before administering cooperation experiment (Exp. 3) that included condition dispersed divisible and clumped .	No dyadic “warm-up” to either species, only solitary pre-test trials for rope-pulling understanding provided to both species (see above)
Accessibility of food	Width of mesh allowed apes to reach for food by sticking their arm through the mesh. ⁽¹⁾	Width of mesh is narrower: Apes can reach for food by sticking their fingers through the mesh.

(1) No published information provided for WKPRC apes

Table 1b. Population and sampling differences between the original and the current study.

	Hare et al, 2007	Current study
Chimpanzee subjects	Subspecies P. schweinfurthii at Ngamba and P. verus at WKPRC.	Subspecies P. verus and four females are verus-schweinfurthii hybrids.
Sample procedure	Exp. 1 & 2: Lola ya Bonobo: Bonobos selected from several populations of 12–20 individuals. WKPRC: Bonobos lived together in social group. Ngamba Island: Chimpanzees selected from one population of 39 individuals. Exp. 3: Most tolerant chimpanzee dyads selected of previous population and additionally three tolerant dyads at WKPRC selected. Bonobo pairs of previous population were selected to match the age and sex of the most tolerant chimpanzee pairs.	WKPRC: Both species sampled at same facility. Bonobos were selected from population of 12 individuals, and chimpanzees from population of 19 individuals. In both experiments, as many dyad combinations were sampled as was allowed by the current handling regime. Data collection was immediately stopped if either partner showed signs of fear towards being released into the same room with the other.
Experimental history	Bonobos and chimpanzees experimentally naïve in Exp. 1 (co-feeding task) and Exp. 2 (cooperation task, condition dispersed divisible).	Both species have extensive general experience in participating in tests, with experience across various cooperation tasks, and are trained to be separated from group.
Environment	Sanctuaries: 5-40 ha of primary tropical forest. WKPRC: Bonobos have access to indoor and outdoor enclosure (2600 m²). No information provided on chimpanzees (however see “Current study”).	Both species live in indoor enclosure (430 m ² chimpanzees, 256 m ² bonobos) and have access to outdoor enclosure (4000 m ² chimpanzees, 2300 m ² bonobos) during warm months.
Mean age and range	Exp. 1 & 2: Bonobo: 9.6 years (range: 5 – 22); Chimpanzee: 11.7 years (range: 4 – 21)	Co-feeding experiment: Bonobo: 18.9 years (range 6.2 – 36.5) Chimpanzee: 25.9 years (range 3.2 – 43.5)

	Exp 3: Bonobo: 7.9 years (range: 5 - 20) Chimpanzee: 7.5 years (range: 5 - 12)	Cooperation experiment (subsample based on passing solitary training, only administered to adult apes): Bonobo: 22.5 years (range 13.6 - 36.5) Chimpanzee: 30.4 years (range 15.1 - 43.5) Age is incorporated in the statistical models.
Kinship	Selection of only non-kin dyads.	Some dyads related; incorporated in the statistical models.
Test location	15 m ² testing room and open space in sanctuaries, 9 m ² testing room at WKPRC.	7 m² testing room for bonobos with available adjacent rooms of 6,3 m² to left and of 11,3 m² to right. 13,32 m² testing room for chimps with available adjacent rooms of 6,8 m² to left and right.
Dyad composition	In Exp. 3 (cooperation task) same partners used in two unique dyads.	To increase effective sample size, we tested apes in multiple dyads.

Appendix H

Universiteit Utrecht

Suska Nolte, Ph.D.
Lilienstr 35
04315 Leipzig
Germany

+49-157-52461971
suskanolte@gmx.de

October the 16th, 2022

Dear Editor,

With this letter, we submit the full manuscript of *“Does Tolerance Allow Bonobos to Outperform Chimpanzees on a Cooperative Task? A conceptual replication of Hare et al., 2007”* for consideration at Stage 2 in *Royal Society Open Science*.

We are grateful for the detailed feedback on our manuscript. Moreover, we are happy that the editor and reviewers recognize the value of our attempted replication and accepted the manuscript at Stage 1. We hope that our full manuscript fulfills the criteria for publication in *Royal Society Open Science*.

On behalf of all the co-authors,

Sincerely,

Suska Nolte